# Long-Context Attention Benchmark: From Kernel Efficiency to Distributed Context Parallelism

**Tao Bu**[1][*]   **Qiangang Wang**[1][*]   **Bowen Zeng**[2]   **Hanwen Sun**[3]
**Yunpeng Huang**[1]   **Chun Cao**[1]   **Jingwei Xu**[1][†]

[1]State Key Laboratory for Novel Software Technology, Nanjing University, China
[2]Zhejiang University, China
[3]Peking University, China
{butao,qgwang,hyp}@smail.nju.edu.cn, zbw.cs@zju.edu.cn
sunhanwen@stu.pku.edu.cn,  {caochun,jingweix}@nju.edu.cn

## Abstract

Transformer-based large language models (LLMs) have achieved remarkable success, yet their standard softmax-operator-based attention mechanism incurs quadratic computation and memory costs with respect to sequence length, posing a major bottleneck for long-context training. Prior work tackles this challenge along two directions: (1) kernel-level optimizations, which accelerate dense and sparse attention operators; and (2) module-level strategies, often referred to as distributed attention or context parallel training, which scale attention across multiple devices. However, systematic evaluation still remains limited: operator-level comparisons are often incomplete, while context parallel strategies are typically framework-specific, with unclear performance analysis across contexts. To address these gaps, we propose a unified benchmark that integrates representative attention kernels and context parallel mechanisms with a modular and extensible interface for evaluation. The benchmark evaluates methods along two critical dimensions: (1) attention mask patterns, which strongly affect efficiency, scalability, and usability, and (2) sequence length and distributed scale, which determine performance under extreme long-context training. Through comprehensive experiments on the cluster of up to 96 GPUs, our benchmark enables reproducible comparisons, highlights method-specific trade-offs, and provides practical guidance for designing and deploying attention mechanisms in long-context LLM training.

## 1 Introduction

The Transformer architecture, powered by the attention mechanism, has become the foundation of large language models (LLMs) (Achiam et al., 2023; Team et al., 2023; Dubey et al., 2024). With the guidance of the scaling law (Kaplan et al., 2020; Tay et al., 2022), current state-of-the-art LLMs, such as GPT (Agarwal et al., 2025), Gemini (Team et al., 2024), and DeepSeek (Liu et al., 2024), contain billions of parameters and are trained on trillions of data using large-scale distributed GPU clusters. However, as model size and training data continue to grow, the computational and memory costs of conventional attention scale quadratically with sequence length, posing a fundamental efficiency bottleneck for large-scale LLM training (Dai et al., 2024). Although the context window of LLMs has expanded dramatically from 4K tokens to 128K (Grattafiori et al., 2024), 1M (Yang et al., 2025a), and even 10M (Team et al., 2024) tokens, the design and performance characteristics of long-context attention mechanisms at these scales in distributed training remain insufficiently understood (Gao et al., 2024b).

Recent research on efficient long-context attention at scale has progressed along two main directions. The first focuses on kernel-level optimizations (Zhang et al.), such as dense and sparse kernels,

---

[*]Equal contribution
[†]Corresponding author

reducing attention complexity on a single GPU. The second emphasizes module-level designs, or context parallelism (Duan et al., 2024), which partition long sequences (e.g., 32K–128K tokens) across multiple GPUs with tailored communication and scheduling for scalability. Despite these advances, comprehensive analyses of long-context attention mechanisms remain lacking. Attention operators differ significantly in their support for mask patterns, and even the same operator can exhibit substantial performance variation across masks. Currently, no unified evaluation has been established. Furthermore, existing context parallel attention mechanisms are often tightly integrated with specific training frameworks (e.g., DeepSpeed (Rasley et al., 2020) and InternEvo (Chen et al., 2024)), which limits reusability and hinders systematic comparison. As a result, researchers lack a clear understanding of the trade-offs between methods, and practitioners have no reliable benchmark or reference to guide the selection of attention mechanisms in long-context training.

To address these issues, we collect representative attention operators and context parallel mechanisms, and design a unified framework to systematically benchmark their capabilities, limitations, and potential risks in ultra-long context training. In our framework, we establish a unified data preparation interface that supports both non-distributed kernels and context parallel attention mechanisms, enabling fair evaluation across methods. Specifically, for non-distributed kernel scenarios, we integrate a variety of dense and sparse attention kernels, implementing standardized interfaces that eliminate inconsistencies in data representation and ensure comparability under the benchmark. For distributed scenarios, we reconstruct and optimize representative context parallel attention mechanisms within the unified framework, providing efficient, scalable implementations with modular interfaces. Building on the foundation, we conduct large-scale experiments and in-depth analyses along two critical dimensions: (1) attention mask patterns (up to 14 mask patterns), which are often overlooked but have a significant impact on efficiency, scalability, and usability, and (2) context length and distributed scale, where we systematically evaluate performance trends and capability limits as both the input length and distributed scale grow, reaching up to 768K on 96 GPUs. We hope our results offer valuable insights for research on long-context training of large models, as well as for the design and development of next-generation distributed attention mechanisms. Our contributions are threefold:

1. Unified benchmarking: we provide a standardized framework with consistent data preparation for fair evaluation of attention mechanisms across diverse long-context scenarios.

2. Modular components: we unify dense and sparse kernels under a high-level modular interface, and provide optimized distributed attention in terms of context parallelism.

3. In-depth analysis: we conduct extensive experiments across dense long-context scenarios to identify key factors affecting attention efficiency and scalability, providing valuable guidance for ultra-long context training and development.

## 2 LONGCA-BENCH

LongCA-bench is a benchmark designed to evaluate the efficiency of long-context attention across both single-device kernels and distributed context parallel mechanisms. The benchmark consists of three core components: (1) a unified data preparation interface that standardizes preprocessing, (2) a unified input representation interface that supports 7 dense and 5 sparse attention kernels, and (3) an optimized context parallelism framework that incorporates 5 distributed attention mechanisms. Together, these components provide a systematic and extensible platform for analyzing long-context attention, enabling fair comparisons across operator-level efficiency and distributed scalability.

### 2.1 DATA PREPARATION

We first describe the data preparation process in the benchmark. To generate inputs practical for long-context attention benchmarking, we introduce a dedicated data preparation interface. Rather than directly using the downstream datasets, our interface combines *diverse mask patterns* with *variable lengths of sequence sampling*, ensuring that the evaluation data accurately reflects the characteristics and challenges of long-context training.

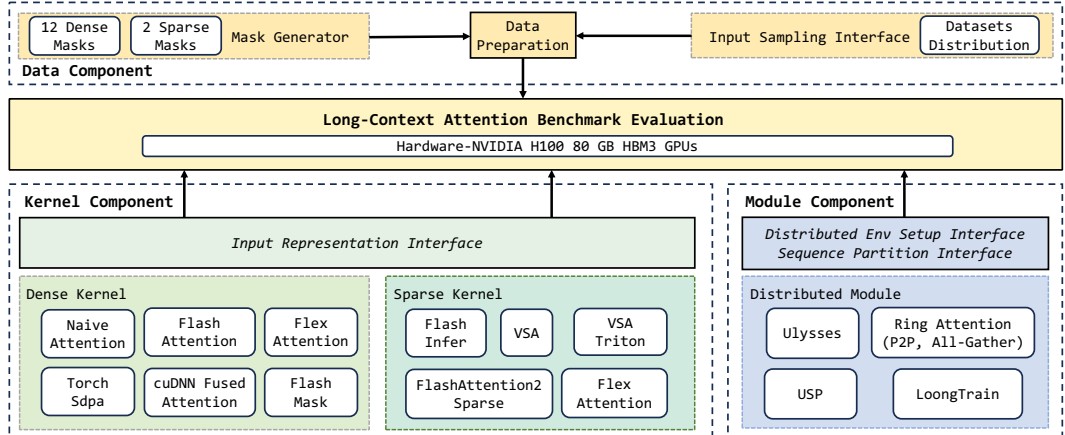

Figure 1: The architecture of LongCA benchmark

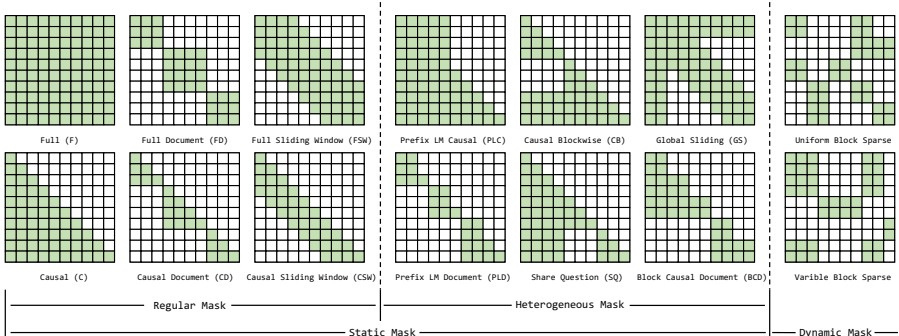

Figure 2: Attention mask patterns

### 2.1.1 INPUT MASK PATTERNS

Different tasks require specific mask types based on the training scenario. In LongCA-bench, we categorize a total of 14 mask patterns into two major classes (see Figure 2): 12 static masks (6 regular and 6 heterogeneous), and 2 dynamic masks. The key distinction lies in whether the mask can be predetermined before training or must be generated adaptively during the training process.

**Static regular mask.** The FULL and CAUSAL masks are the most widely used in training (Vaswani et al., 2017). Considering the document-level variants, FULL DOCUMENT and CAUSAL DOCU-MENT are employed for efficient sequence packing and in-batch/in-token processing (Krell et al., 2021; Dehghani et al., 2023). In addition, by applying the sliding window variants, FULL SLIDING WINDOW and CAUSAL SLIDING WINDOW can leverage sparsity to balance computational cost and token coverage (Beltagy et al., 2020).

**Static heterogeneous mask.** SHARED QUESTION mask used in reward models allows multiple answers to share the same question (Ouyang et al., 2022). GLOBAL SLIDING mask is designed to effectively capture both global context and local details (Zaheer et al., 2020). CAUSAL BLOCK-WISE mask, which is widely adopted in in-context learning, restricts demonstrations to local blocks while letting the test example attend globally, supporting long-context evaluation (Bertsch et al., 2024). The PREFIX LM CAUSAL and PREFIX DOCUMENT masks are specifically tailored to introduce a prefix for language modeling tasks (Raffel et al., 2020). BLOCK CAUSAL DOCU-MENT mask combines the block and document concepts and is widely used in multimodal model training (Zewei & Yunpeng, 2025).

**Dynamic mask.** In long-context scenarios, block sparse masks reduce computational latency and memory usage by restricting attention computation to the most salient blocks of the input. Since

the selected blocks depend on the contextual input, the mask pattern varies across examples. Block sparse masks have been widely adopted in both natural language processing (Lu et al., 2025; Yuan et al., 2025; Xu et al., 2025; Guo et al., 2024; Ye et al., 2025) and visual generation (Zhang et al., 2025d; Zewei & Yunpeng, 2025; Yang et al., 2025b). We categorize block sparse masks into two types: uniform and variable (Figure 2). The uniform block mask applies attention blocks of a fixed size (e.g., 64×64) across the entire attention map and computes the selected blocks during attention. In contrast, the variable block mask provides greater flexibility by allowing blocks of different sizes, offering a more efficient and expressive representation of sparse attention patterns.

### 2.1.2 INPUT DATA SAMPLING

**Dense data sampling.** The mask specifies the area of attention interactions within a context window. As the context window expands (e.g., from 8K to 768K), input data sampling becomes crucial. To ensure that the benchmark data reflects realistic training scenarios, we analyzed several widely used public pretraining datasets, including Pile (Gao et al., 2020), ProLong64K (Gao et al., 2024a), ProLong512K (Gao et al., 2024a), Slimpajama-Per-Source-Length-Upsample (Yaofu, 2024), Open-WebText (Gokaslan et al., 2019), and C4 (Raffel et al., 2020) (see Appendix A.2 for details). Following prior findings (Fu et al., 2024; Gao et al., 2024b), we note that: (1) language model training typically requires datasets from diverse sources; (2) extending the context length requires maintaining domain diversity while upsampling long-sequence samples; and (3) mixing long-context sources (e.g., code repositories and books) with high-quality short-text data improves long-context modeling without sacrificing overall performance. In our benchmark, the data sampling method therefore uses the Pile dataset for samples up to 8K, ProLong64K for long-context samples up to 64K, and Pro-Long512K for ultra-long samples up to 768K. This combination ensures that evaluation data reflects realistic training scenarios across different context scales.

**Sparse data sampling.** Mask generation for block sparse attention in our benchmark follows standard methodology (Xia et al., 2025; Lu et al., 2025; Yuan et al., 2025; Zhang et al., 2025d). The attention matrix is first partitioned into a two-dimensional grid of blocks, with either uniform or variable pre-defined block sizes. An importance score is then computed for each block in the grid, which in multi-head attention may be assigned on a per-head or per-group basis. Guided by a target sparsity ratio, a top-$K$ selection is performed for each query block to identify the most salient key blocks for attention computation. In our benchmark, however, we simplify the process to specifically evaluate kernel performance under varying sparsity levels (e.g., 0.2, 0.5, and 0.8). Instead of computing explicit important scores, we simulate the scoring and selection process by randomly generating block masks to achieve the desired sparsity. To create real-world workloads, we evaluate sequence lengths from 32K to 128K, sampled at 32K intervals. This range is derived from analyzing prominent benchmarks for block sparse attention's primary applications in video generation (e.g., VBench (Huang et al., 2024)) and LLMs (e.g., RULER (Hsieh et al., 2024)). Our evaluation covers both MHA and GQA using uniform block sizes of 64×64 and 128×128 (see Appendix A.3 for full results).

## 2.2 ATTENTION KERNEL

Efficient attention kernels aim to reduce time and memory complexity without compromising expressiveness. The most straightforward approach is hardware acceleration, which speeds up computation without altering the original attention logic. Another common strategy leverages the inherent sparsity of attention, skipping unnecessary computations, often guided by a dynamic sparse mask.

**Dense attention kernel.** We integrate seven dense attention kernels and categorize their support for different mask types (see Table 1). Since kernels often apply for different requirements on data structure and mask formats, we implement dedicated adapter interfaces for each. These interfaces generate kernel-specific input representations from a unified data format, eliminating inconsistencies in data expression across kernels. This design simplifies input preparation for diverse mask scenarios, ensures comparability within the benchmark, and provides a unified solution for future kernel extensions.

As baselines in our benchmark, we include both the step-by-step naïve attention and PyTorch's fused scaled dot product attention (SDPA) (PyTorch Contributors, 2024b), both construct full 2D masks and theoretically support arbitrary masking patterns. For hardware-optimized kernels, we

Table 1: Dense kernel support across mask patterns

| Mask Type | Dense Kernels | | | | | | |
|---|---|---|---|---|---|---|---|
| | Naive-Torch | SDPA | FA2 | FA3 | cuDNN-Fused-Attn | FlexAttn | FlashMask |
| FULL | ✓ | ✓ | ✓ | ✓ | ✓ | ✓ | ✓ |
| CAUSAL | ✓ | ✓ | ✓ | ✓ | ✓ | ✓ | ✓ |
| FULL SLIDING WINDOW | ✓ | ✓ | ✓ | ✓ | ✓ | ✓ | ✓ |
| CAUSAL SLIDING WINDOW | ✓ | ✓ | ✓ | ✓ | ✓ | ✓ | ✓ |
| FULL DOCUMENT | ✓ | ✓ | ✓ | ✓ | ✓ | ✓ | ✓ |
| CAUSAL DOCUMENT | ✓ | ✓ | ✓ | ✓ | ✓ | ✓ | ✓ |
| SHARE QUESTION | ✓ | ✓ | ✗ | ✗ | ✗ | ✓ | ✓ |
| CAUSAL BLOCKWISE | ✓ | ✓ | ✗ | ✗ | ✗ | ✓ | ✓ |
| GLOBAL SLIDING | ✓ | ✓ | ✗ | ✗ | ✗ | ✓ | ✓ |
| PREFIX LM CAUSAL | ✓ | ✓ | ✗ | ✗ | ✗ | ✓ | ✓ |
| PREFIX LM DOCUMENT | ✓ | ✓ | ✗ | ✗ | ✗ | ✓ | ✓ |
| BLOCK CAUSAL DOCUMENT | ✓ | ✓ | ✗ | ✗ | ✗ | ✓ | ✓ |

Table 2: Characteristics of sparse kernels

| Characteristics | Sparse Kernels | | | | |
|---|---|---|---|---|---|
| | VSA | Triton VSA | FA2 Sparse | FlexAttention | FlashInfer |
| Uniform/Variable Masks | Uniform only | Uniform only | Uniform only | Both | Both |
| Forward/Backward | Both | Both | Forward only | Both | Forward only |
| Block Size | 64 only | 64 only | 128 only | Arbitrary | Arbitrary |
| GQA Support | ✗ | ✗ | ✓ | ✓ | ✓ |
| GPU Support | $\geq$ sm90 | $\geq$ sm80 | $\geq$ sm80 | $\geq$ sm80 | $\geq$ sm80 |
| Performance ↑ | High | Medium | Medium | Low | Medium |
| Memory Overhead ↓ | Low | Low | Low | High | Medium |

integrate the FlashAttention series, including FA (Dao et al., 2022), FA2 (Dao, 2023), FA3 (Shah et al., 2024), as well as cuDNN fused kernels (NVIDIA Corporation, 2025). These kernels employ advanced techniques such as shared memory, block-wise partitioning, warp scheduling, FP8, and asynchronous processing. For flexible kernels, we integrate FlexAttention (Flex) (Dong et al., 2024), a general fused operator with memory complexity close to $O(S^2)$, where $S$ denotes the sequence length, that generates specialized kernels based on per-position boolean functions and enables compatibility with arbitrary masks. We also include FlashMask (Wang et al., 2025a), which introduces a column-wise representation to optimize heterogeneous computation.

**Sparse attention kernel.** Sparse attention significantly reduces the computational complexity of attention for long sequences. Due to its versatile mask representation, block sparse attention is widely used in state-of-the-art sparse attention methods (Lu et al., 2025; Zhang et al., 2025b; Yuan et al., 2025; Zhang et al., 2025d; Xu et al., 2025). Therefore, our benchmark incorporates five block sparse attention kernels to evaluate long-context sparse attention.

We categorize these kernels into two main types. The first type consists of dedicated block sparse attention kernels, which are highly optimized for sparse patterns with uniform block sizes (e.g., 64×64). Representative implementations include VSA (Zhang et al., 2025d), its Triton-based version (Triton VSA), and the FlashAttention-2-based block sparse attention (FA2 Sparse) (Guo et al., 2024). The second type comprises general-purpose sparse attention kernels, which offer greater flexibility and support arbitrary block structures. They are compatible with both uniform and variable block sparse masks. This category includes FlexAttention (Dong et al., 2024) and FlashInfer (Ye et al., 2025). These kernels exhibit different characteristics, as summarized in Table 2 (refer to Appendix A.4 for details). We evaluate performance through comparisons using two mask types: uniform block mask and variable block mask. Note that backward computation in training is supported by only a limited set of block-sparse attention kernels. For comprehensiveness, we select FA2 Sparse and FlashInfer, two inference-side methods, for comparisons in our benchmark.

## 2.3 DISTRIBUTED ATTENTION MECHANISM

In our benchmark, we reproduce and optimize 5 representative distributed attention mechanisms under a unified framework, including Ulysess, Ring P2P, Ring All-Gather, USP, and LoongTrain. We establish a unified infrastructure that standardizes distributed setup and sequence partitioning, ensuring a consistent invocation protocol across all methods. The integrated distributed attention mechanisms can be categorized into three architectural designs:

**All-to-all based design.** DeepSpeed's Ulysses (Jacobs et al., 2023) partitions both the sequence and head dimensions in multi-head attention, using All-to-All communication to switch parallel dimensions. This approach is simple, general, and numerically precise, but the scalability is constrained by the number of attention heads, particularly under GQA, MQA, or tensor parallelism.

**Ring P2P based design.** Ring P2P (Liu et al., 2023) uses multi-round ring-structured point-to-point communication, while Ring All-Gather (NVIDIA, 2025) performs a single all-gather of key-value tensors, relying on ring topologies. These approaches exhibit strong scalability and naturally overlap computation with communication via pipelining. However, they suffer from lower efficiency and potential numerical error accumulation.

**Hybrid design.** USP (Fang & Zhao, 2024) and LoongTrain (Gu et al., 2024) extend Ulysses and ring-based designs into a two-dimensional scheme. An inner layer applies Ulysses with All-to-All for intra-node bandwidth, while an outer layer uses ring-based attention to enhance scalability and enable compute–communication overlap. LoongTrain further proposes DoubleRing Attention, enhancing Ring P2P with a two-level sliding window to improve communication efficiency.

In our reproduction and optimization, we draw inspiration from TransformerEngine (NVIDIA, 2025), achieving perfect load balancing through double-parallel partitioning combined with head-to-tail reordering (zhuzilin, 2024). We also incorporate optimizations such as double buffering and multi-stream overlap of computation. For each method, we implement backend support for both Flash Attention v3 and cuDNN Fused Attention operators. We extend the input layout to a variable-length (varlen) format, allowing multiple sequences of different lengths to be concatenated along the sequence dimension while handling padding tokens. This ensures the flexibility and usability of varlen inputs under different distributed scales. Since varlen inputs can introduce substantial synchronization and waiting overhead across devices, we precompute the necessary meta-information for all distributed strategies as a one-time preprocessing step, thereby minimizing distribution-related performance degradation. Despite these extensions and optimizations, our benchmark remains constrained by the underlying distributed attention designs, thus currently supporting only FULL, CAUSAL, FULL/CAUSAL DOCUMENT masks.

## 3 EVALUATION

In this section, we present experiments evaluating the speed and memory efficiency of different attention methods under long-context scenarios. The speed is measured in TFLOPs/s metric, and peak memory usage is reported in gigabytes (GB). Kernel performance is evaluated on a single GPU, while the performance of distributed context parallel attention is assessed across multi-GPU clusters of varying scales. All context parallel evaluations are performed on NVIDIA H100 GPUs (80GB HBM3). Kernel-level experiments are conducted on both NVIDIA H100 and NVIDIA A800 GPUs. We primarily present and analyze the results on H100 here, with most A800 results deferred to the appendix. The code is publicly available for the community[1].

### 3.1 DENSE ATTENTION KERNEL PERFORMANCE

We evaluate dense kernels across 12 static mask configurations to assess both expressiveness and efficiency. Sequence lengths range from 1K to 48K, with BFloat16 precision, a hidden dimension of 128, and two head settings: GQA (64:8) and MHA (64:64)[2]. We record forward and backward throughput as well as peak memory usage (see Appendix A.5 for full results on H100, see Appendix A.9 for full A800 results).

Figures 3 and 8 report TFLOPs at 8K sequence length under GQA (64:8), where ✗ denotes unsupported configurations. The six groups on the left correspond to static regular masks, and the remaining on the right show static heterogeneous masks. Note that the FA series and cuDNN fused kernels do not support heterogeneous masks. In particular, cuDNN fused kernel does not support the FULL SLIDING WINDOW mask with GQA (64:8), though other configurations are recommended.

Although the naive implementation and Torch SDPA theoretically support arbitrary masks, their quadratic complexity leads to severe efficiency degradation and excessive memory overhead, mak-

---

[1]The implementation is accessible at: https://github.com/NJUDeepEngine/LongCA-bench
[2]$\text{head}_q : \text{head}_{kv}$

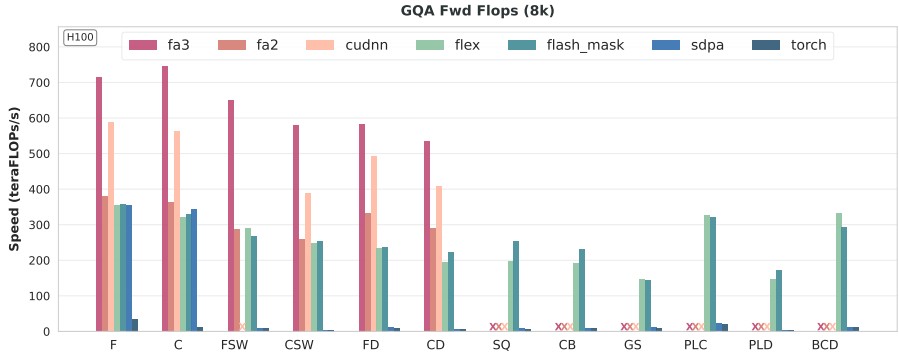

Figure 3: Forward TFLOPs of dense kernels with different masks (8K length)

ing them impractical in long-context settings. Under computation-intensive dense settings (e.g., FULL or CAUSAL), SDPA achieves performance comparable to general fused operators such as FlexAttention. These results provide a baseline for fused operators without hardware-specific optimizations. FlashMask, another generic fused operator, leverages a column-wise mask representation to mitigate computational sparsity. While optimized for heterogeneous masks, its column-wise representation cannot cover all scenarios, making it less general than FlexAttention.

For regular scenarios, FA series and cuDNN fused attention are all hardware-optimized kernels. On H100 GPUs, FA3, specifically optimized for the Hopper architecture, achieves the best performance. cuDNN fused attention supports multiple architectures but imposes stricter constraints on data patterns (e.g., GQA (64:8) with FULL SLIDING WINDOW). While some of these limitations can be circumvented by preprocessing techniques such as padding, doing so introduces extra overhead. Note that although FA2 and cuDNN fused attention yield lower performance, kernel selection should be guided by the target hardware architecture. In Figures 4 and 5, we present the main experimental results on both the H100 and A800.

## 3.2 SPARSE ATTENTION KERNEL PERFORMANCE

To comprehensively evaluate the functionality and computational efficiency of various sparse kernels, we include VSA (Zhang et al., 2025d), Triton VSA, FA2 sparse (Guo et al., 2024) and Flash-Infer (Ye et al., 2025) in our evaluation (see Appendix A.6 for full results). We perform kernel-level evaluations across two kinds of block sizes (64 and 128), both forward and backward computation, two attention variants (MHA (64:64) and GQA (64:8)), and sequence lengths ranging from 32K to 128K. Note that FlexAttention (Dong et al., 2024) is excluded due to severe out-of-memory (OOM) issues originating from its mask representations.

From a functionality perspective, comparisons in Figures 4 (a) and (b) reveal that FA2 sparse does not support a block size of 64, while FlashInfer lacks backward computation. As shown in Figures 4 (a), (c), and (d), VSA does not support a block size of 128. Due to its specific design for the MHA architecture in video diffusion models, VSA does not currently support GQA. In contrast, FlashInfer is prone to OOM errors at longer sequence lengths and smaller block sizes, stemming from the substantial metadata storage it requires.

These limitations highlight the need for further engineering optimizations in block-sparse kernels. Backward computation is essential for trainable sparse attention, particularly in GQA and MHA. Flexibility across block sizes is required to support diverse sparse attention designs, and memory challenges in block-sparse mask representations also need to be addressed.

From a performance perspective, Figures 4 (a) and (b) show that VSA outperforms both Triton VSA and FlashInfer, while Figures 4 (c) and (d) indicate that FlashInfer outperforms fa2 sparse. Across all kernels, the forward pass consistently achieves a higher percentage of theoretical TFLOPs than the backward pass (with theoretical TFLOPs taken from FA3 in Figure 8). Additionally, Figures 4 (c) and (d) show minimal performance differences between MHA and GQA, though GQA achieves better GPU memory efficiency. By comparing Figures 4 (a) and (c), we observe that FlashInfer

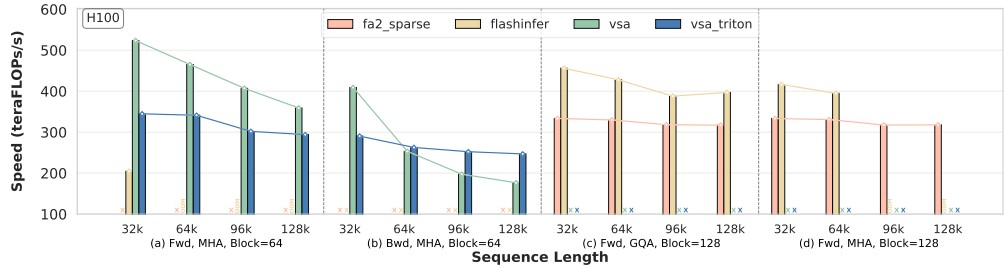

Figure 4: Performance results (TFLOPs) of sparse kernels with a 50% sparsity ratio on H100 GPU

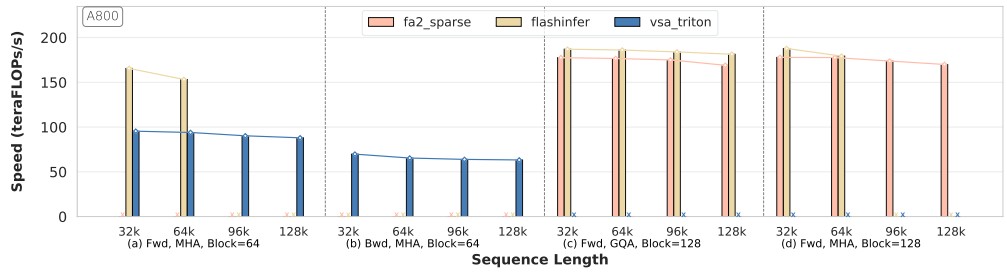

Figure 5: Performance results (TFLOPs) of sparse kernels with a 50% sparsity ratio on A800 GPU

performs significantly better with a block size of 128 than with 64, suggesting that larger block sizes are more effective for achieving higher performance.

Overall, these results demonstrate that performance in block sparse attention is significantly improved through specialization, where kernels tailored to particular parameters (e.g., block size or hardware architecture) consistently outperform general implementations. Meanwhile, the backward pass remains a major bottleneck, underscoring an urgent need for optimization. A key future direction is the development of more flexible, comprehensive kernels that deliver high performance across a wide range of block sizes. Achieving this goal requires moving beyond single-parameter tuning toward deeper, hardware-level optimizations.

### 3.3 CONTEXT PARALLEL ATTENTION PERFORMANCE

We evaluate four mask patterns: FULL, CAUSAL, FULL DOCUMENT, and CAUSAL DOCUMENT. The per-device sequence length is fixed at 8K with hidden size 128, validated under the GQA (64:8) setting. Experiments are conducted on NVIDIA H100 GPUs, scaling from 8 to 96 GPUs across 12 servers, with total context windows from 64K to 768K. Since Ulysses requires divisibility constraints, GQA is converted to MHA by replicating KV heads (✗ indicates divisibility failure). Performance under FULL DOCUMENT is shown in Figures 6 and 24, with additional details provided in Appendix A.7.

Context parallel attention inevitably involves distributed communication, raising two major concerns: (1) whether communication effectively overlaps with computation, and if not, how efficient the communication is; and (2) whether workload balance is achieved in terms of both data volume and computation across devices. Ideally, a context parallel strategy should behave close to a non-distributed setting. Inter-node communication constitutes the dominant bottleneck compared to computation and intra-node communication. In our experiments, we fix the large-load AllToAll groups within the mixed architecture to 8 per node, while the small-load P2P groups are placed across nodes and scale with the number of nodes. For the secondary P2P communication groups in LoongTrain, we adopt a balanced configuration (e.g., $12 = 3 \times 4$) to maximize inter-node bandwidth utilization. All experiments are performed on the FA3 backend for consistency.

Ulysses' AllToAll communication is entirely exposed outside the computation. Thanks to its collective communication pattern (NVIDIA Corporation, 2020) with low communication overhead and

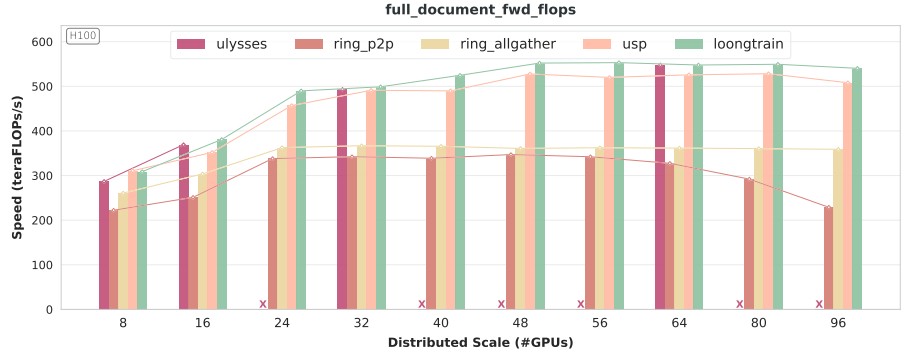

Figure 6: Forward TFLOPs of Context Parallel Attention on FULL DOCUMENT

Table 3: Communication volume per device of Context Parallel Attention

| | | | | Context Parallel Attention | |
|---|---|---|---|---|---|
| Stage | Ulysess | Ring All-Gather | Ring P2P | USP | LoongTrain |
| Forward | $\frac{N-1}{N^2}t(h_{kv}+h_q)d*2$ | $\frac{N-1}{N}th_{kv}d*2$ | $\frac{N-1}{N}th_{kv}d*2$ | $\frac{N_a-1}{N_p*N_a^2}t(h_{kv}+h_q)d*2+\frac{N_p-1}{N_p*N_a}th_{kv}d*2$ | $\frac{N_a-1}{N_p*N_a^2}t(h_{kv}+h_q)d*2+\frac{N_p-1}{N_p*N_a}th_{kv}d*2$ |
| Backward | $\frac{N-1}{N^2}t(h_{kv}+h_q)d*2$ | $\frac{N-1}{N}th_{kv}d*4$ | $\frac{N-1}{N}th_{kv}d*4$ | $\frac{N_a-1}{N_p*N_a^2}t(h_{kv}+h_q)d*2+\frac{N_p-1}{N_p*N_a}th_{kv}d*4$ | $\frac{N_a-1}{N_p*N_a^2}t(h_{kv}+h_q)d*2+\frac{2N_p+k-1}{N_p*N_a}th_{kv}d*2$ |

its head-sharded computation pattern leading to perfectly balanced workloads, Ulysses still delivers solid performance. However, its scalability is bounded by the number of attention heads. A load-balanced Ring P2P ensures that each GPU processes the same amount of computation and communication per iteration. However, Ring P2P communication is mask-independent, always transferring in a fixed ratio of $D/N$, where $D$ is the total data and $N$ is the world size, meaning performance depends entirely on the amount of computation workload. Ring P2P performs optimally in the FULL scenario. However, in the DOCUMENT scenario, variable-length padding depends on scale and sampling, leading to noticeable per-GPU computation variation and performance fluctuations.

The hybrid architecture alleviates the above issues. While the intra-node AllToAll communication group still remains exposed outside computation, its per-communication volume is reduced from $D \times (N - 1)/N$ in Ulysses to $D \times (8 - 1)/N$ per group (one-way). Meanwhile, the inter-node Ring P2P computation volume increases from $D/N$ in pure Ring P2P to $D/K$, enabling USP and LoongTrain to achieve optimal performance improvements, where $K$ denotes the size of the inter-node communication group. Additionally, LoongTrain introduces a secondary P2P architecture to further improve inter-node bandwidth utilization, providing modest forward speedups compared to USP. However, because the secondary architecture involves extra window synchronization, the Ring backward pass cannot directly continue from the forward state, negating the overall performance gains. Overall, the experimental results demonstrate that fully leveraging MHA by first partitioning the heads yields significant performance benefits.

### 3.4 COMMUNICATION ANALYSIS FOR CONTEXT PARALLEL ATTENTION

We provide a theoretical analysis of the communication volume of the entire pipeline per device for each context parallel attention. For each mechanism, we describe the communication operators involved, their invocation timing, call frequency, and data types, and we further report the corresponding theoretical communication volume per device. We further include an analysis of communication–computation overlap. We denote $t$ as the total sequence length, $h_q$ as the number of query heads, $h_{kv}$ as the number of key or value heads, and $d$ as the hidden dimension. Full detailed analysis are provided in Appendix A.8, we present only the results in Table 3.

## 4 RELATED WORK

**Long context language modeling.** Models such as BERT (Devlin et al., 2019) and GPT (Brown et al., 2020) can process thousands of tokens, supporting document and dialogue level tasks, with full document understanding and long range retrieval emerging as key challenges. Recently, model con-

text windows have expanded dramatically, from 4K tokens to 128K (Dubey et al., 2024), 1M (Yang et al., 2025a), and even 10M tokens (Team et al., 2024). The ability to model ultra-long contexts enables continuous reference, reasoning, and summarization over extended input sequences. This enhances advanced capabilities such as long-text reasoning (Guo et al., 2025; Muennighoff et al., 2025), improved in-context learning (Li et al., 2025; Team et al., 2024), efficient information compression (Lee et al., 2024; Wang et al., 2024), and multimodal understanding (Weng et al., 2024).

**Attention kernels.** Attention is the core component in Transformers with the time complexity of $O(n^2)$ in terms of context length. Hardware-efficient attention leverages hardware features to reduce the time and memory costs. Dao et al. (2022); Dao (2023); Shah et al. (2024) employs matrix tiling and kernel fusion. Zhang et al. (2024b;a; 2025a) uses quantization to leverage low-bit Tensor Cores. Sparse Kernels use the inherent sparsity (Child et al., 2019; Zhang et al., 2025b) of the attention map $P = \text{Softmax}(QK^\top/\sqrt{d})$ to accelerate computation. Other directions include KV cache compression (Zhao et al., 2023a) via weight sharing (Ainslie et al., 2023) or low-rank decomposition (Liu et al., 2024) to reduce memory overhead without extra computation.

**Parallelism for distributed training.** Various parallel paradigms have been developed to tackle resource challenges in large-scale distributed model training. Data parallelism (PyTorch Contributors, 2024a; Rajbhandari et al., 2020; Zhao et al., 2023b) partitions data along the batch dimension. Tensor parallelism (Shoeybi et al., 2019; Xu & You, 2023; Wang et al., 2022), pipeline parallelism (Huang et al., 2019; Li & Hoefler, 2021; Narayanan et al., 2019; Qi et al., 2024) [39–42], and expert parallelism (Gale et al., 2023; Hwang et al., 2023; Li et al., 2023; Liu et al., 2024) partition model parameters along different dimensions. Hybrid parallel strategies (Smith et al., 2022; Ge et al., 2025; Wang et al., 2025b) are used to meet diverse needs and balance computation and memory. However, these strategies cannot fully address activation memory overhead from ultra-long sequences. Context parallelism (Korthikanti et al., 2023; Li et al., 2021) partitions data by sequence, but faces challenges in computation–communication overlap, balancing, scalability, and usability; many designs remain underexplored, and near-linear scalability is still difficult.

## 5 Conclusion

The complexity of distributed environments is far greater than that of single-device settings. In ultra-long context training, selecting or developing appropriate kernels and context parallel strategies poses significant challenges and requires substantial effort and resources. To address this, we present a fair and unified benchmark for attention mechanisms in ultra-long context training, covering the spectrum from single-device kernels to large-scale distributed context parallel methods. Although our work has limitations, it aims to improve fairness in comparing different approaches, expose their performance trade-offs and constraints, and provide objective references to guide future research and development in ultra-long context training.

## Acknowledgments

We are thankful to the anonymous reviewers for their helpful comments. This work is supported by Jiangsu Science and Technology Major Project (#BG2025005), the Collaborative Innovation Center of Novel Software Technology and Industrialization, and Jiangsu Wukong Intelligent Computing Digital Technology. Jingwei Xu (`jingweix@nju.edu.cn`) is the corresponding author.

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

# A APPENDIX

## A.1 LLM USAGE

In this work, LLMs served as supporting tools for manuscript polishing. They were mainly used to enhance textual clarity, refine grammar, and adjust minor phrasing. All conceptual development, baseline optimization, benchmark design, and experimental analysis were conducted independently by the authors, without reliance on LLMs.

## A.2 DETAILS OF DENSE DATA SAMPLING

In large language model construction, the quality and diversity of the training data are crucial for enhancing model performance (Gunasekar et al., 2023). Numerous studies have explored various methods to improve data quality. Our benchmark builds upon these efforts by systematically analyzing several publicly available, high-quality, and widely used English datasets. Full results are presented in Figure 7.

We mainly analyze the Pile and SlimPajama (Soboleva et al., 2023) datasets to study their effects on the model's short-context modeling capabilities. The Pile dataset is a large-scale, diverse English text corpus designed for training large language models, with a total size of 825 GB. It consists of 22 high-quality subsets, many drawn from academic or professional sources, including Common Crawl, Wikipedia, OpenWebText, ArXiv, and PubMed. Such diversity across multiple domains and topics substantially increases the richness and variety of the training data. SlimPajama is an open-source dataset obtained from the original RedPajama corpus through multiple preprocessing steps such as NFC normalization, cleaning, deduplication, and document interleaving, comprising a total of 627B tokens. Compared to Pile, SlimPajama contains less web data and more content from Books, ArXiv, and Wikipedia. These are high-quality long-form text sources that help improve the model's long-context modeling capabilities. Owing to its large scale, SlimPajama is not fully sampled; instead, our benchmark samples sequences of up to 8k tokens from Pile, which is sufficient to represent realistic short-context modeling scenarios.

The above data cleaning mainly focused on a limited context window (e.g., 8k). To extend the model's context window, recent studies have begun exploring data mixing strategies for long contexts. We follow the findings of (Fu et al., 2024) and (Gao et al., 2024b): (1) continual pretraining on long-context data can significantly improve the model's ability to accurately retrieve information in long contexts; (2) when extending the context length, oversampling long sequences while preserving the original domain diversity of the pretraining dataset is crucial; (3) mixing high-quality long-context sources with high-quality short-context sources is essential for enhancing long-context modeling capability while maintaining performance on short contexts. In our benchmark, we collected statistics on the publicly available long-text upsampled dataset slimpajama-per-source-length-upsample (referred to as Upsampled SlimPajama) (Yaofu, 2024) from (Fu et al., 2024), as well as the datasets prolong-data-64K (ProLong64K) (Gao et al., 2024a) and prolong-data-512K (ProLong512K) (Gao et al., 2024a) from (Gao et al., 2024b), which are used to extend the model's context window to 64K and 768K tokens, respectively.

Considering the significant differences resulting from various tokenization methods, we directly split English samples by spaces in our statistics (approximately reflecting tokenized lengths). All statistics are shown in Figure 7, with short-context and long-context distributions arranged side by side. It is worth noting that datasets are generally expressed in terms of the number of tokens rather than the number of samples. Although long-text tokens in datasets such as ProLong64 and ProLong512K account for up to 60%, their prominence in the length distribution may still be limited.

In real training scenarios, the proportion of long- and short-text tokens seen by the model should reflect the inherent distribution of the dataset. Prior to training, data preprocessing typically shuffles the dataset and packs all samples into batches, ensuring that the model eventually traverses every sample. However, in benchmark evaluation, it is impractical to cover the entire dataset in each run. Therefore, we set a fixed number of steps per evaluation. To avoid significant deviations between the sampled data and the true dataset distribution when using only a small number of steps, we adopt stratified sampling, enforcing a 6:4 ratio of long- to short-text tokens. This achieves a trade-off between evaluation cost and result fidelity. The 6:4 ratio is a common choice, but for special cases, we recommend defining scenario-specific ratios for evaluation.

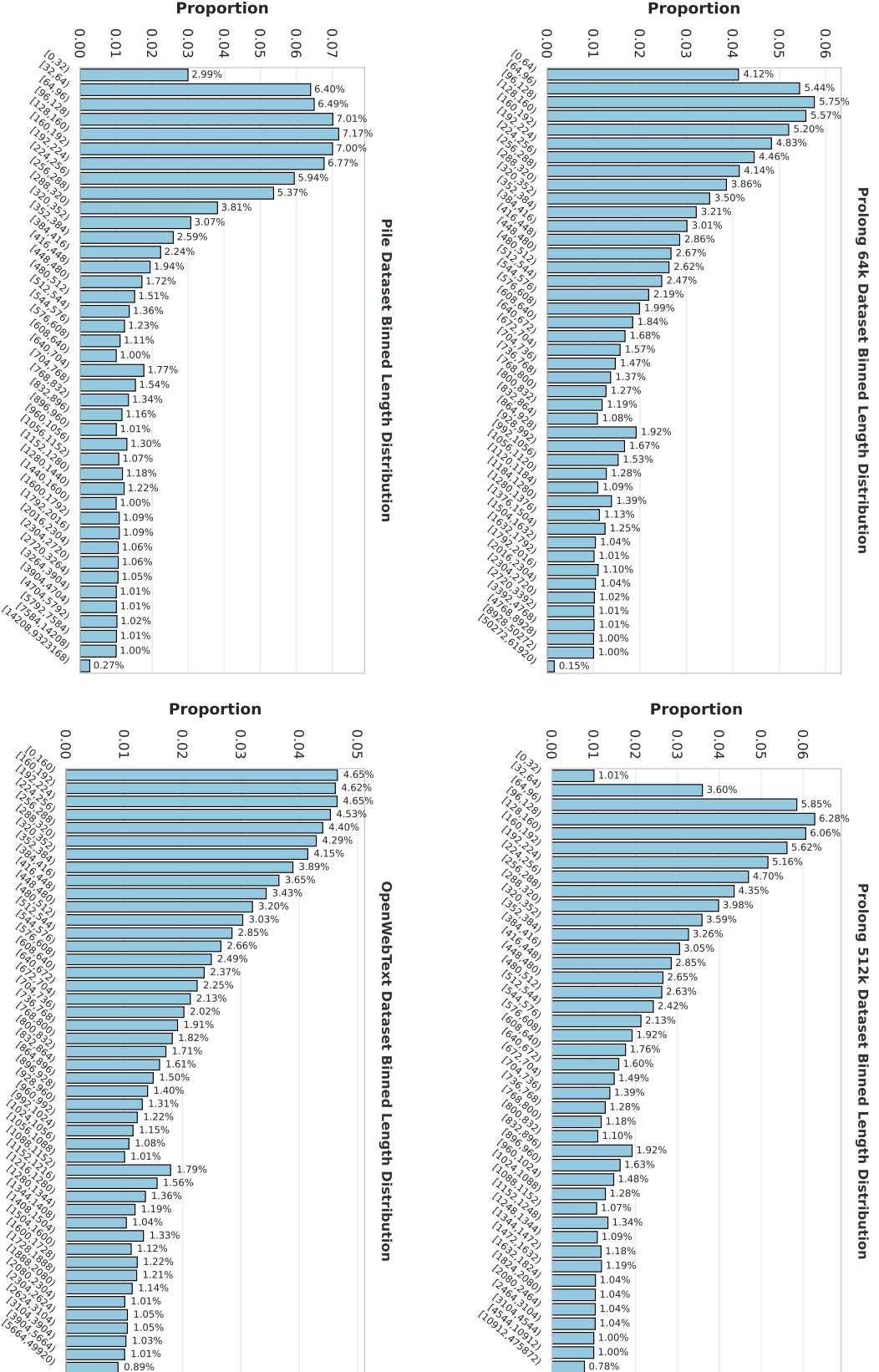

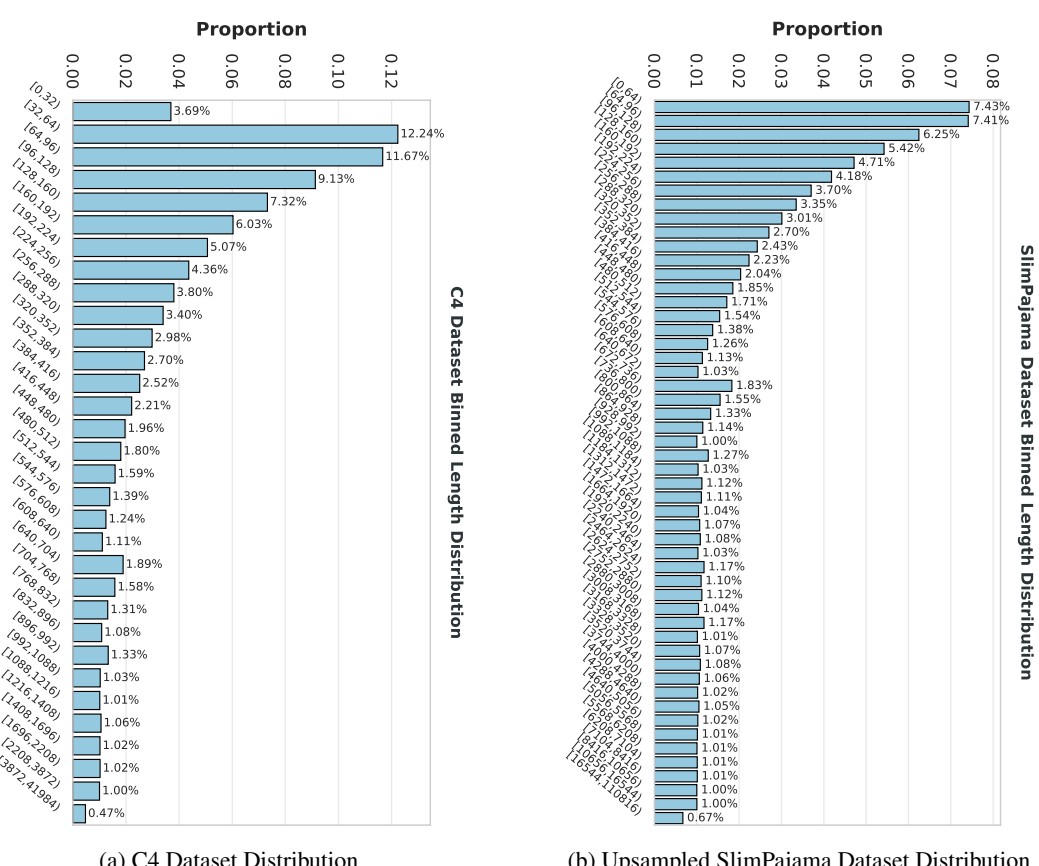

(a) C4 Dataset Distribution      (b) Upsampled SlimPajama Dataset Distribution

Figure 7: Pretraining Dataset Length Distributions.

### A.3 DETAILS OF SPARSE DATA SAMPLING

Referencing common block sparse mask generation methods, our block sparse mask generation process is as follows:

1. Block Partitioning: For a task with an input sequence length of $seqlen$, we can conceptualize a $seqlen \times seqlen$ attention map. Based on the provided $q\_block\_lists$ and $k\_block\_lists$, this two-dimensional attention map is partitioned into $len(q\_block\_lists) \times len(k\_block\_lists)$ blocks. For a uniform block mask, the block sizes within $q\_block\_lists$ and $k\_block\_lists$ are the same. In contrast, for a variable block mask, these block sizes can differ.

2. Score Calculation: Common block sparse methods typically calculate a score for each block in some manner (e.g., by applying mean pooling over each block) and generate a score matrix which represents the importance of the block. Blocks with higher importance are more likely to be selected. It is important to note that scores may vary across different attention heads. Generally, a distinct mask is generated for each KV head. This means for Multi-Head Attention, the score matrix can be unique for each head. For Grouped-Query Attention, masks are the same within a group but can differ between groups. In our experiments, we abstract away the specifics of score calculation and use randomly generated numbers, thereby focusing on the final block sparse attention computation.

3. Top-k Selection: For each block in the query dimension ($q$), we select the top-$k$ blocks from the key dimension ($k$) for computation. The overall degree of sparsity can be expressed as the fraction $\frac{k}{len(k\_block\_lists)}$. We define a $sparsity\_ratio$ to represent this degree of sparsity, which has a direct conversion relationship with top-$k$ and is essentially equivalent. To observe the kernel's performance under different sparsity levels, we select $0.2$, $0.5$, and $0.8$ as representative sparsity ratios.

### A.4 DETAILS OF SPARSE ATTENTION KERNELS

VSA (Zhang et al., 2025d) is a trainable block sparse attention implementation designed specifically for video diffusion models. It employs a two-stage methodology. In the coarse-grained stage, it applies a token rearrangement strategy from STA (Zhang et al., 2025c) to increase computational density. It then calculates inter-block scores via cube partition and mean-pooling on the QK matrix, which selects the top-k K blocks for each Q block. Subsequently, in the fine-grained stage, it utilizes ThunderKittens (Spector et al., 2024) to develop a high-performance block sparse attention kernel. This kernel is customized for Hopper GPUs to maximize hardware utilization. According to VSA's analysis, the fine-grained stage accounts for over 80% of latency at context lengths of 32K or more. This finding highlights the critical need to optimize the block sparse attention kernel, and our performance benchmarks also focus on this kernel for fair comparisons. As a variant, Triton VSA implements the algorithm in the Triton language. This approach aims to enhance cross-hardware compatibility but results in some performance degradation. However, both implementations are specifically optimized for video models and they do not support common LLM attention modes like Grouped-Query Attention (GQA). FA2 Sparse Guo et al. (2024) is an open-source implementation based on the FlashAttention-2 (Dao, 2023) codebase. It enables block sparse functionality by modifying the computation logic, allowing each Q block to traverse only its designated KV blocks. The primary limitation of this implementation is its lack of support for the backward pass and without optimization for advanced GPUs like NVIDIA Hopper GPUs. FlexAttention Dong et al. (2024) leverages compiler technology to introduce a more flexible mask description method. This approach enables it to support sparse attention in the form of a block mask. However, the representation for block masks is relatively complex. Its compilation technique can therefore degenerate to instantiating a full $O(S^2)$ mask, which causes significant memory overhead. FlashInfer Ye et al. (2025) is a general-purpose kernel library oriented toward LLM inference. It designs a block sparse matrix structure as a unified format for the KV cache. This design allows block sparse attention input to be converted into a paged attention format where page size equals to 1. This process enables the reuse of its efficient attention kernel and supports arbitrary block sizes. Due to its positioning as an inference library, it does not support the backward pass.

## A.5 DETAILS OF DENSE KERNEL PERFORMANCE ON H100 GPU

For single-sequence samples such as FULL/CAUSAL and SLIDING WINDOW, we conduct 2 runs of sampling, with each run followed by 5 warm-up steps and 20 kernel computation steps. For multi-sequence data such as DOCUMENT and SHARE QUESTION, considering the sparsity differences introduced by sampling, we perform 30 runs with independent sampling in each run, also followed by 5 warm-up steps and 20 kernel computation steps. We record the median values of FLOPS and peak memory. It is worth noting that our results represent the expected outcomes under these specific settings, and occasional large deviations in individual kernel runs are considered normal.

### A.5.1 PERFORMANCE METRIC: FLOPs

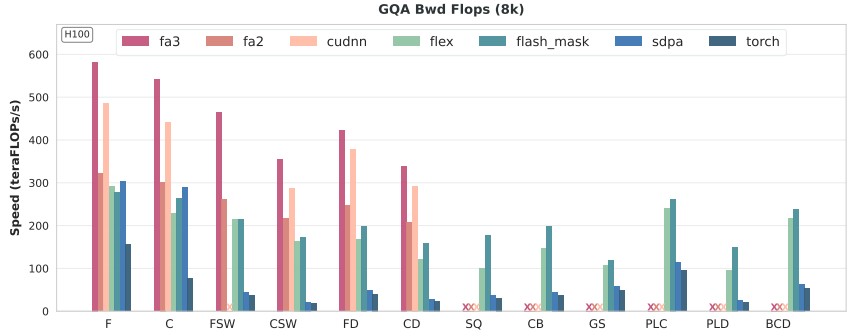

Figure 8: Backward TFLOPs of dense kernels with different masks (8K length)

FULL and CAUSAL are the most common masks used in language model pretraining, as shown in Figure 9. SDPA and Flex approximately represent the baselines of fused operators without hardware-specific optimization. In general, increasing sequence length improves kernel performance, which typically stabilizes around 16K. The results highlight the importance of hardware-aware optimization: on the H100 Hopper architecture, FA3 achieve significant performance gains.

FULL/CAUSAL DOCUMENT is primarily designed for concatenating variable-length input sequences to reduce unnecessary padding while preserving full or causal connectivity. It is important to note that concatenating variable-length sequences can introduce computational instability, which becomes particularly pronounced when the data contains many small fragmented chunks, as shown in Figure 10.

In our experiments, we fixed the sliding window size to 1024, as shown in Figure 11, though we recommend evaluating with other window sizes as well (Fu et al., 2025).

Overall, in heterogeneous mask scenarios, Flex and FlashMask show varying performance gains depending on the context.

PREFIX LM and PREFIX LM DOCUMENT extend the standard language model regular mask by introducing a prefix, allowing the prefix to attend to all tokens. In our experiments, a prefix is randomly generated for each run, and the median across multiple runs is reported to reflect expected performance in realistic scenarios with varying prefixes, as shown in Figure 12. Models trained with a prefix demonstrate advantages in handling long-text and multi-turn dialogue tasks. This approach enables the model to better leverage contextual information, improving performance in generation tasks (Raffel et al., 2020).

SHARE QUESTION and CAUSAL BLOCKWISE can be viewed as variants of DOCUMENT, as shown in Figure 13. SHARE QUESTION allows all query tokens to share the key tokens from the first document and is commonly used in Reward Models (RM) as a shared-question mask, enabling multiple answers to reference the same question (Ouyang et al., 2022). This eliminates redundant computation and accelerates training. CAUSAL BLOCKWISE, on the other hand, allows all key tokens to share the query tokens from the last document and is typically applied in demonstration–test tasks, where test examples can attend to all demonstrations. This facilitates studying model performance improvements in long-context tasks (Bertsch et al., 2024).

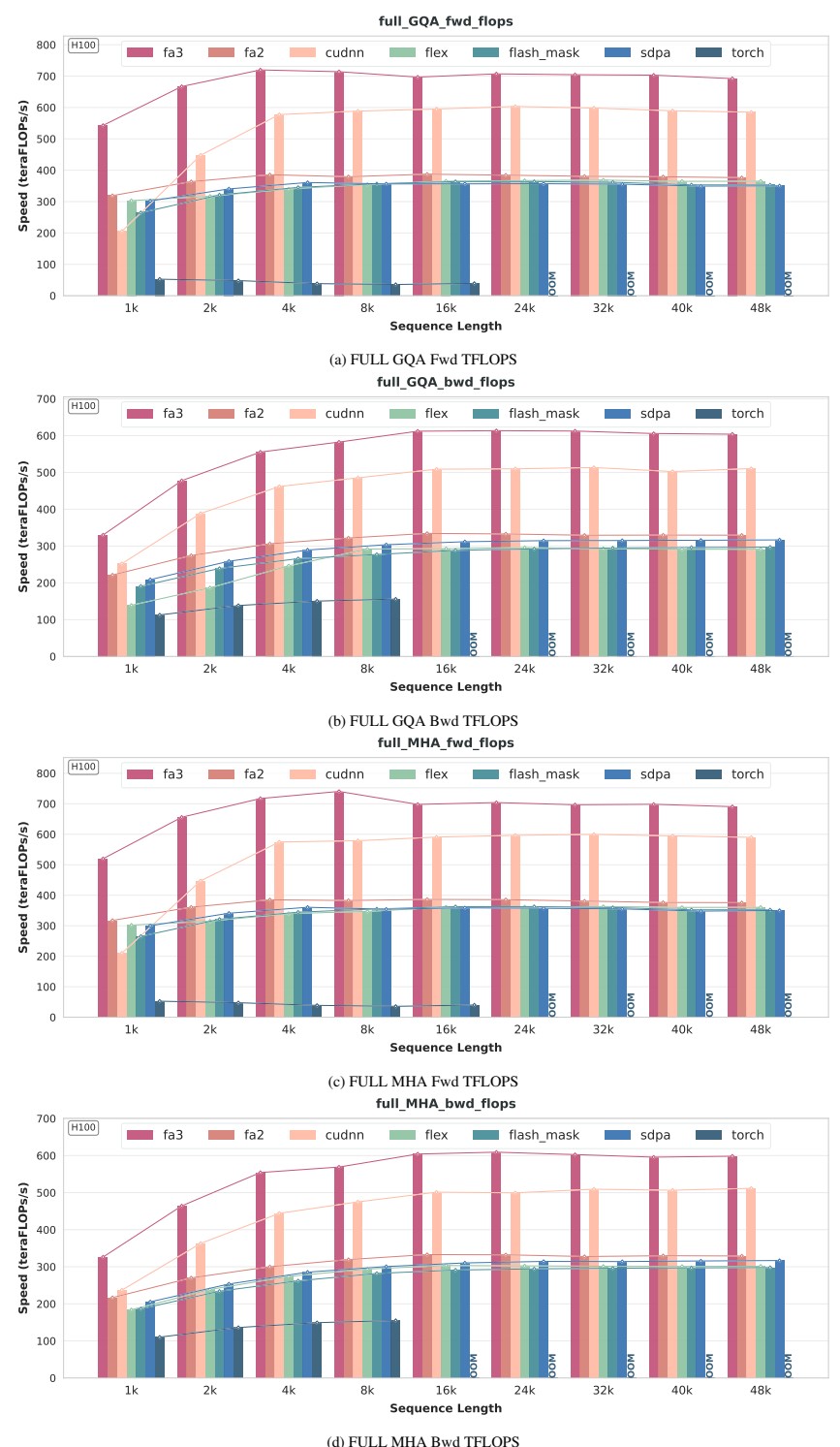

(a) FULL GQA Fwd TFLOPS

(b) FULL GQA Bwd TFLOPS

(c) FULL MHA Fwd TFLOPS

(d) FULL MHA Bwd TFLOPS

GLOBAL SLIDING combines global attention with sliding-window attention. In each run, we randomly sample a window size, treating the leftmost window_size tokens in the Query and Key as global tokens, which attend to all Key and Query tokens, respectively. Due to the increased sparsity of the mask, the performance of Flex and FlashMask correspondingly decreases, as shown in Figure 14.

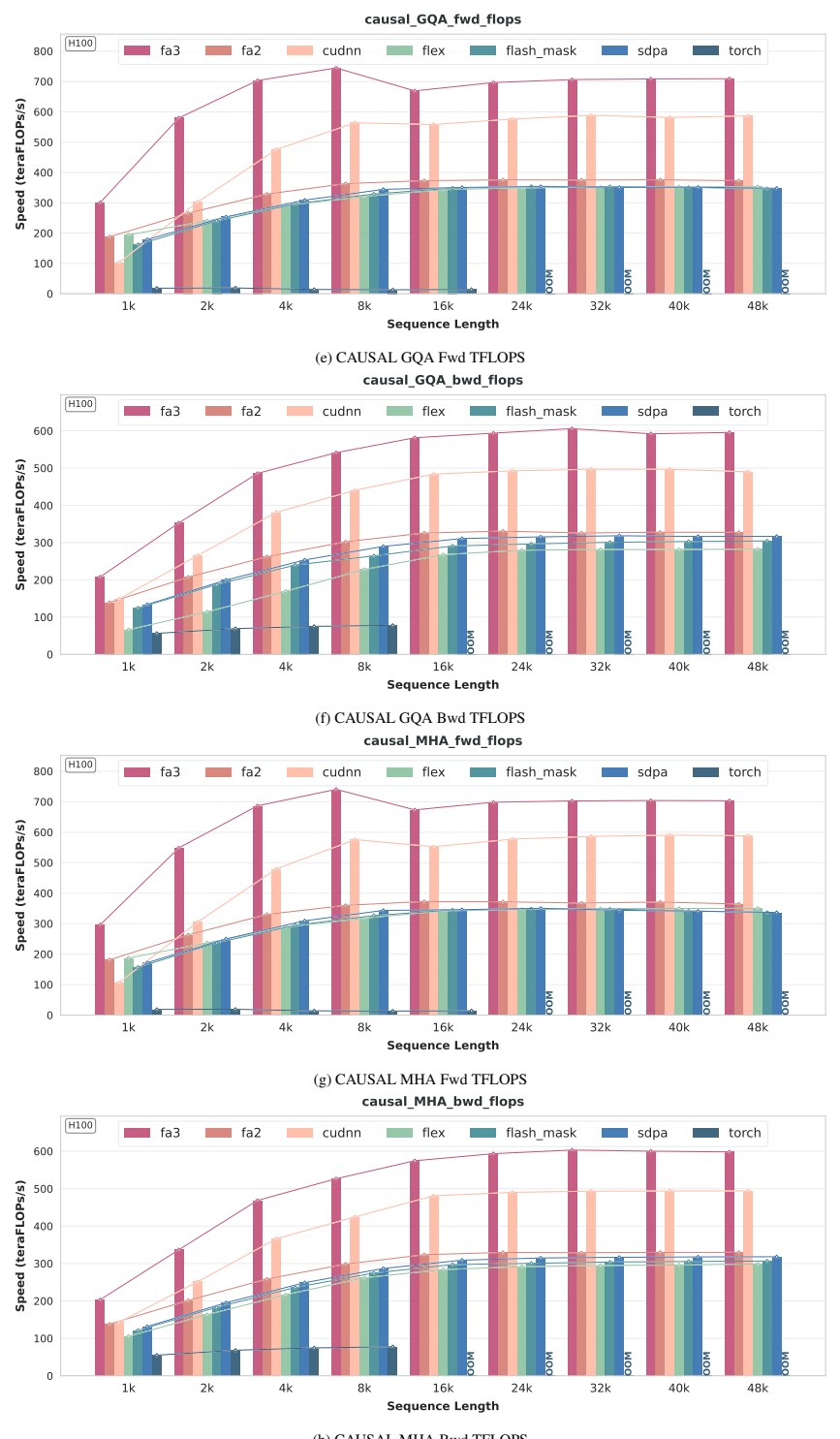

(e) CAUSAL GQA Fwd TFLOPS

(f) CAUSAL GQA Bwd TFLOPS

(g) CAUSAL MHA Fwd TFLOPS

(h) CAUSAL MHA Bwd TFLOPS

Figure 9: TFLOPS of FULL and CAUSAL

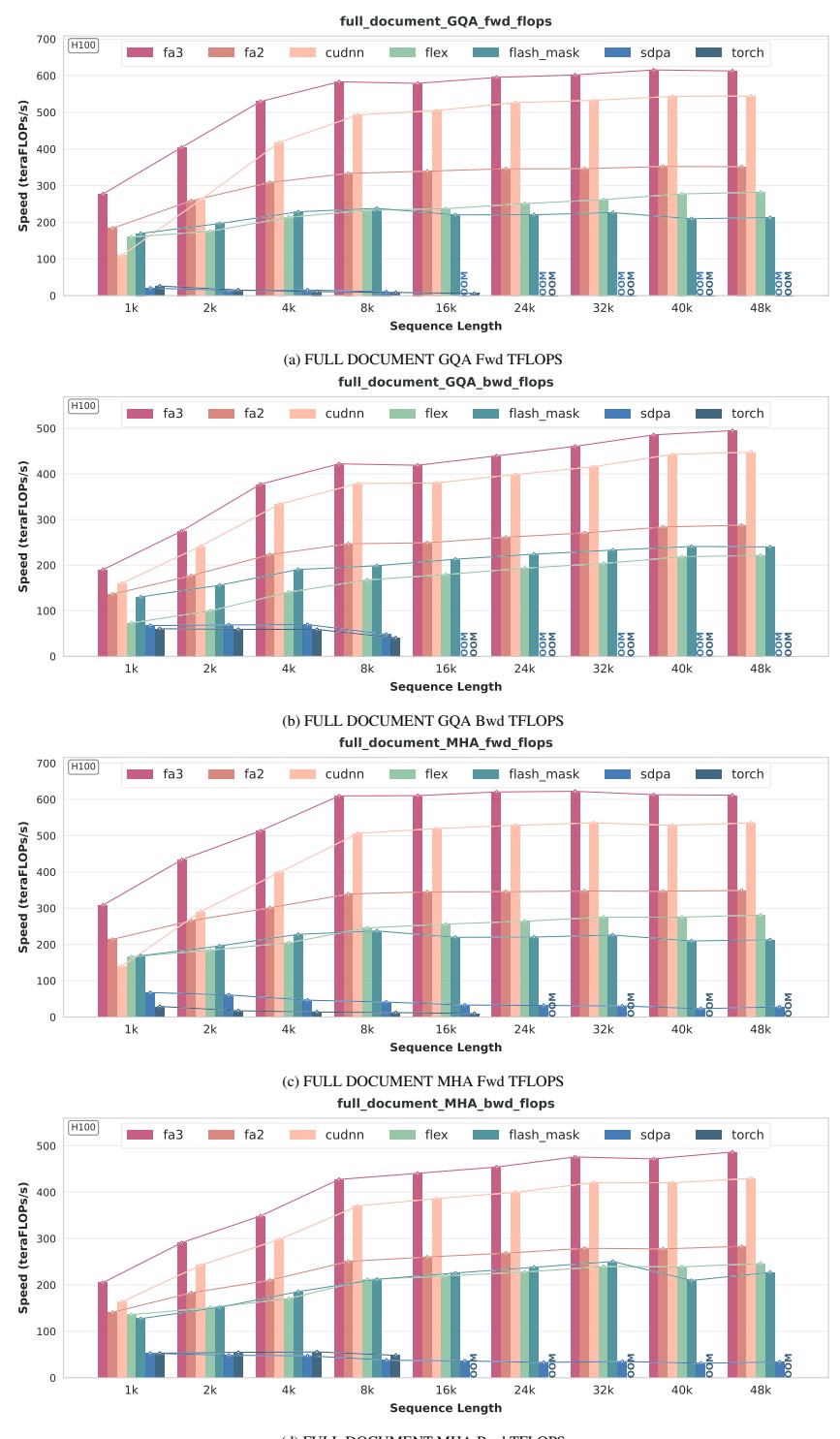

(a) FULL DOCUMENT GQA Fwd TFLOPS

(b) FULL DOCUMENT GQA Bwd TFLOPS

(c) FULL DOCUMENT MHA Fwd TFLOPS

(d) FULL DOCUMENT MHA Bwd TFLOPS

BLOCK CAUSAL DOCUMENT can also be viewed as a variant of DOCUMENT, elevating computation from the token level to the block level. In our experiments, we fix block_size = 1024, as shown in Figure 15. This approach is commonly used in training autoregressive multimodal large models (ai et al., 2025).

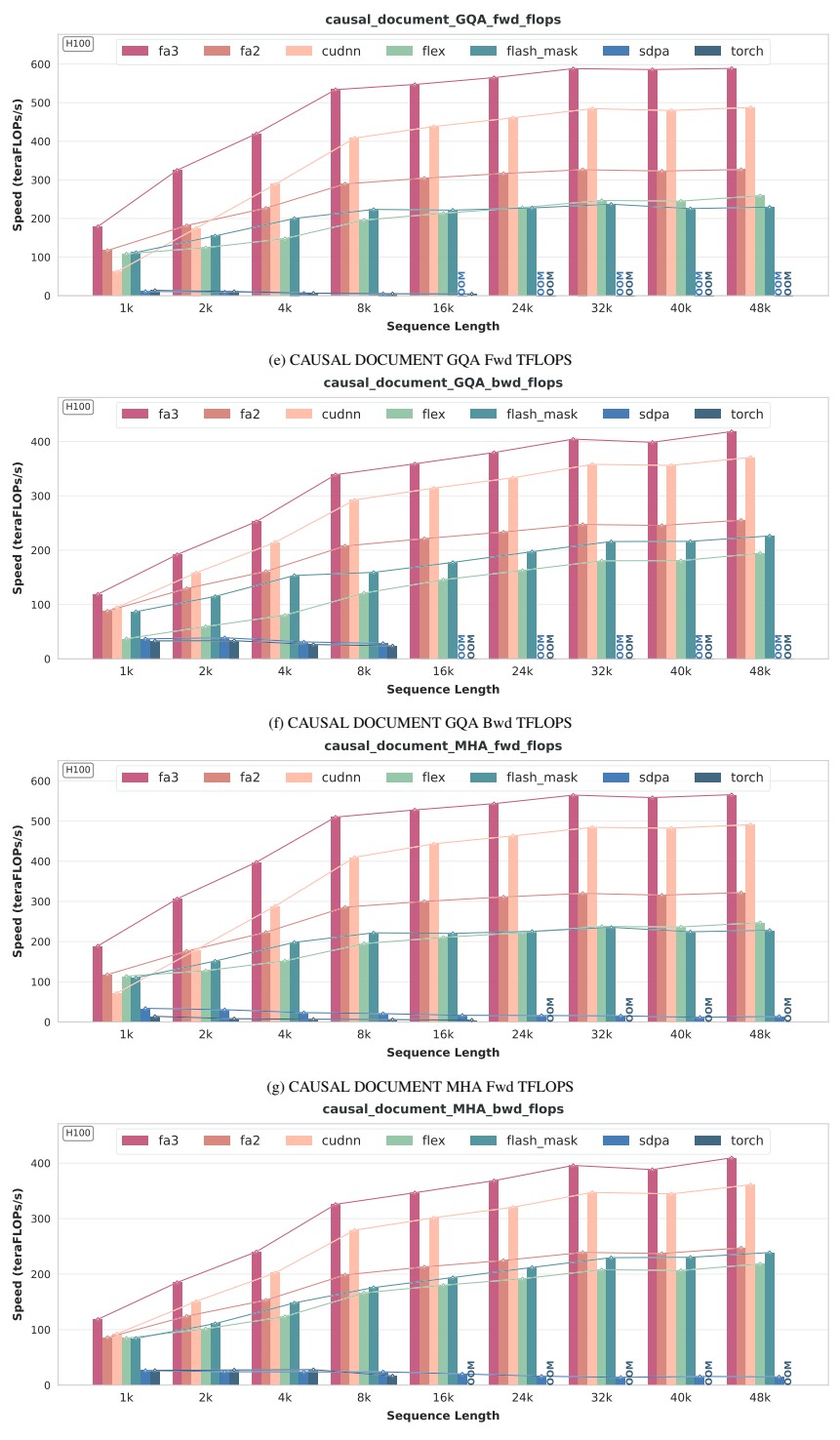

(e) CAUSAL DOCUMENT GQA Fwd TFLOPS

(f) CAUSAL DOCUMENT GQA Bwd TFLOPS

(g) CAUSAL DOCUMENT MHA Fwd TFLOPS

(h) CAUSAL DOCUMENT MHA Bwd TFLOPS

Figure 10: TFLOPS of FULL/CAUSAL DOCUMENT

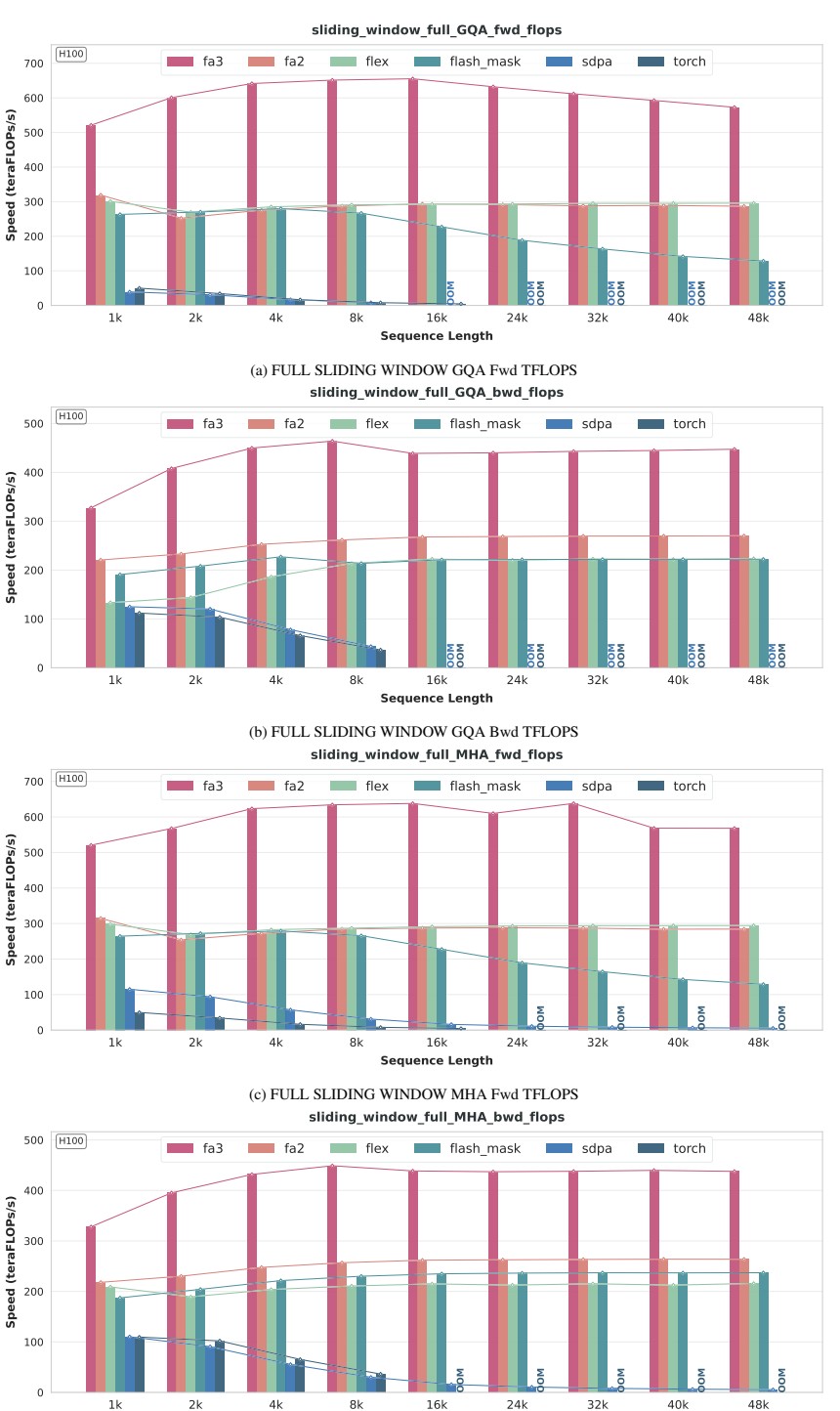

(a) FULL SLIDING WINDOW GQA Fwd TFLOPS

(b) FULL SLIDING WINDOW GQA Bwd TFLOPS

(c) FULL SLIDING WINDOW MHA Fwd TFLOPS

(d) FULL SLIDING WINDOW MHA Bwd TFLOPS

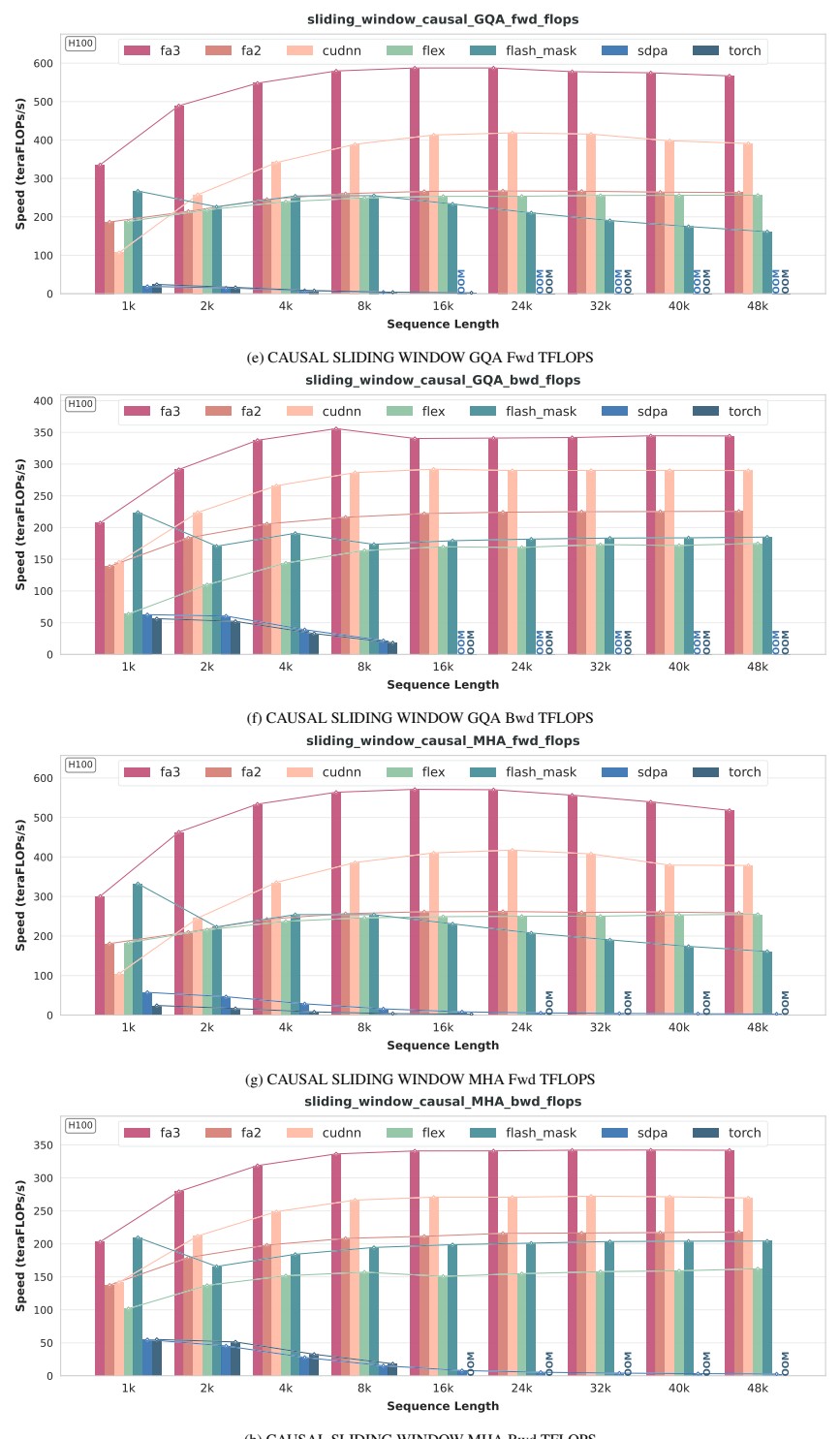

(e) CAUSAL SLIDING WINDOW GQA Fwd TFLOPS

(f) CAUSAL SLIDING WINDOW GQA Bwd TFLOPS

(g) CAUSAL SLIDING WINDOW MHA Fwd TFLOPS

(h) CAUSAL SLIDING WINDOW MHA Bwd TFLOPS

Figure 11: TFLOPS of FULL/CAUSAL SLIDING WINDOW

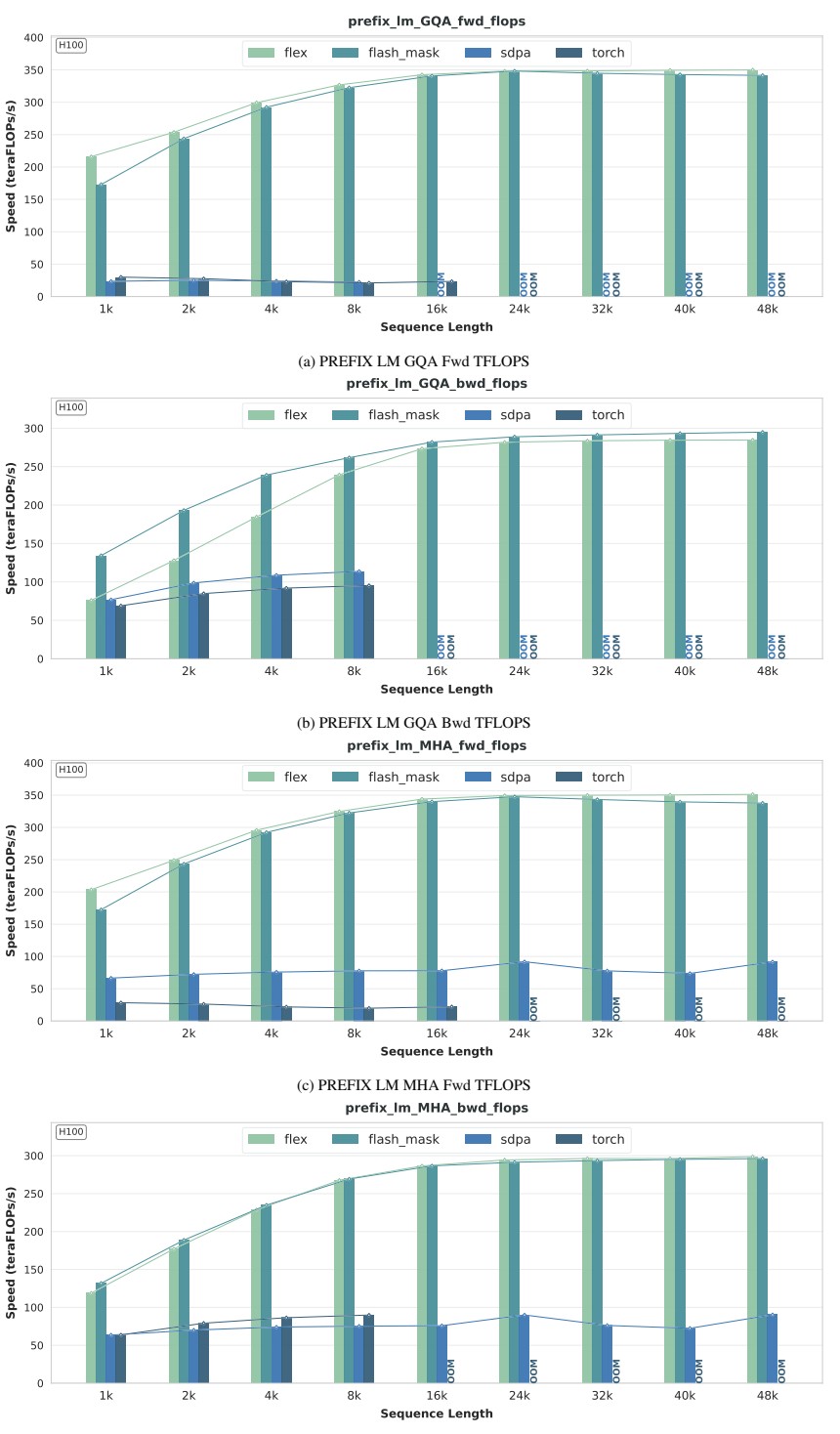

(a) PREFIX LM GQA Fwd TFLOPS

(b) PREFIX LM GQA Bwd TFLOPS

(c) PREFIX LM MHA Fwd TFLOPS

(d) PREFIX LM MHA Bwd TFLOPS

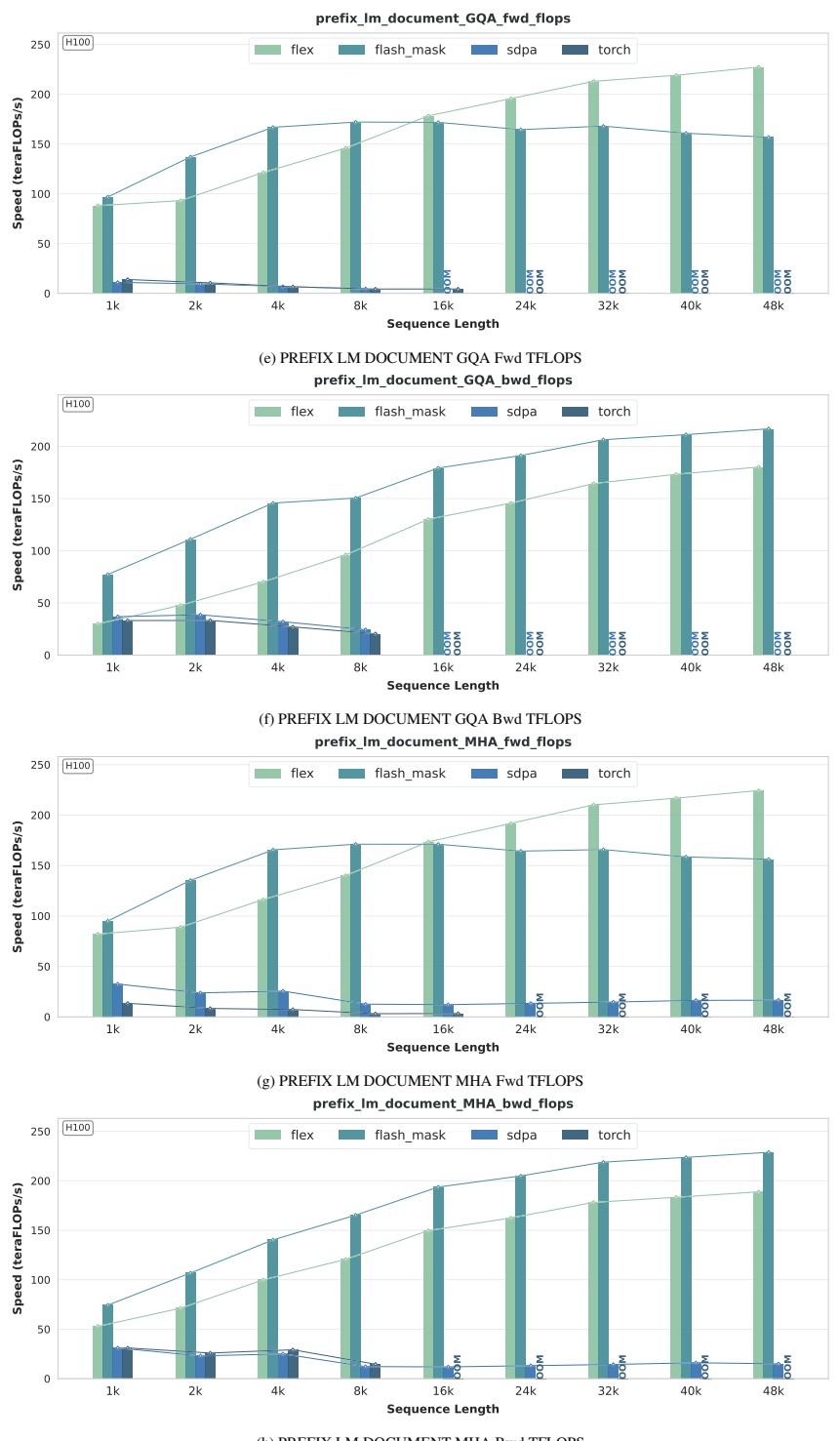

(e) PREFIX LM DOCUMENT GQA Fwd TFLOPS

(f) PREFIX LM DOCUMENT GQA Bwd TFLOPS

(g) PREFIX LM DOCUMENT MHA Fwd TFLOPS

(h) PREFIX LM DOCUMENT MHA Bwd TFLOPS

Figure 12: TFLOPS of PREFIX LM and PREFIX LM DOCUMENT

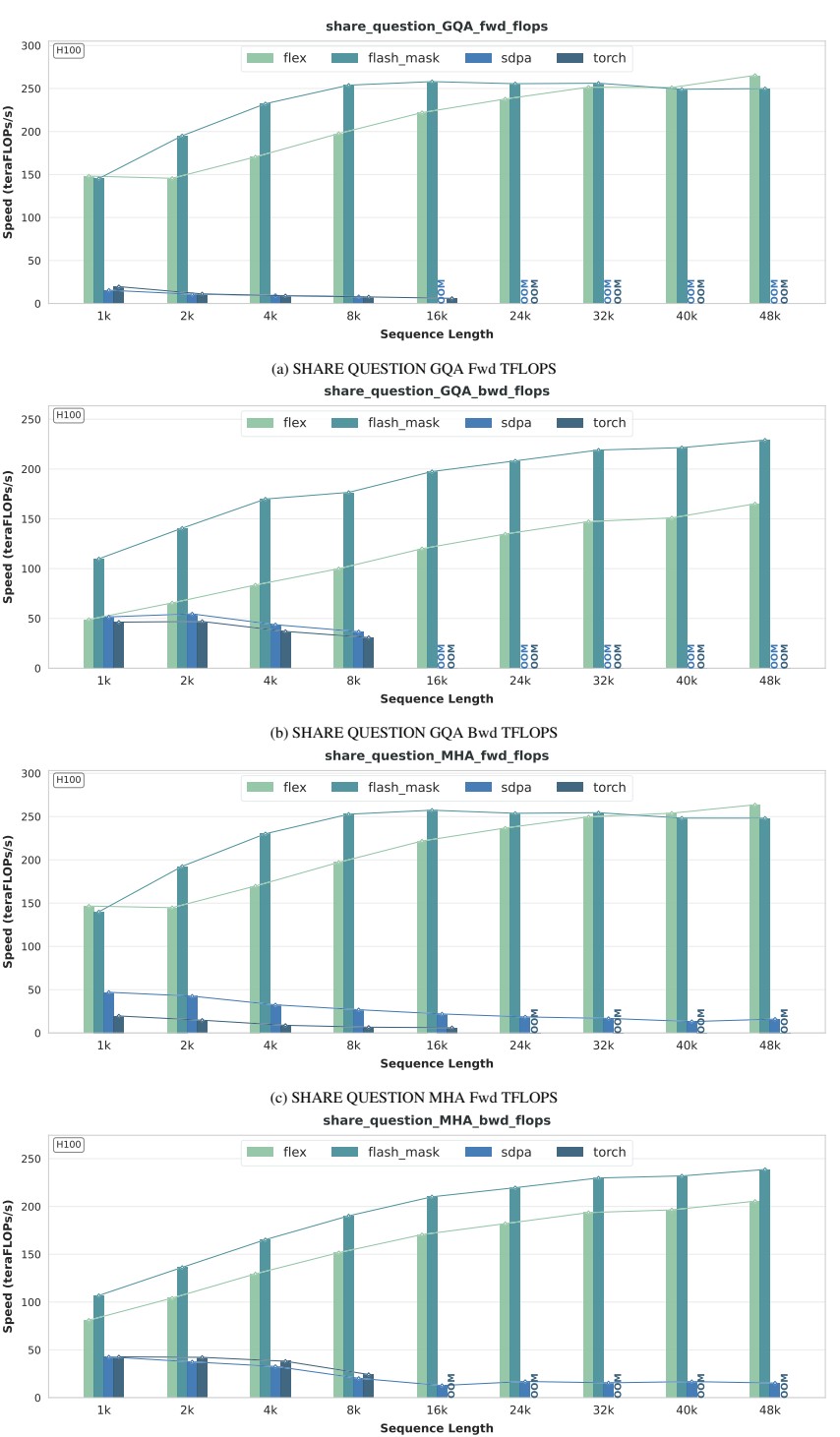

(a) SHARE QUESTION GQA Fwd TFLOPS

(b) SHARE QUESTION GQA Bwd TFLOPS

(c) SHARE QUESTION MHA Fwd TFLOPS

(d) SHARE QUESTION MHA Bwd TFLOPS

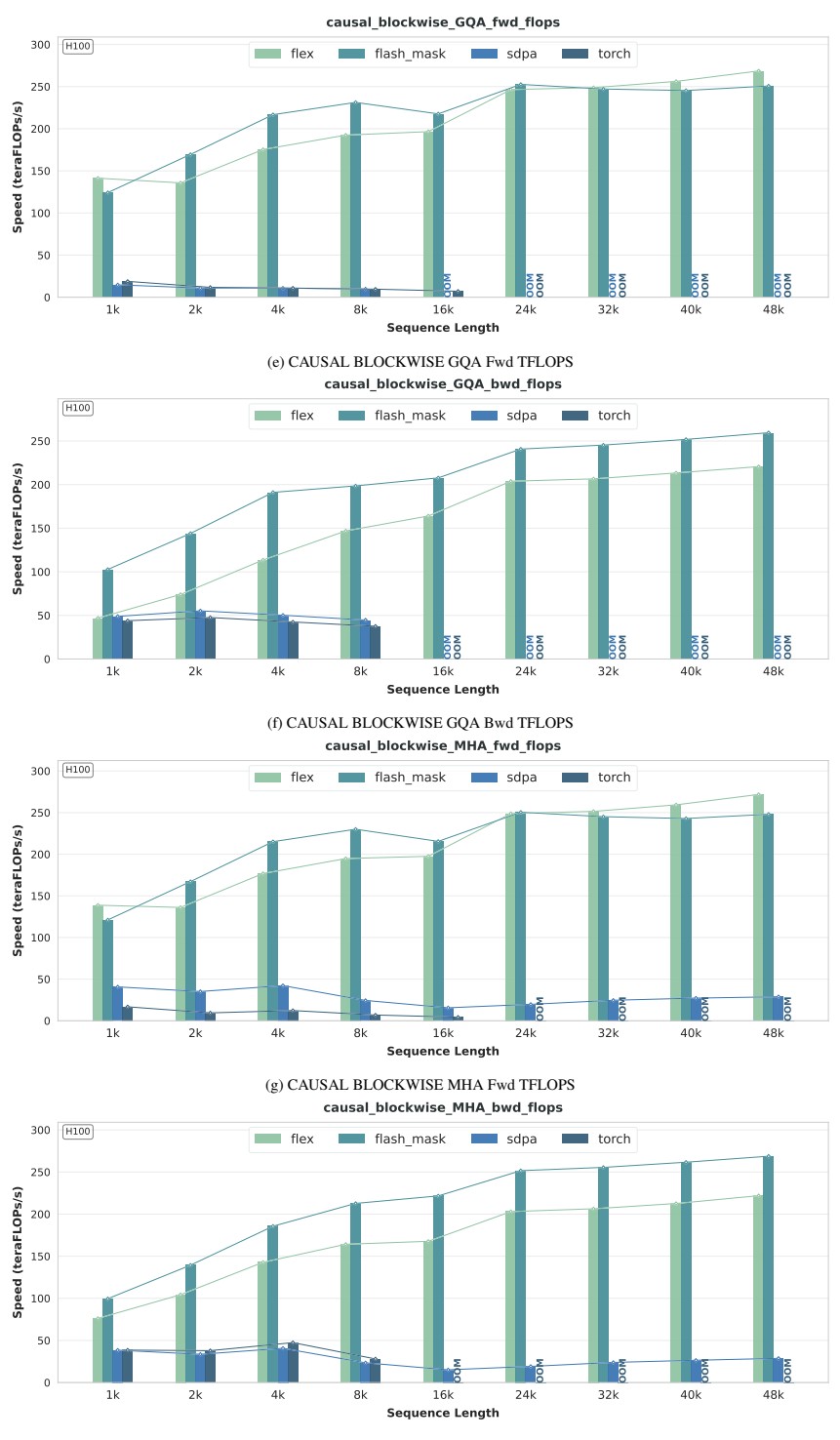

(e) CAUSAL BLOCKWISE GQA Fwd TFLOPS

(f) CAUSAL BLOCKWISE GQA Bwd TFLOPS

(g) CAUSAL BLOCKWISE MHA Fwd TFLOPS

(h) CAUSAL BLOCKWISE MHA Bwd TFLOPS

Figure 13: TFLOPS of SHARE QUESTION and CAUSAL BLOCKWISE

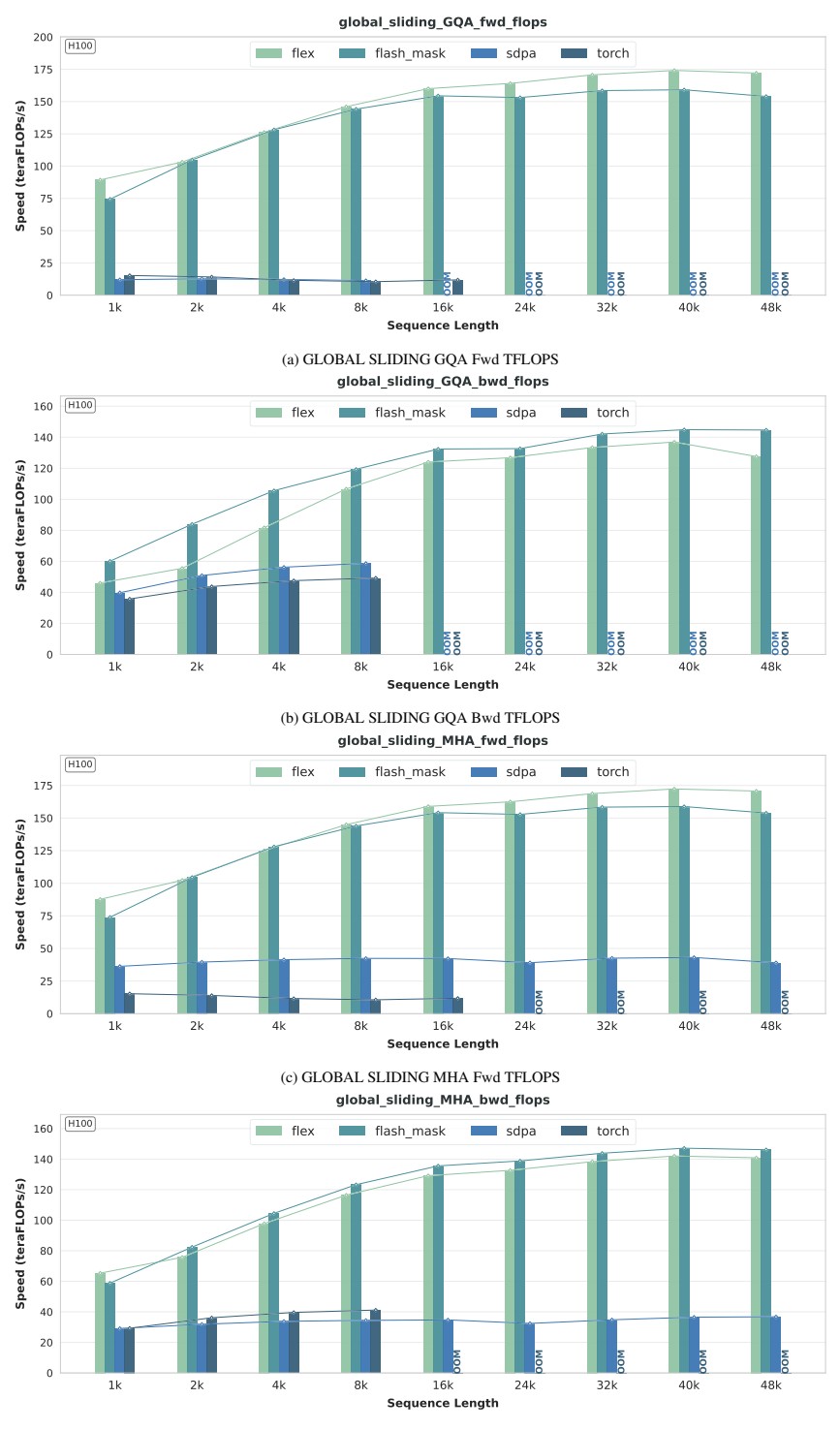

(a) GLOBAL SLIDING GQA Fwd TFLOPS

(b) GLOBAL SLIDING GQA Bwd TFLOPS

(c) GLOBAL SLIDING MHA Fwd TFLOPS

(d) GLOBAL SLIDING MHA Bwd TFLOPS

Figure 14: TFLOPS of GLOBAL SLIDING

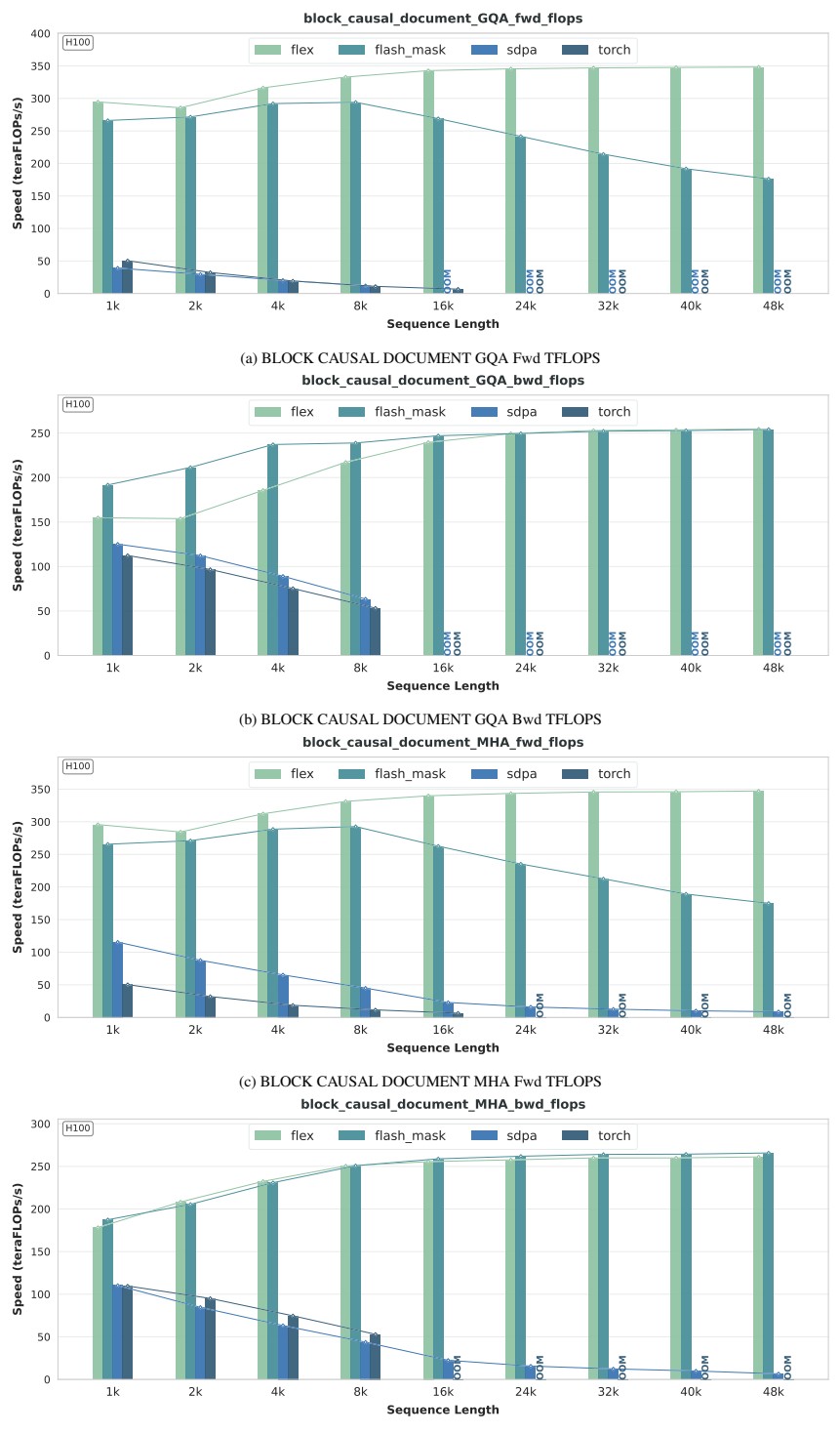

(a) BLOCK CAUSAL DOCUMENT GQA Fwd TFLOPS

(b) BLOCK CAUSAL DOCUMENT GQA Bwd TFLOPS

(c) BLOCK CAUSAL DOCUMENT MHA Fwd TFLOPS

(d) BLOCK CAUSAL DOCUMENT MHA Bwd TFLOPS

Figure 15: TFLOPS of BLOCK CAUSAL DOCUMENT

### A.5.2 PERFORMANCE METRIC: PEAK MEMORY USAGE

We report the peak memory usage of the kernel as a reference. The memory plots (Figure 17, Figure 18) clearly illustrate the detrimental impact of quadratic storage complexity on training: Naive Attention and SDPA scale only up to around 16K in the forward pass and about 8K in the backward pass. We truncate the plots at certain points and annotate the corresponding values. Under the same mask setting, different kernels exhibit similar peak memory in the forward pass, but show noticeable differences in the backward pass. Overall, the trend of peak memory scaling with sequence length across different kernels aligns well with intuition. Since sequence length is extended on a single GPU, model parameters remain fixed, and the growth in peak memory is entirely determined by activations. In the standard attention module, activation memory is computed as $11bshd + 5bhs^2 + 2bshd$, where $b$ is the batch size, $s$ the sequence length, $h$ the number of heads, and $d$ the hidden dimension (see Figure 16). We do not consider any gradient checkpointing (Chen et al., 2016) or offloading (Ren et al., 2021) techniques. Although different kernels may employ various strategies to optimize memory usage, the overall growth trend of activations still approximately follows a quadratic curve, which is confirmed by our experimental results. At the same time, recording peak memory further highlights the performance bottlenecks of the attention mechanism when handling ultra-long contexts. Due to the presence of activations, other distributed strategies such as tensor parallelism and data parallelism are insufficient to alleviate peak memory usage. Only context parallelism, which balances the workload across devices along the sequence dimension, can effectively address this issue.

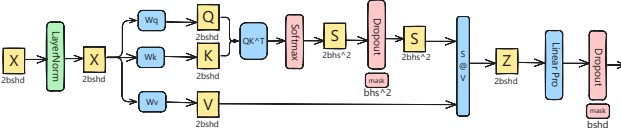

Figure 16: Full Activations in Attention Module

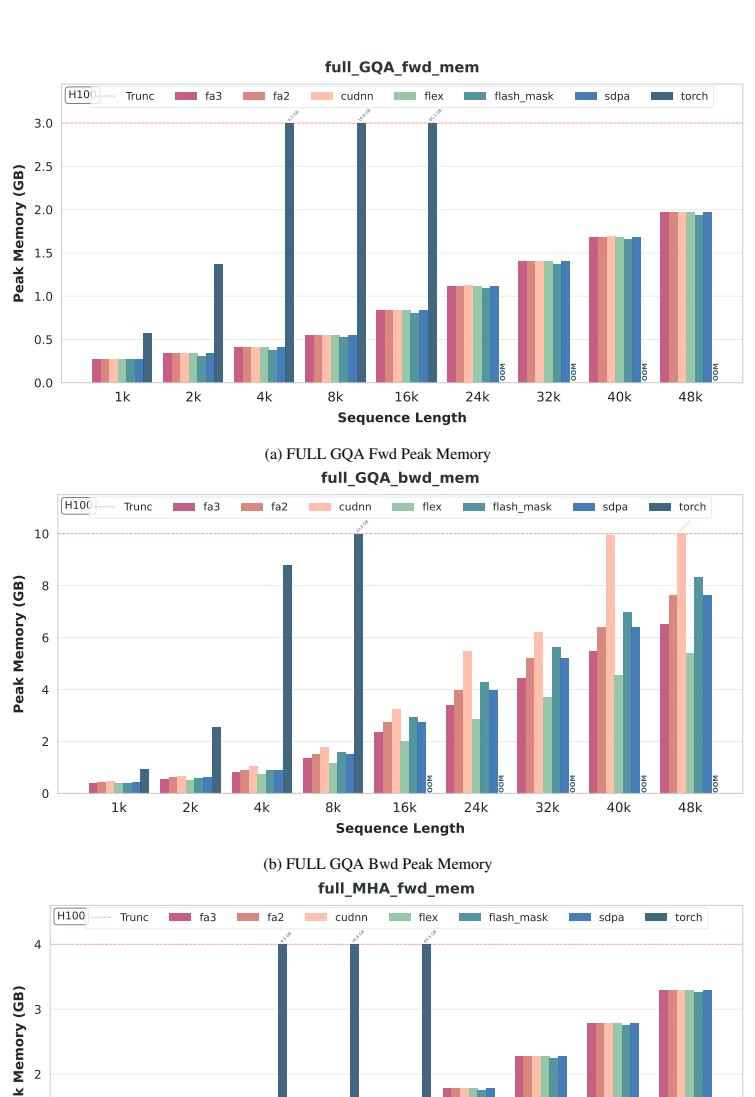

(a) FULL GQA Fwd Peak Memory

(b) FULL GQA Bwd Peak Memory

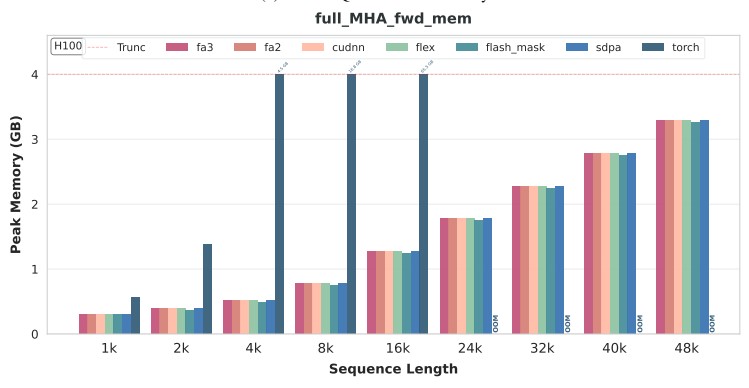

(c) FULL MHA Fwd Peak Memory

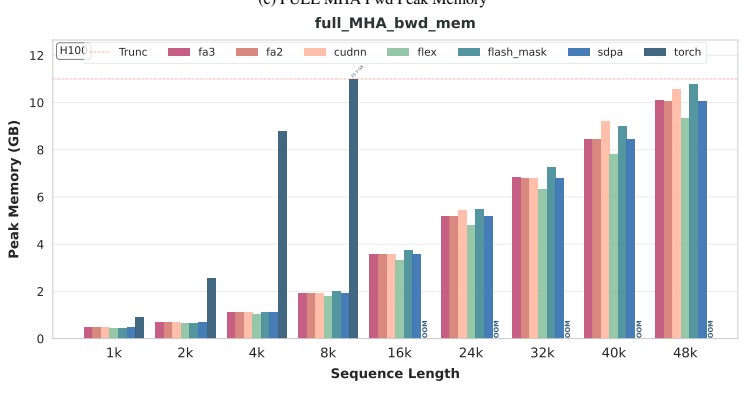

(d) FULL MHA Fwd Peak Memory

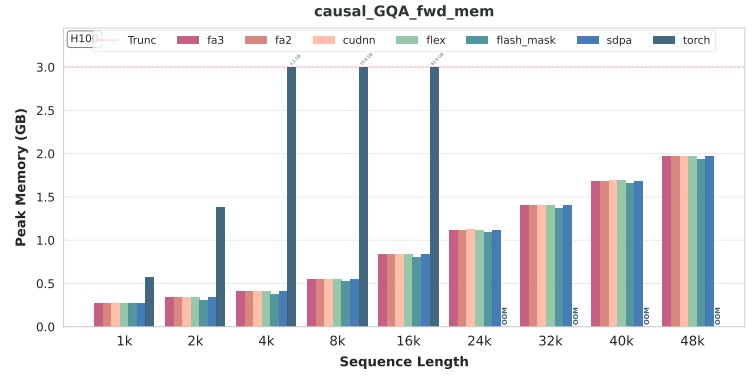

(e) CAUSAL GQA Fwd Peak Memory

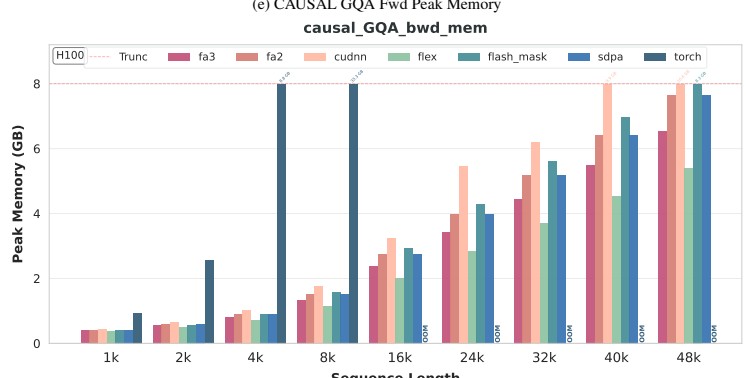

(f) CAUSAL GQA Bwd Peak Memory

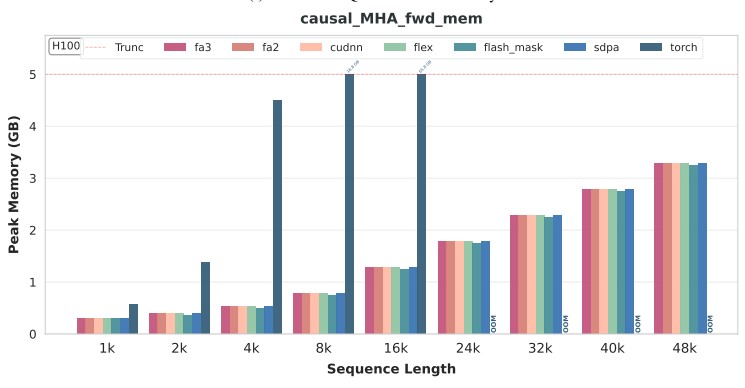

(g) CAUSAL MHA Fwd Peak Memory

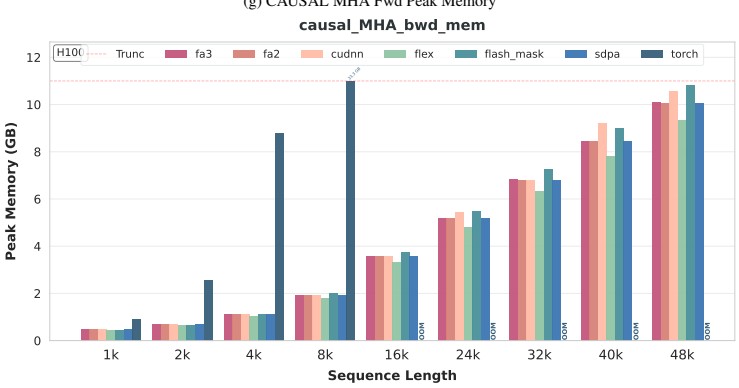

(h) CAUSAL MHA Fwd Peak Memory

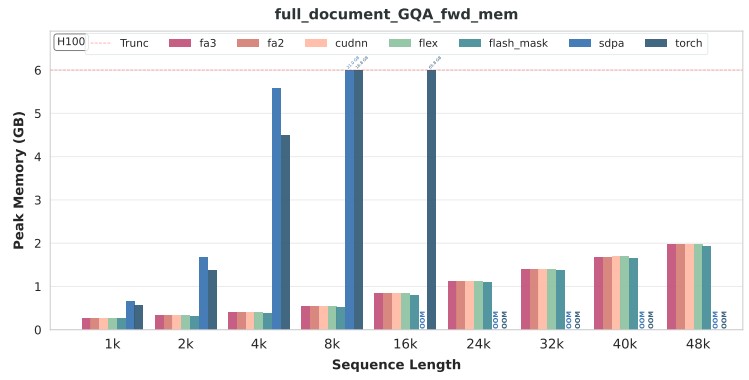

(i) FULL DOCUMENT GQA Fwd Peak Memory

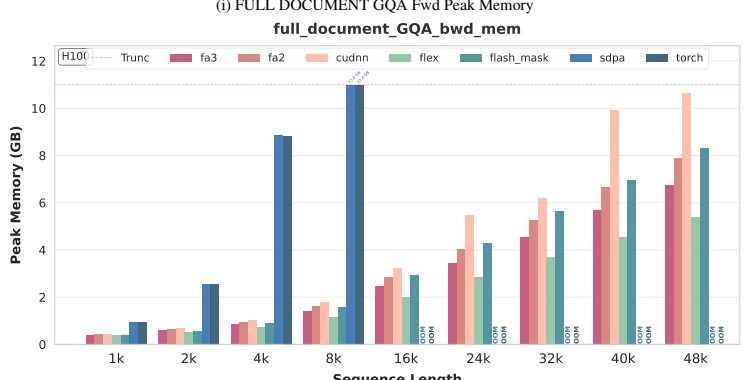

(j) FULL DOCUMENT GQA Bwd Peak Memory

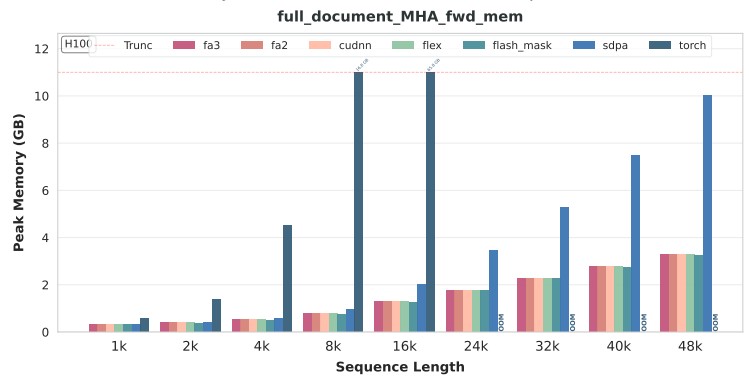

(k) FULL DOCUMENT MHA Fwd Peak Memory

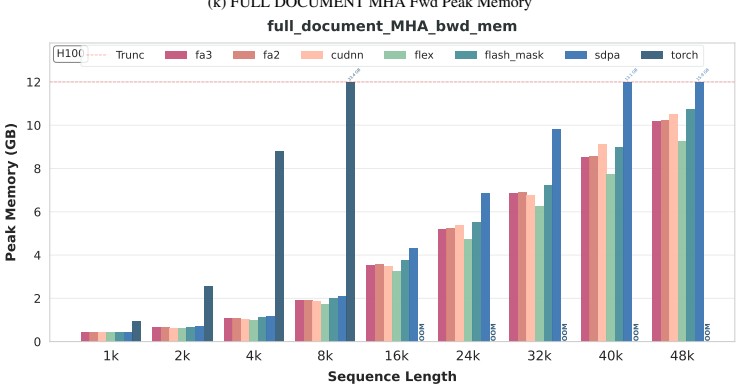

(l) FULL DOCUMENT MHA Fwd Peak Memory

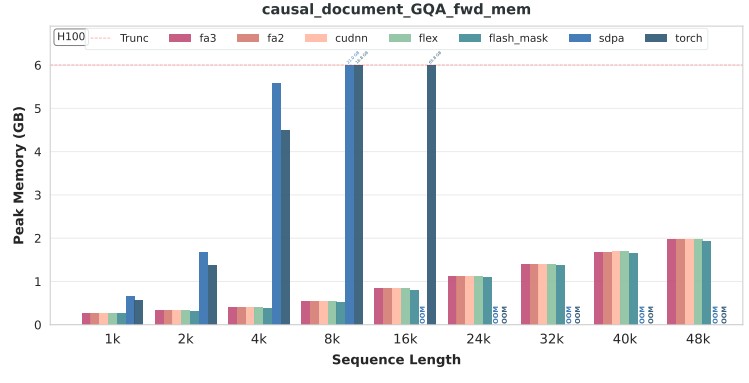

(m) CAUSAL DOCUMENT GQA Fwd Peak Memory

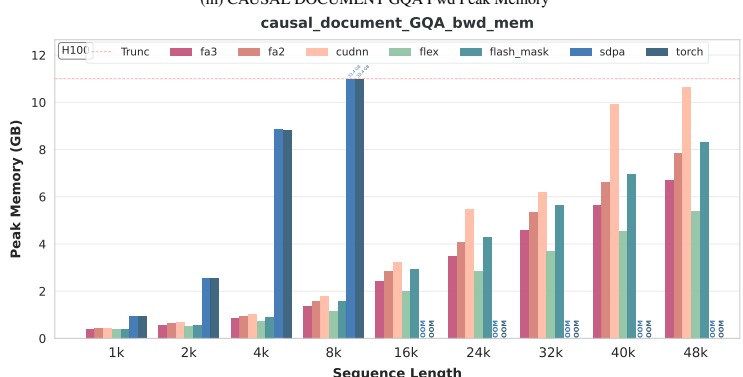

(n) CAUSAL DOCUMENT GQA Bwd Peak Memory

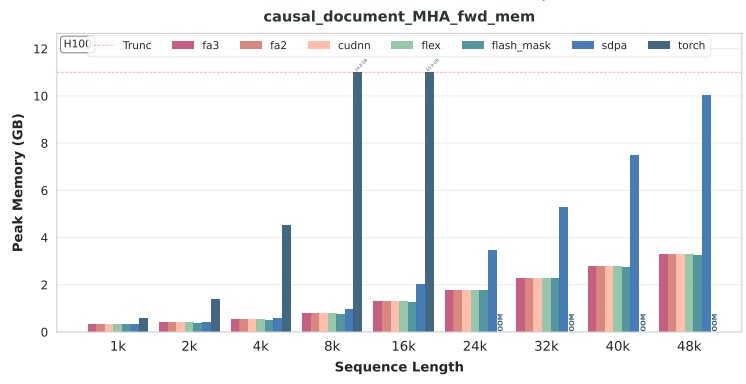

(o) CAUSAL DOCUMENT MHA Fwd Peak Memory

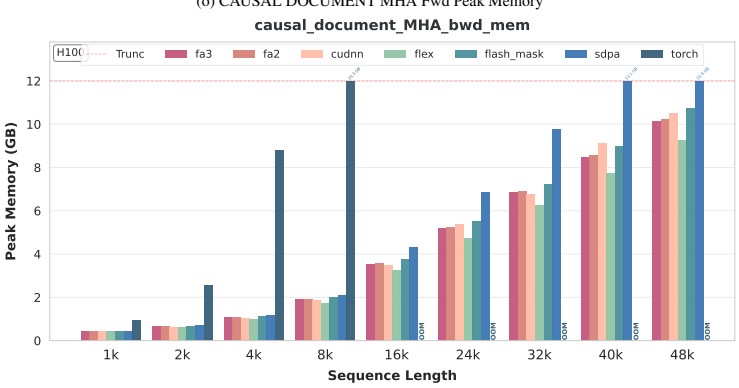

(p) CAUSAL DOCUMENT MHA Fwd Peak Memory

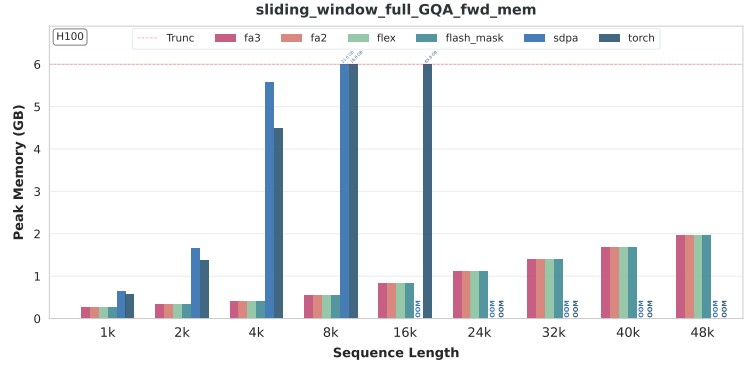

(q) FULL SLIDING WINDOW GQA Fwd Peak Memory

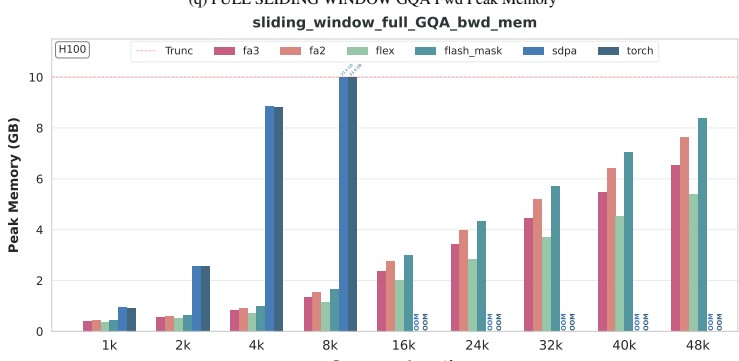

(r) FULL SLIDING WINDOW GQA Bwd Peak Memory

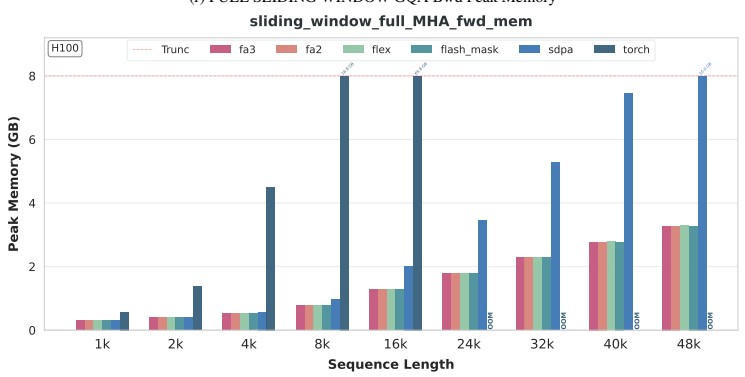

(s) FULL SLIDING WINDOW MHA Fwd Peak Memory

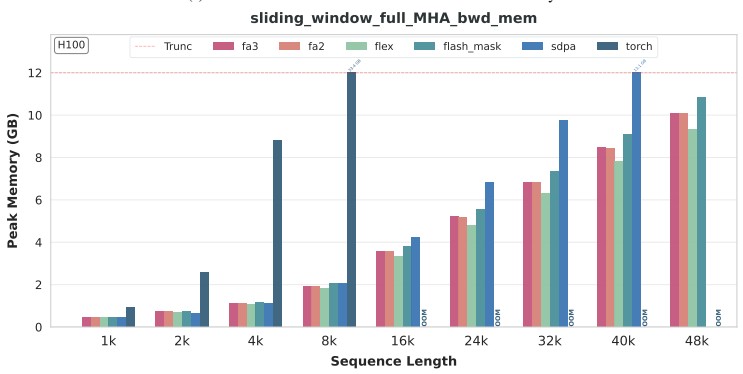

(t) FULL SLIDING WINDOW MHA Bwd Peak Memory

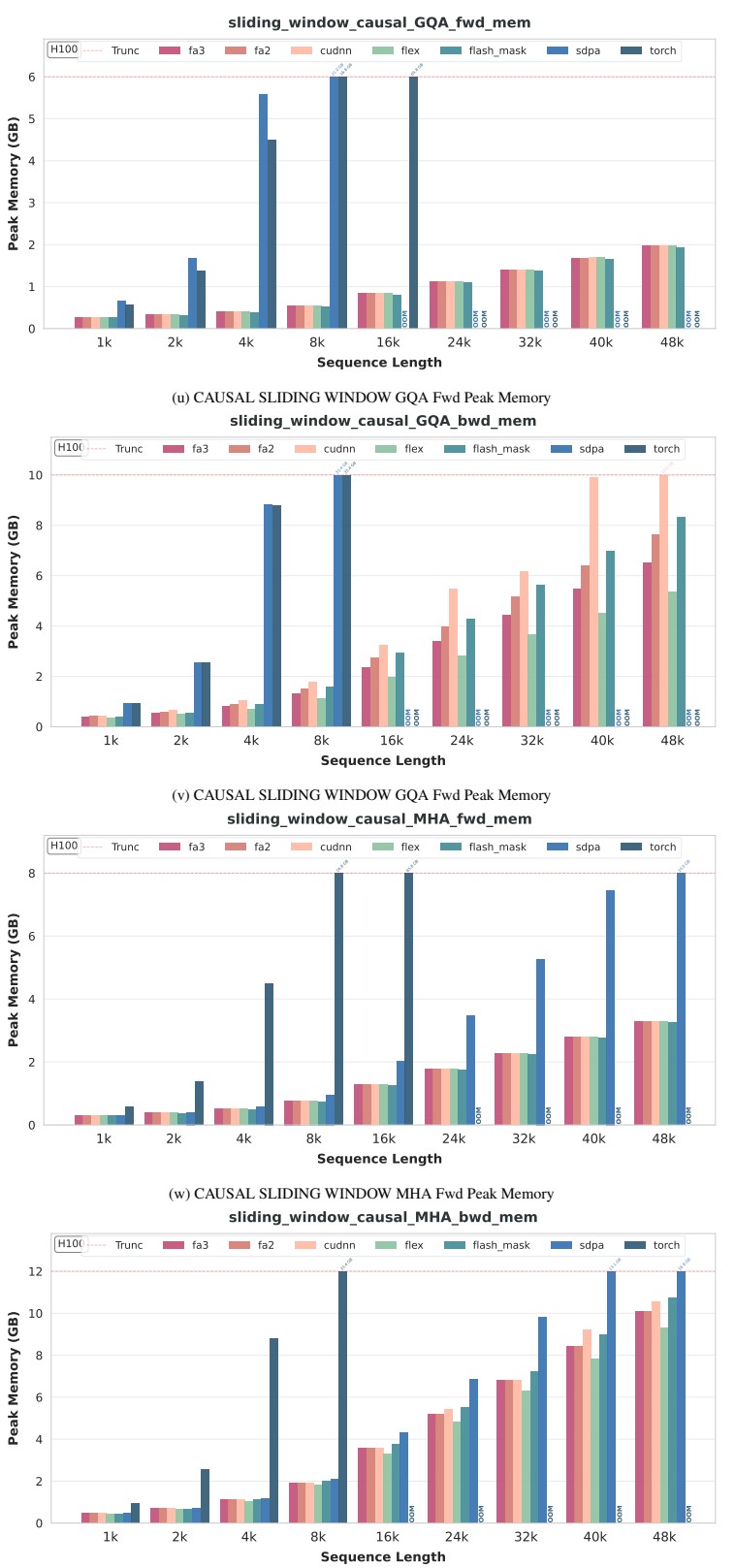

(u) CAUSAL SLIDING WINDOW GQA Fwd Peak Memory

(v) CAUSAL SLIDING WINDOW GQA Fwd Peak Memory

(w) CAUSAL SLIDING WINDOW MHA Fwd Peak Memory

(x) CAUSAL SLIDING WINDOW MHA Bwd Peak Memory

Figure 17: Peak Memory of Static Regular Masks

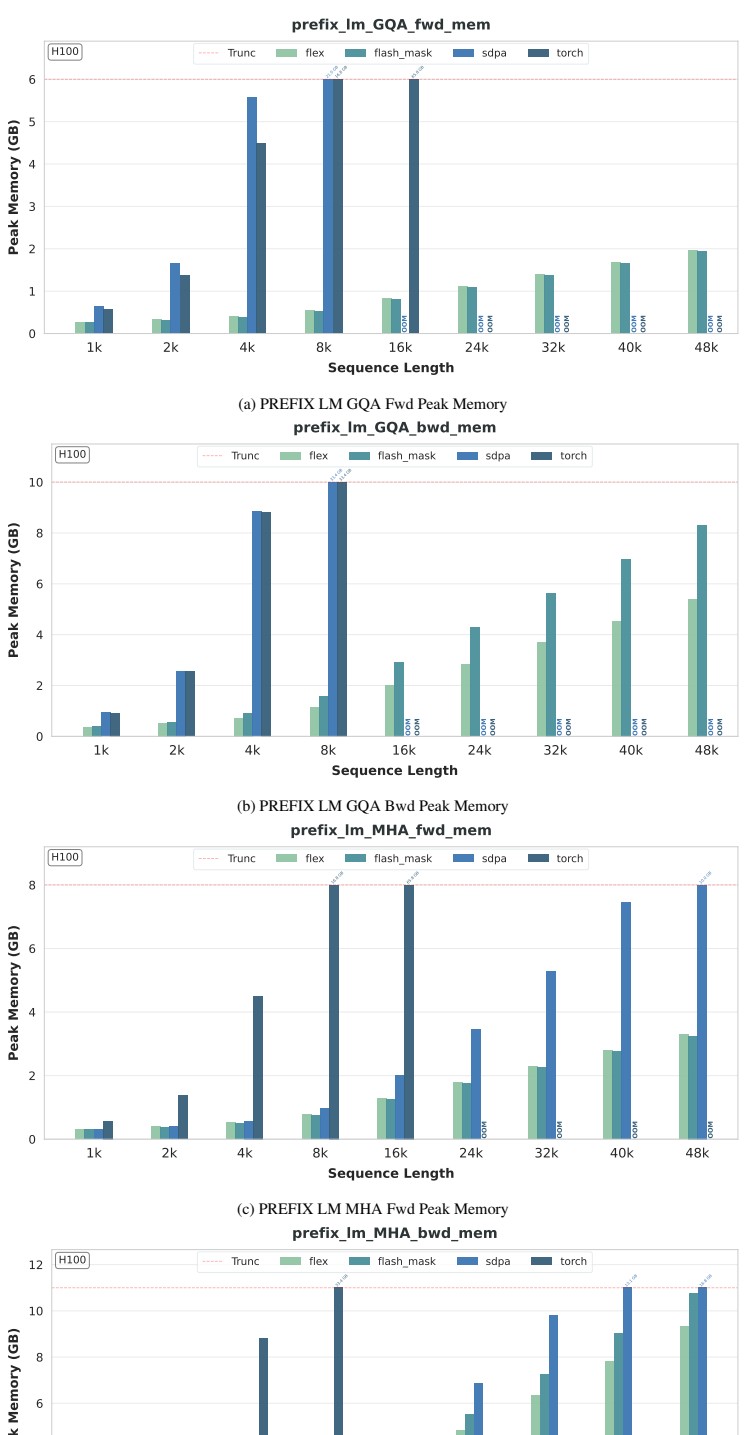

(a) PREFIX LM GQA Fwd Peak Memory

(b) PREFIX LM GQA Bwd Peak Memory

(c) PREFIX LM MHA Fwd Peak Memory

(d) PREFIX LM MHA Bwd Peak Memory

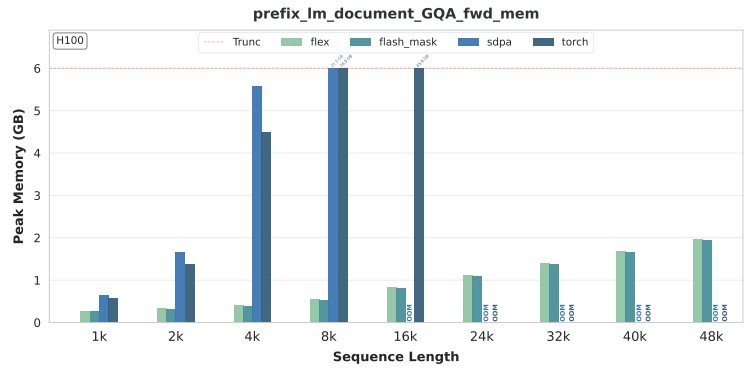

(e) PREFIX LM DOCUMENT GQA Fwd Peak Memory

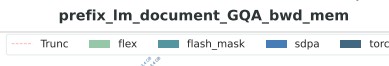

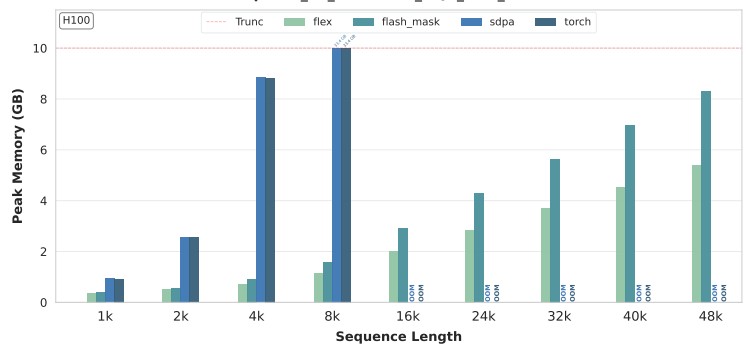

(f) PREFIX LM DOCUMENT GQA Bwd Peak Memory

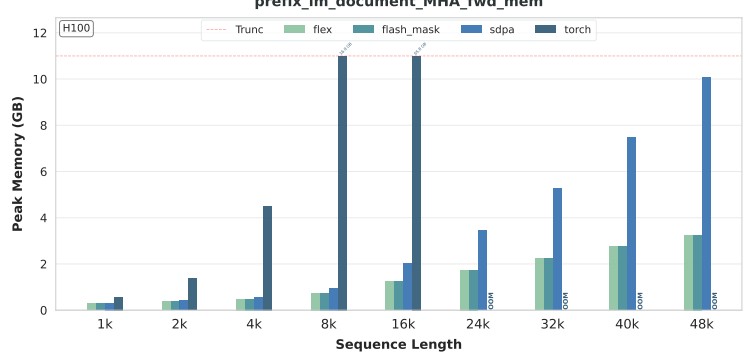

(g) PREFIX LM DOCUMENT MHA Fwd Peak Memory

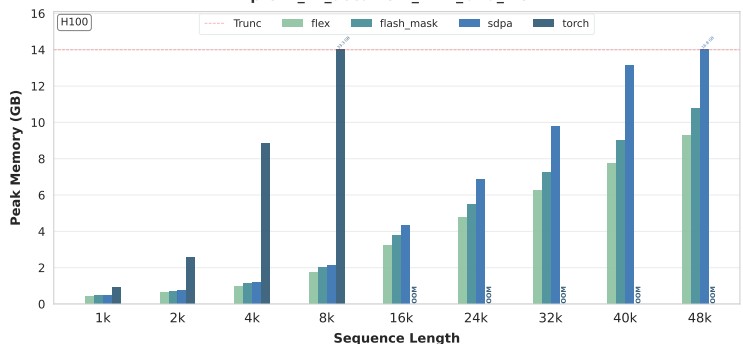

(h) PREFIX LM DOCUMENT MHA Bwd Peak Memory

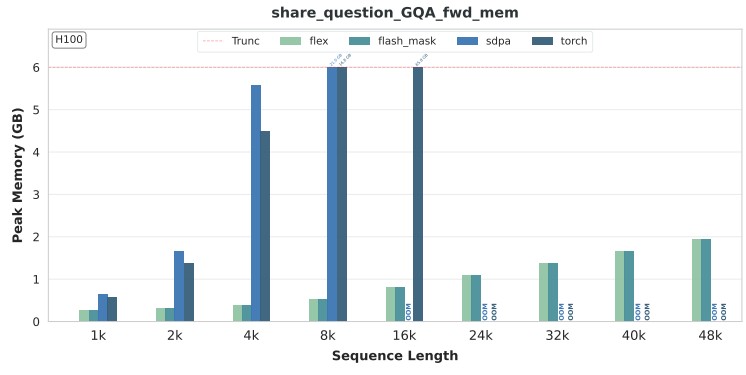

(i) SHARE QUESTION GQA Fwd Peak Memory

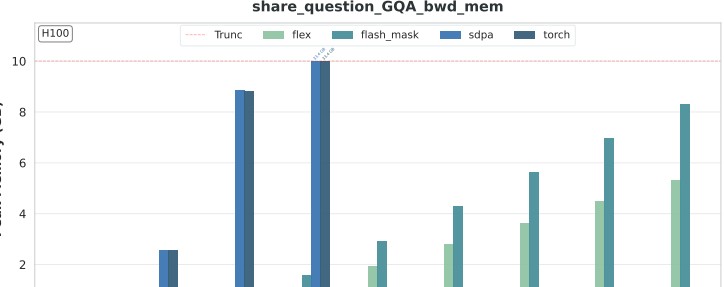

(j) SHARE QUESTION GQA Bwd Peak Memory

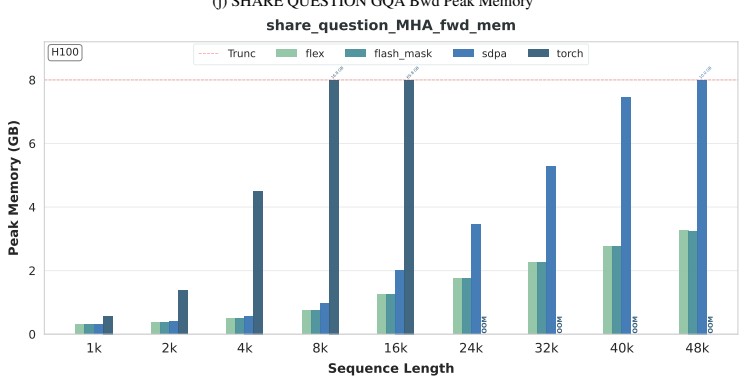

(k) SHARE QUESTION MHA Fwd Peak Memory

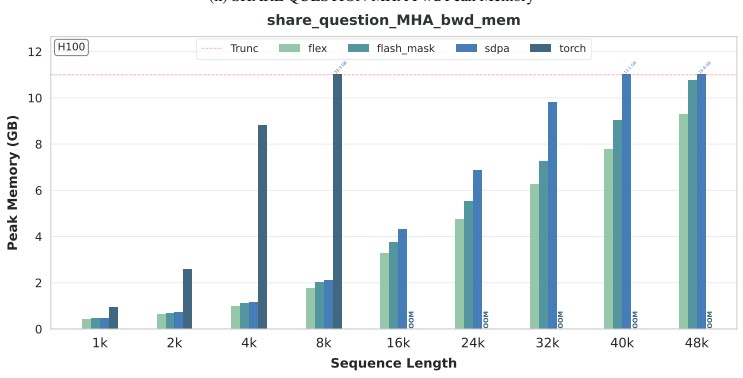

(l) SHARE QUESTION MHA Bwd Peak Memory

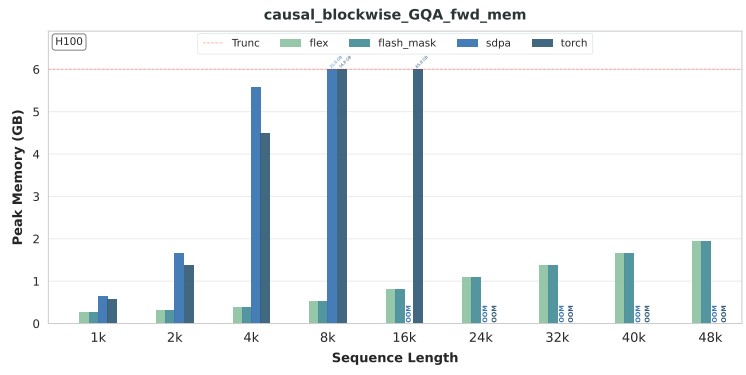

(m) CAUSAL BLOCKWISE GQA Fwd Peak Memory

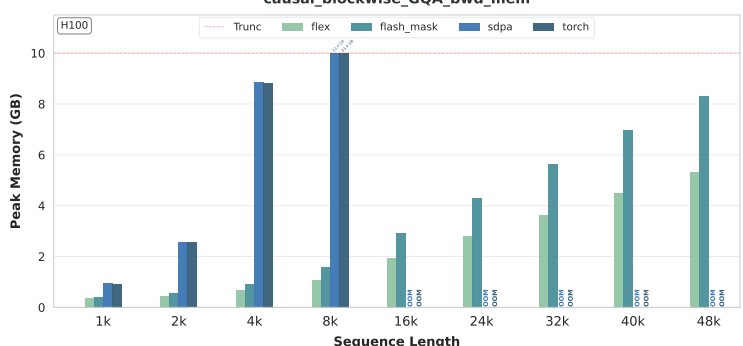

(n) CAUSAL BLOCKWISE GQA Bwd Peak Memory

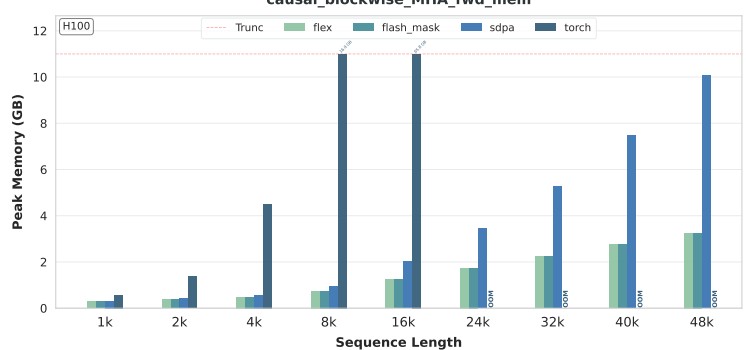

(o) CAUSAL BLOCKWISE MHA Fwd Peak Memory

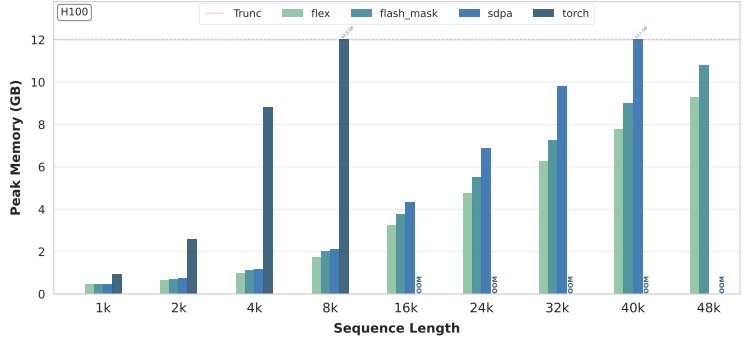

(p) CAUSAL BLOCKWISE MHA Bwd Peak Memory

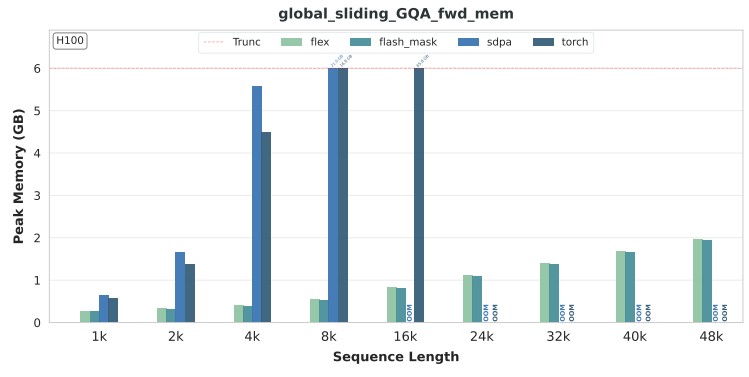

(q) GLOBAL SLIDING GQA Fwd Peak Memory

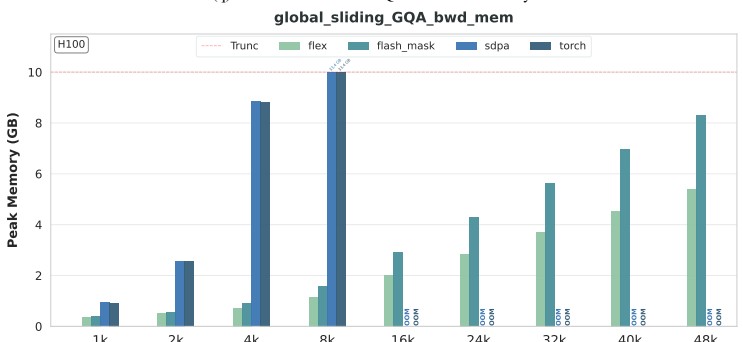

(r) GLOBAL SLIDING GQA Bwd Peak Memory

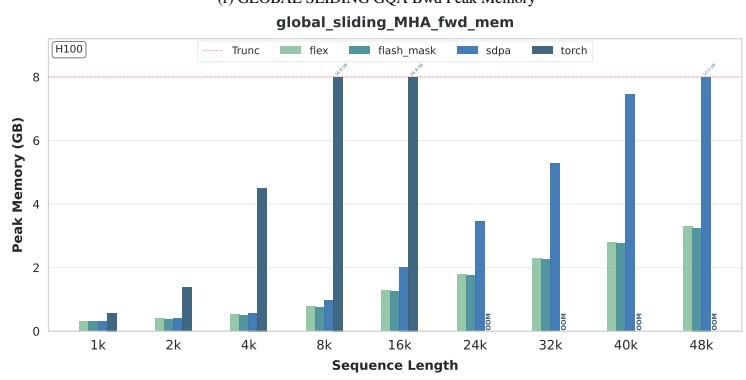

(s) GLOBAL SLIDING MHA Fwd Peak Memory

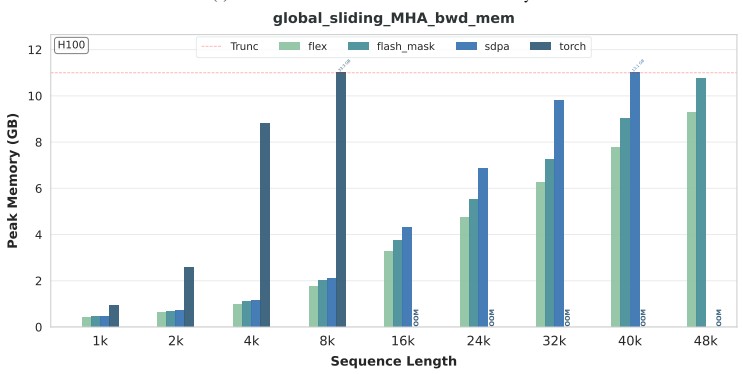

(t) GLOBAL SLIDING MHA Bwd Peak Memory

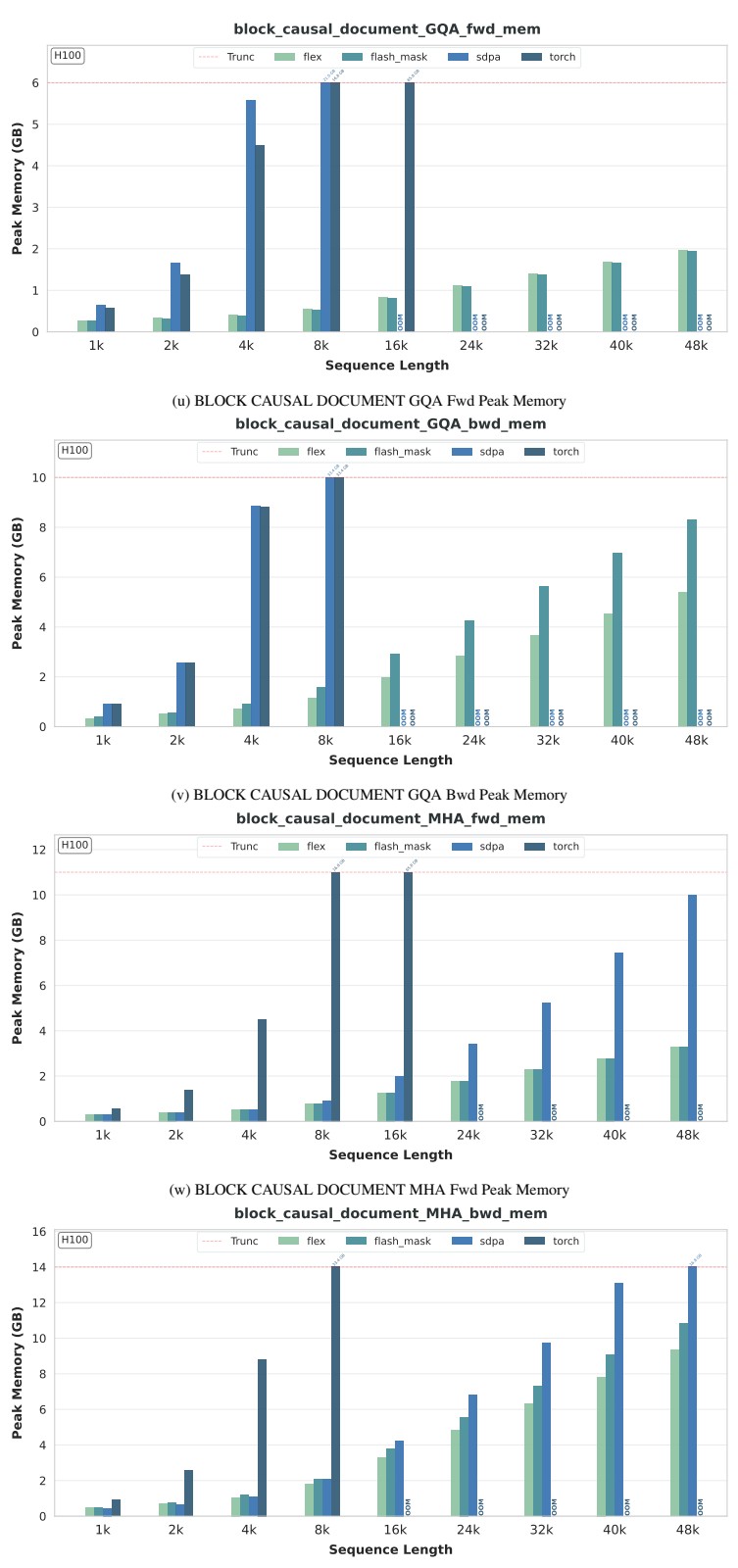

(u) BLOCK CAUSAL DOCUMENT GQA Fwd Peak Memory

(v) BLOCK CAUSAL DOCUMENT GQA Bwd Peak Memory

(w) BLOCK CAUSAL DOCUMENT MHA Fwd Peak Memory

(x) BLOCK CAUSAL DOCUMENT MHA Bwd Peak Memory

Figure 18: Peak Memory of Static Heterogeneous Masks

### A.6 Details of sparse kernel performance

In this section, we present the detailed comparisons for sparse kernels, including TFLOPS performance and peak memory usage. There are only partial baselines in each figure or table because current block sparse kernels have such limitations in functionality: VSA and Triton VSA do not support GQA and 128 block size, FA2 Sparse does not support 64 block size, FA2 Sparse and FlashInfer do not support backward computation, and FlexAttention faces severe memory issues so we discuss it separately.

#### A.6.1 Performance metric: FLOPs

**TFLOPS of MHA with 64 block size.** We report the performance of block sparse attention kernels with MHA and 64 block size in Figure 19. Only VSA, Triton VSA and FlashInfer support this setting. The left side shows the foward FLOPS with different sparsity ratios while the right side shows the backward FLOPS. Our findings reveal that VSA performs stably across different sparsity ratios, showing their robustness with adaptive computation. FlashInfer also performs stably but suffers from OOM issues with higher sparsity ratios because there are more blocks to compute, causing memory overhead with its metadata. However, we find that the performance of VSA reduces with the increase of the context length, especially for backward computation. This indicates that there may exists optimization opportunities for larger context lengths and backward computation for training scenarios.

**TFLOPS of MHA with 128 block size.** We report the performance of block sparse attention kernels with MHA and 128 block size with forward computation in Table 5. Only FlashInfer and FA2 Sparse support this setting. Comparing with the performance of 64 block size in Figure 19, the TFLOPS of FlashInfer in 128 block size increases about 2x, proportional to the block size increase, showing the scalability of FlashInfer block sparse kernels. Its OOM issues also decrease compared with 64 block size, because larger block size means smaller number of blocks, leading to smaller metadata storage. For the TFLOPS of FA2 Sparse, it demonstrates robustness across different sparsity ratios and context lengths. Its average performance is about 300+ FLOPS because it is not optimized for NVIDIA Hopper GPUs. There exists opportunities for tailored optimizations for specific hardware platforms to unleash the hardware performance.

**Separate explanation of FlexAttention TFLOPS.** FlexAttention is separately discussed because it is hard to generate the dynamic block sparse block through compilation, causing severe memory overhead with $O(S^2)$ block mask representations. So we only test its TFLOPS performance with 4 heads and 16K context length in Table 6. It performs bad in 64 block size due to the lack of optimizations with small block size. While in 128 block size, it is comparable to TFLOPS of FA2 Sparse in Table 5. The reason behind is that FlexAttention uses 128 as its default block size, so the kernels it generates demonstrate relative good TFLOPS. It also supports backward computation with the similar performance.

**TFLOPS of trainable block sparse attention.**

For NSA (Yuan et al., 2025), we follow the same setting, with 64 query heads and 4 key, value heads. In this configuration, each query group (comprising 16 query heads) selects the same key, value indices. Each query selects 32 Key-blocks with a block size of 64, resulting in a total of 2048 selected tokens. Since the selection attention module dominates the end-to-end overhead and aligns with our block sparse framework, our experiments focus on this module. This configuration maps to $q\_block\_size = 16$ and $k\_block\_size = 64$ within our block sparse attention benchmark framework. We conduct both NSA-Triton and NSA-Tilelang in the benchmark, as shown in Table 10, implemented by Triton and Tilelang, respectively.

For DSA (DeepSeek-AI, 2025), since it utilizes token-level sparse attention and is designed based on DeepseekMLA model architecture, we set the configuration to 128 query heads and 1 key, value head, matching DeepseekMLA. Each query group therefore consists of 128 query heads sharing the same key, value indices. We set the top-k selection size to 2048 tokens. In our framework, this corresponds to $q\_block\_size = 128$ and $k\_block\_size = 1$. We benchmark this setup using the DSA-Tilelang and DSA-FlashMLA, as shown in Table 11, implemented by Tilelang and CUDA, respectively.

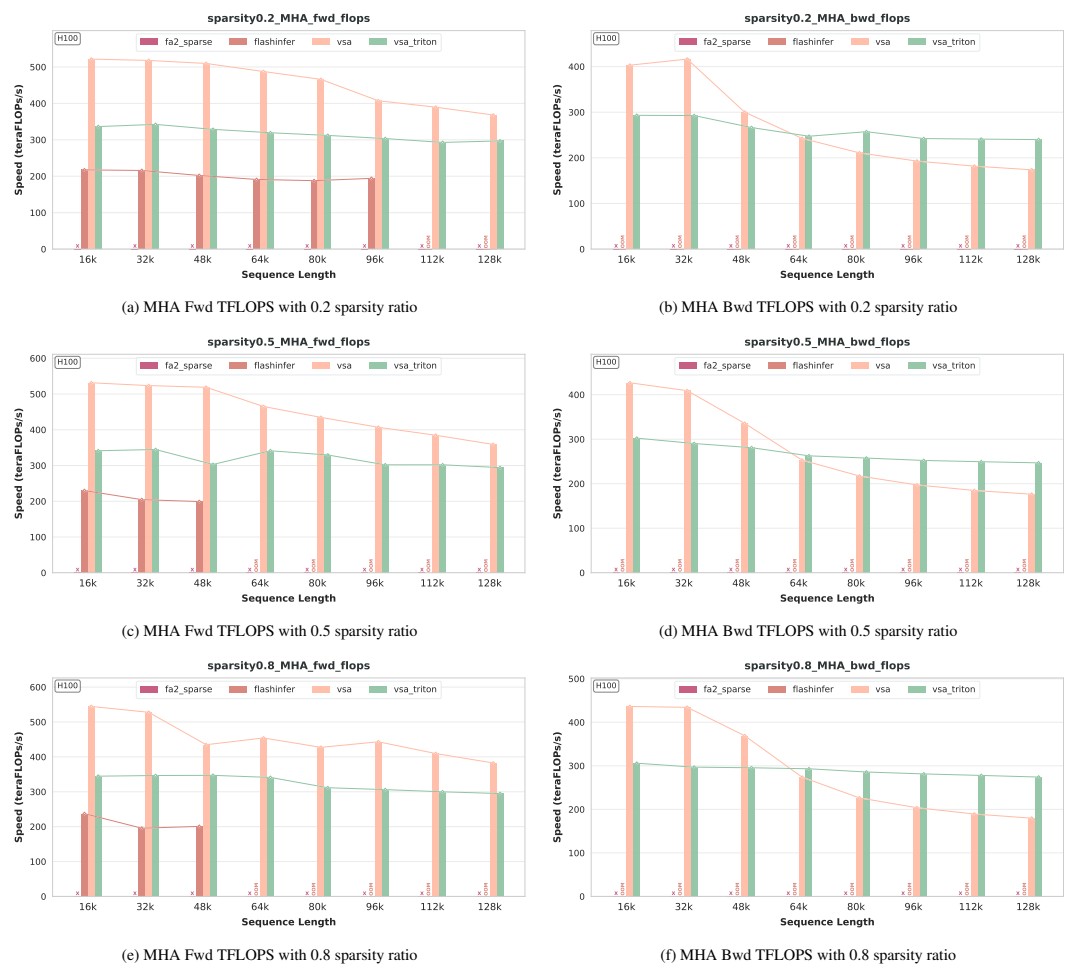

Figure 19: TFLOPS of block sparse attention kernels with 64 block size and MHA

### A.6.2 PERFORMANCE METRIC: PEAK MEMORY USAGE

**Peak memory of MHA with 64 block size.** We report the peak memory of block sparse attention kernels with MHA and 64 block sizes in Figure 20. Only VSA, Triton VSA and FlashInfer support this setting. The left side window shows the forward pass with different sparsity ratio while the right side window shows the backward pass. VSA and VSA_trition share the same memory usage while flashinfer exhibit higher GPU memory consumption which may because of its metadata representation for block sparse. It is also clear that The GPU memory consumption of VSA and VSA_trition shows almost no correlation with the sparsity_ratio. In contrast, the memory footprint of FlashInfer grows in proportion to the increase in the sparsity_ratio. **Peak memory in GQA scenarios.** As shown in Table 7, in GQA scenarios, FlashInfer's GPU memory usage is significantly lower than in MHA scenarios, for both block sizes of 64 and 128. This is because the metadata required to represent the block sparse structure is greatly reduced. As for FA2 Sparse, its' memory consumption is greatly lower than flashinfer, which indicates flashinfer's poor performance in terms of GPU memory usage. **Peak memory of MHA with 128 block size.** We report the peak memory of block sparse attention kernels with MHA and 64 block sizes in Table 8, compared with MHA with 64 block size, flashinfer consumes less GPU memory. However, FA2 Sparse utilizes more GPU memory compared with GQA scenerio. **Separate explanation of FlexAttention peak memory.** FlexAttention is separately discussed because it is hard to generate the dynamic block sparse block through compilation, causing severe memory overhead with $O(S^2)$ block mask representations. So we only test its TFLOPS performance with 4 heads and 16K context length in Table 6, most of FlexAttention's

Table 4: TFLOPs of Sparse Kernels for GQA (64:8) Forward

Note: Seqlen = Sequence Length, SR = Sparsity Ratio, ✗ = Not Supported, H100 GPU.

| Kernels | Seqlen | BlockSize 64 | | | BlockSize 128 | | |
|---|---|---|---|---|---|---|---|
| | | SR 0.2 | SR 0.5 | SR 0.8 | SR 0.2 | SR 0.5 | SR 0.8 |
| FlashInfer | 16k | 211.78 | 236.30 | 247.83 | 387.63 | 449.79 | 472.52 |
| | 32k | 232.81 | 243.77 | 203.54 | 422.95 | 455.97 | 485.06 |
| | 48k | 211.50 | 218.30 | 215.47 | 413.11 | 434.61 | 432.16 |
| | 64k | 190.73 | 210.82 | 204.16 | 414.25 | 427.77 | 399.22 |
| | 80k | 201.71 | 198.39 | 206.43 | 351.87 | 393.34 | 410.88 |
| | 96k | 202.03 | 197.48 | 201.42 | 393.16 | 387.89 | 398.73 |
| | 112k | 198.14 | 196.82 | 199.41 | 393.41 | 389.33 | 397.02 |
| | 128k | 196.47 | 195.61 | 197.93 | 389.55 | 396.90 | 393.80 |
| FA2 Sparse | 16k | ✗ | ✗ | ✗ | 312.54 | 326.62 | 332.00 |
| | 32k | ✗ | ✗ | ✗ | 327.46 | 333.45 | 339.16 |
| | 48k | ✗ | ✗ | ✗ | 331.99 | 281.27 | 331.24 |
| | 64k | ✗ | ✗ | ✗ | 328.84 | 329.88 | 329.99 |
| | 80k | ✗ | ✗ | ✗ | 293.43 | 330.62 | 319.52 |
| | 96k | ✗ | ✗ | ✗ | 328.38 | 318.18 | 321.05 |
| | 112k | ✗ | ✗ | ✗ | 298.22 | 316.92 | 322.47 |
| | 128k | ✗ | ✗ | ✗ | 317.48 | 317.11 | 321.04 |

Table 5: Forward TFLOPs of Sparse MHA Kernels (64:64, Block Size = 128)

Note: Seqlen = Sequence Length, SR = Sparsity Ratio, ✗ = Not Supported, H100 GPU.

| Kernels | Seqlen | SR 0.2 | SR 0.5 | SR 0.8 |
|---|---|---|---|---|
| FlashInfer | 16k | 414.93 | 447.76 | 484.21 |
| | 32k | 425.38 | 416.37 | 425.88 |
| | 48k | 407.67 | 401.60 | 402.69 |
| | 64k | 398.01 | 394.79 | 392.05 |
| | 80k | 342.30 | 384.46 | OOM |
| | 96k | 388.09 | OOM | OOM |
| | 112k | 380.67 | OOM | OOM |
| | 128k | 384.05 | OOM | OOM |
| FA2 Sparse | 16k | 315.12 | 326.09 | 331.27 |
| | 32k | 327.63 | 333.32 | 337.07 |
| | 48k | 331.79 | 282.22 | 335.95 |
| | 64k | 325.86 | 330.73 | 329.45 |
| | 80k | 295.20 | 328.64 | 318.44 |
| | 96k | 325.09 | 317.42 | 320.26 |
| | 112k | 301.61 | 318.13 | 322.56 |
| | 128k | 308.39 | 317.80 | 321.67 |

GPU memory is consumed during mask creation, while its runtime memory usage is not high. In block sparse scenarios, an efficient mask representation is crucial for GPU memory consumption.

Table 6: TFLOPs of Sparse FlexAttention (SeqLen = 16K)

Note: Seqlen = Sequence Length, SR = Sparsity Ratio, H100 GPU.

| Mode | SR | BlockSize 64 | | BlockSize 128 | |
|---|---|---|---|---|---|
| | | Forward | Backward | Forward | Backward |
| GQA (4:1) | 0.2 | 29.05 | 57.22 | 339.74 | 299.43 |
| | 0.5 | 48.21 | 95.33 | 352.09 | 312.76 |
| | 0.8 | 99.80 | 170.40 | 358.55 | 325.52 |
| MHA (4:4) | 0.2 | 28.75 | 63.14 | 341.55 | 305.37 |
| | 0.5 | 48.76 | 103.37 | 345.84 | 337.60 |
| | 0.8 | 101.58 | 186.44 | 352.18 | 349.00 |

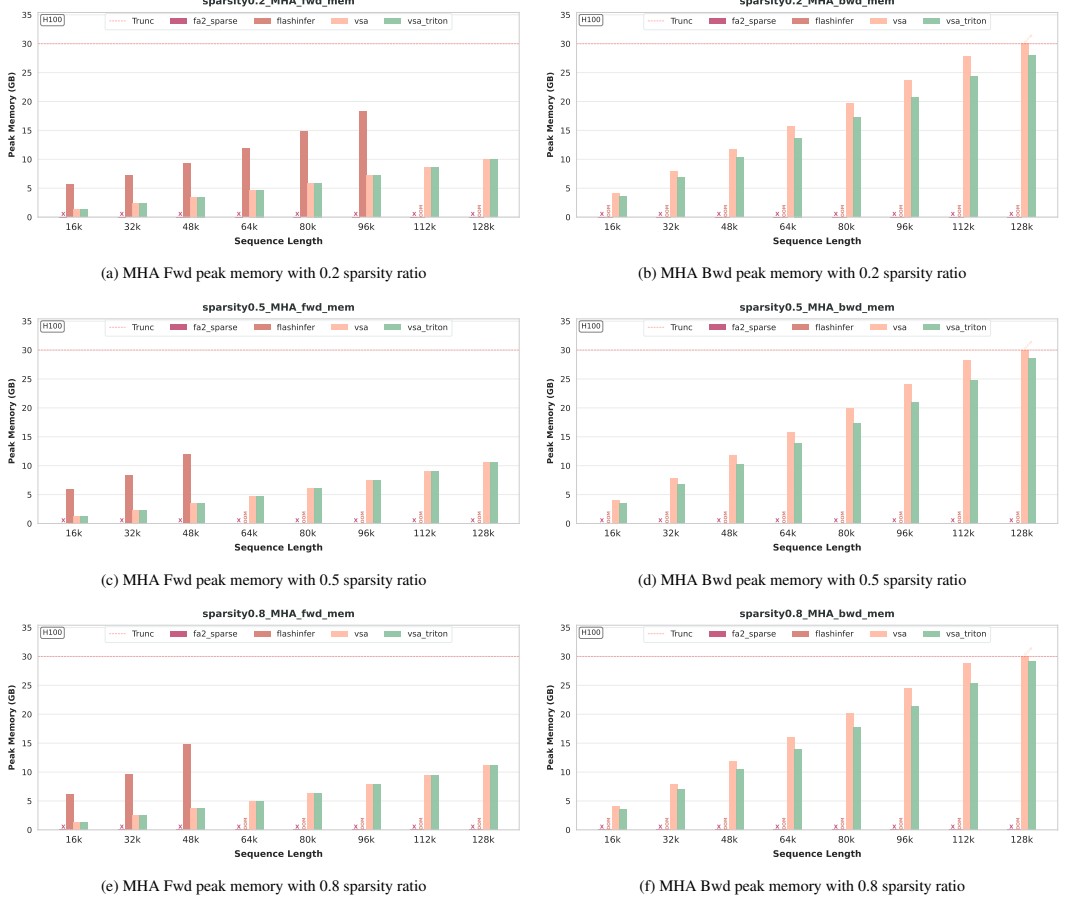

(a) MHA Fwd peak memory with 0.2 sparsity ratio

(b) MHA Bwd peak memory with 0.2 sparsity ratio

(c) MHA Fwd peak memory with 0.5 sparsity ratio

(d) MHA Bwd peak memory with 0.5 sparsity ratio

(e) MHA Fwd peak memory with 0.8 sparsity ratio

(f) MHA Bwd peak memory with 0.8 sparsity ratio

Figure 20: Peak memory of block sparse attention kernels with 64 block size and MHA

Table 7: Peak Memory (GB) of Sparse Kernels for GQA (64:8) Forward

Note: Seqlen = Sequence Length, SR = Sparsity Ratio, ✗ = Not Supported, H100 GPU.

| Kernels | Seqlen | BlockSize 64 | | | BlockSize 128 | | |
|---|---|---|---|---|---|---|---|
| | | SR 0.2 | SR 0.5 | SR 0.8 | SR 0.2 | SR 0.5 | SR 0.8 |
| FlashInfer | 16k | 5.468 | 5.502 | 5.540 | 5.448 | 5.466 | 5.485 |
| | 32k | 6.620 | 6.770 | 6.920 | 6.552 | 6.627 | 6.702 |
| | 48k | 7.837 | 8.174 | 8.512 | 7.684 | 7.853 | 8.022 |
| | 64k | 9.115 | 9.715 | 10.317 | 8.845 | 9.145 | 9.444 |
| | 80k | 10.455 | 11.393 | 12.330 | 10.034 | 10.502 | 10.970 |
| | 96k | 11.857 | 13.207 | 14.557 | 11.250 | 11.923 | 12.599 |
| | 112k | 13.322 | 15.159 | 16.997 | 12.493 | 13.413 | 14.331 |
| | 128k | 14.846 | 17.247 | 19.647 | 13.766 | 14.964 | 16.165 |
| FA2 Sparse | 16k | ✗ | ✗ | ✗ | 0.810 | 0.810 | 0.810 |
| | 32k | ✗ | ✗ | ✗ | 1.393 | 1.393 | 1.393 |
| | 48k | ✗ | ✗ | ✗ | 1.986 | 1.986 | 1.986 |
| | 64k | ✗ | ✗ | ✗ | 2.590 | 2.590 | 2.590 |
| | 80k | ✗ | ✗ | ✗ | 3.205 | 3.205 | 3.205 |
| | 96k | ✗ | ✗ | ✗ | 3.830 | 3.830 | 3.830 |
| | 112k | ✗ | ✗ | ✗ | 4.466 | 4.466 | 4.466 |
| | 128k | ✗ | ✗ | ✗ | 5.113 | 5.113 | 5.113 |

Table 8: Forward Peak Memory of Sparse MHA Kernels (64:64, Block Size = 128)

Note: Seqlen = Sequence Length, SR = Sparsity Ratio, ✗ = Not Supported, H100 GPU.

| Kernels | Seqlen | SR 0.2 | SR 0.5 | SR 0.8 |
|---|---|---|---|---|
| FlashInfer | 16k | 5.477 | 5.627 | 5.776 |
| | 32k | 6.791 | 7.392 | 7.991 |
| | 48k | 8.316 | 9.666 | 11.015 |
| | 64k | 10.051 | 12.452 | 14.849 |
| | 80k | 11.994 | 15.746 | OOM |
| | 96k | 14.148 | OOM | OOM |
| | 112k | 16.514 | OOM | OOM |
| | 128k | 19.085 | OOM | OOM |
| FA2 Sparse | 16k | 1.251 | 1.251 | 1.251 |
| | 32k | 2.281 | 2.281 | 2.281 |
| | 48k | 3.329 | 3.329 | 3.329 |
| | 64k | 4.395 | 4.395 | 4.395 |
| | 80k | 5.478 | 5.478 | 5.478 |
| | 96k | 6.587 | 6.578 | 6.578 |
| | 112k | 7.696 | 7.696 | 7.696 |
| | 128k | 8.832 | 8.832 | 8.832 |

Table 9: Peak Memory (GB) of Sparse FlexAttention (SeqLen = 16K)

Note: Seqlen = Sequence Length, SR = Sparsity Ratio, H100 GPU.

| Mode | SR | BlockSize 64 | | BlockSize 128 | |
|---|---|---|---|---|---|
| | | Forward | Backward | Forward | Backward |
| GQA (4:1) | 0.2 | 0.285 | 0.383 | 0.294 | 0.392 |
| | 0.5 | 0.287 | 0.384 | 0.296 | 0.394 |
| | 0.8 | 0.288 | 0.386 | 0.297 | 0.395 |
| MHA (4:4) | 0.2 | 0.304 | 0.449 | 0.313 | 0.458 |
| | 0.5 | 0.306 | 0.450 | 0.315 | 0.460 |
| | 0.8 | 0.307 | 0.452 | 0.316 | 0.461 |

Table 10: TFLOPs of NSA for GQA (64:8)

Note: Seqlen = Sequence Length, H100 GPU.

| Seqlen | NSA-Triton | | NSA-Tilelang | |
|--------|---------|----------|---------|----------|
|        | Forward | Backward | Forward | Backward |
| 32k  | 97.35 | 140.34 | 60.91 | 3.97 |
| 64k  | 90.5  | 132.99 | 49.19 | 1.08 |
| 96k  | 83.39 | 129.65 | 45.16 | 1.3  |
| 128k | 76.74 | 120.62 | 39.5  | 0.55 |

Table 11: TFLOPs of DSA for GQA (64:8)

Note: Seqlen = Sequence Length, H100 GPU.

| Seqlen | DSA-Tilelang | | DSA-FlashMLA | |
|--------|---------|----------|---------|----------|
|        | Forward | Backward | Forward | Backward |
| 32k  | 334.13 | 114.64 | 610.49 | ✗ |
| 64k  | 326.89 | 103.17 | 607.13 | ✗ |
| 96k  | 331.28 | 98.49  | 589.91 | ✗ |
| 128k | 331.99 | 95.95  | 555.27 | ✗ |

## A.7 DETAILS OF CONTEXT PARALLEL ATTENTION PERFORMANCE

**Performance of ring P2P.** The main limitation of ring P2P is that its communication volume cannot be adjusted. In each iteration, while computing with the KV pairs for the current stage, each device simultaneously sends its own KV to the next device in the topology and receives the KV needed for the next stage. In a multi-node, multi-GPU setup, all GPUs are connected in a ring-based P2P topology. The communication bottleneck is determined by the slowest inter-node link, forcing intra-node communication to synchronize with the inter-node transfers, which leads to substantial overall bandwidth underutilization.

Ultimately, the overall efficiency of Ring P2P is determined by the actual per-GPU computation, which manifests in whether communication can be overlapped with computation. The FULL scenario represents the optimal performance case for Ring P2P. As shown in the Figure 21, in this scenario, each GPU executes the largest per-kernel computation. When inter-node communication can also be effectively overlapped with computation, further scaling the distributed setup does not significantly change the balance between computation and communication efficiency, so the overall computational efficiency remains essentially constant.

In the CAUSAL scenario, the per-GPU communication volume of Ring P2P remains the same as in the FULL scenario. Even after load balancing, the per-stage computation per GPU is roughly half of that in the FULL scenario (except for the first stage, which is 3/4). The reduced computation may no longer fully overlap with communication, leading to a performance drop in Ring P2P under CAUSAL, as shown in the Figure 22. During the forward pass, when scaling to 8 nodes, we observe further performance degradation. We attribute this mainly to machine instability: even if only a single GPU underperforming in the ring, for example from a sudden drop in computation efficiency, can stall the entire topology and greatly reduce overall efficiency. In large-scale distributed settings, such effects are inevitable, and we report the experimental results faithfully.

For the DOCUMENT scenario, as shown in the Figure 23, Ring P2P exhibits similar trends in both the CAUSAL DOCUMENT and FULL DOCUMENT settings. Unlike the FULL/CAUSAL scenarios, it does not maintain a relatively constant trend, which is expected. First, when handling variable-length data, each segment must be padded according to its specific scale, leading to significant variation in per-GPU computation per iteration, while communication volume remains constant. Second, sampling of variable-length data introduces differences in computational sparsity across iterations, resulting in fluctuations in the overall trend.

**Hybrid Design.** USP and LoongTrain share very similar overall architectures, using the Ulysses design intra-node and Ring P2P inter-node. Overall, they achieve significant and stable performance improvements in the FULL and CAUSAL scenarios, as shown in Figure 21 and 22. In the FULL DOCUMENT and CAUSAL DOCUMENT scenarios, performance gradually decreases due to reduced overall computation, which aligns with expectations. Additionally, in the DOCUMENT scenario, as shown in Figure 24 and 23, both architectures demonstrate improved stability, mitigating the limitations of using only Ulysses or Ring P2P.

Here, we primarily explain why LoongTrain generally outperforms USP during the forward pass, yet performs on par with or slightly worse than USP during the backward pass.

In our benchmark reproduction and optimization, USP and Ring P2P both use the same RingAttn class. In the ring-topology iterations, the forward and backward data flows are essentially opposite. For example, during the forward pass, $GPU_i$ receives data from $GPU_{i-1}$ and sends data to $GPU_{i+1}$; in the backward pass, $GPU_i$ receives from $GPU_{i+1}$ and sends to $GPU_{i-1}$.

This design is both necessary and reasonable. At the end of the forward pass, $GPU_0$ actually holds the initial-stage KV data of $GPU_1$, and so on, with $GPU_{N-1}$ finally holding $GPU_0$'s initial data. If the backward pass rotated data in the same direction as the forward pass, it would require either an additional P2P communication or storing the initial-stage KV on each GPU in advance, both of which incur extra overhead.

Instead, we exploit the time difference between KV and gradient generation: the backward pass directly continues from the forward-pass KV states in reverse rotation, while gradients computed in the current stage are sent during the next stage. After completing the same number of rotations, each

GPU receives exactly its corresponding gradient (e.g., $GPU_0$ receives the gradient for $KV_0$, and so on).

For LoongTrain's DoubleRingAttn, the forward pass is consistent with USP. In addition, LoongTrain leverages a heterarchical P2P architecture to implement a two-level sliding window, decomposing the full ring topology into intra-window and inter-window groups. The intra-window group is identical to the RingAttn class, but in the first stage, an additional P2P communication is performed for the inter-window group to prefetch the initial data for each GPU after the next inter-window rotation. This design fully utilizes inter-node bandwidth, resulting in superior forward-pass performance compared to USP.

However, LoongTrain's backward pass cannot directly leverage the forward-pass end states. While this poses no issue for the last inter-window, it alters the initial state for each subsequent inter-window, and the final states differ depending on the specific intra- and inter-window configuration. As a result, LoongTrain performs forward and backward passes using the same rotation order. Each GPU additionally stores the initial KV data, and to ensure correct gradient propagation, an extra P2P communication and synchronization of gradients is performed at the end of each inter-window. This guarantees that in the next inter-window, each GPU receives the corresponding KV data and gradients. Consequently, LoongTrain gains no significant backward-pass advantage from the heterarchical architecture, yet overall maintains performance comparable to USP. The observed fluctuations in trends are similarly attributed to the instability introduced by the additional P2P communications.

**Ulysess.** For Ulysess, the results are straightforward: different sampling patterns naturally lead to variations in computation, and we recommend evaluating performance based on the specific application scenario.

## A.8 Analysis of context parallel attention communication

**Ulysess.**

Forward (FWD): 2 All-to-all for $Q$ and $O$, and 2 All-to-all for $K$ and $V$. The communication volume is $2 \cdot \frac{N-1}{N^2} th_{kv}d + 2 \cdot \frac{N-1}{N^2} th_q d$.

Backward (BWD): 2 All-to-all for $dQ$ and $dO$, and 2 All-to-all for $dK$ and $dV$. The communication volume is $2 \cdot \frac{N-1}{N^2} th_{kv}d + 2 \cdot \frac{N-1}{N^2} th_q d$.

Overlap: All communications are exposed outside the computation.

**Ring All-Gather.**

Forward (FWD): 2 All-Gather for $K$ and $V$. The communication volume is $\frac{N-1}{N} th_{kv}d * 2$.

Backward (BWD): 2 All-Gather for $K$ and $V$, and 2 Reduce-Scatter for $dK$ and $dV$. The communication volume is $\frac{N-1}{N} th_{kv}d * 2 + \frac{N-1}{N} th_{kv}d * 2$.

Overlap: All communications are exposed outside the computation.

**Ring P2P.**

Forward (FWD): $N - 1$ P2P for $K$ and $V$. The communication volume is $\frac{N-1}{N} th_{kv}d * 2$.

Backward (BWD): $N - 1$ P2P for $K$ and $V$, and $N - 1$ P2P $dK$ and $dV$. The communication volume is $\frac{N-1}{N} th_{kv}d * 2 + \frac{N-1}{N} th_{kv}d * 2$.

Overlap: In theory, the FULL mask scenario achieves communication-computation overlap. Focusing solely on mask-sparse scenarios, computation and communication start simultaneously at each stage across different ranks, with the communication volume remaining unchanged. The portion of the entire pipeline's communication that can be overlapped is approximately $\frac{\sum_i^m t_m^2/flops}{t/bandwidth} * k$, where $t_m^2$ denotes the single computation area, and $k$ merely represents a correction coefficient that exists as a constant for specific settings.

**USP.** We denote $N_a$ as the Ulysses size and $N_p$ as the Ring-P2P size, such that $N_p \cdot N_a = N$.

Forward (FWD): 2 All-to-all for $Q$ and $O$, 2 All-to-all for $K$ and $V$, and $N_p - 1$ P2P for $K$ and $V$. The communication volume is $\frac{N_a-1}{N_p*N_a^2} th_{kv}d * 2 + \frac{N_a-1}{N_p*N_a^2} th_q d * 2 + \frac{N_p-1}{N_p*N_a} th_{kv}d * 2$.

Backward (BWD): 2 All-to-all for $dQ$ and $dO$, 2 All-to-all for $dK$ and $dV$, $N_p - 1$ P2P for $K$ and $V$, and $N_p - 1$ P2P for $dK$ and $dV$. The communication volume is $\frac{N_a-1}{N_p*N_a^2}th_{kv}d*2 + \frac{N_a-1}{N_p*N_a^2}th_qd*2 + \frac{N_p-1}{N_p*N_a}th_{kv}d*2 + \frac{N_p-1}{N_p*N_a}th_{kv}d*2$.

Overlap: the analysis remains the same as Ulysses and Ring P2P.

**LoongTrain.** We denote $n$ as the intra-ring size and $k$ as the window number, such that $n \cdot k = N_p$.

Forward (FWD): 2 All-to-all for $Q$ and $O$, 2 All-to-all for $K$ and $V$, $(n-1)*k$ P2P for intra $K$ and $V$, and $k-1$ P2P for inter $K$ and $V$. The communication volume is $\frac{N_a-1}{N_p*N_a^2}th_{kv}d*2 + \frac{N_a-1}{N_p*N_a^2}th_qd*2 + \frac{(n-1)*k}{N_p*N_a}th_{kv}d*2 + \frac{k-1}{N_p*N_a}th_{kv}d*2$.

Backward (BWD): 2 All-to-all for $dQ$ and $dO$, 2 All-to-all for $dK$ and $dV$, $(n-1)*k$ P2P for intra $K$ and $V$, $k-1$ P2P for inter $K$ and $V$, $n*k$ P2P for intra $dK$ and $dV$, and $k$ P2P for inter $dK$ and $dV$. The communication volume is $\frac{N_a-1}{N_p*N_a^2}th_{kv}d*2 + \frac{N_a-1}{N_p*N_a^2}th_qd*2 + \frac{(n-1)*k}{N_p*N_a}th_{kv}d*2 + \frac{k-1}{N_p*N_a}th_{kv}d*2 + \frac{1}{N_a}th_{kv}d*2 + \frac{k}{N_p*N_a}th_{kv}d*2$.

Overlap: The forward is consistent with USP. For the backward, the intra and inter communications of $K$ and $V$ are the same as in USP. However, regarding intra $dK$ and $dV$, unlike USP, which places intra $K$, $V$, $dK$, and $dV$ in a single buffer, LoongTrain separates $K$, $V$ from $dK$, $dV$ into two buffers, requiring two separate P2P communications. Thus, in each intra stage of LoongTrain, the intra $dK$ and $dV$ communications are asynchronously initiated only after the completion of computations, overlapping with the correction of $dK$ and $dV$ in the next stage. That said, LoongTrain incurs one additional intra communication for $dK$ and $dV$, leaving approximately one P2P communication exposed. After the final intra $dK$ and $dV$ communication concludes, LoongTrain initiates an inter $dK$ and $dV$ communication, which overlaps with the subsequent intra computations. However, the $dK$ and $dV$ communication for the last window remains exposed.

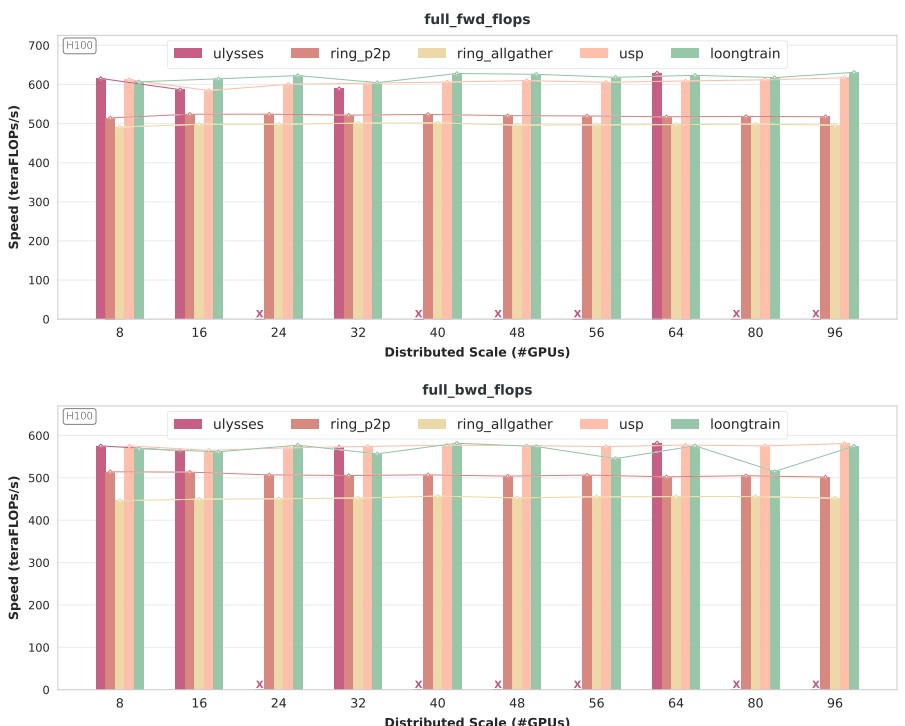

Figure 21: Forward and Backward TFLOPs of Context Parallel Attention on FULL

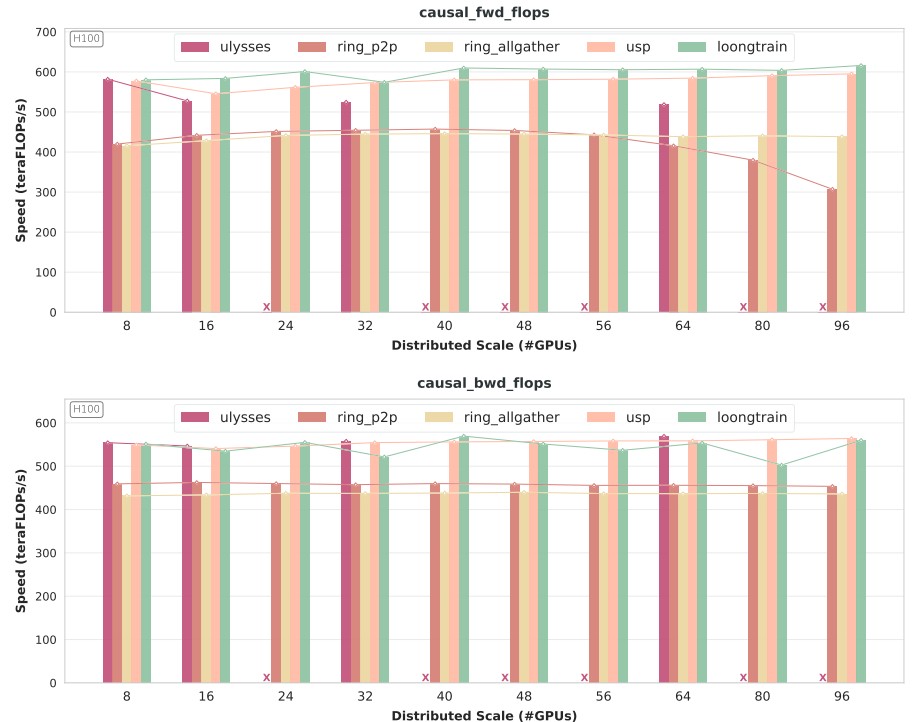

Figure 22: Forward and Backward TFLOPs of Context Parallel Attention on CAUSAL

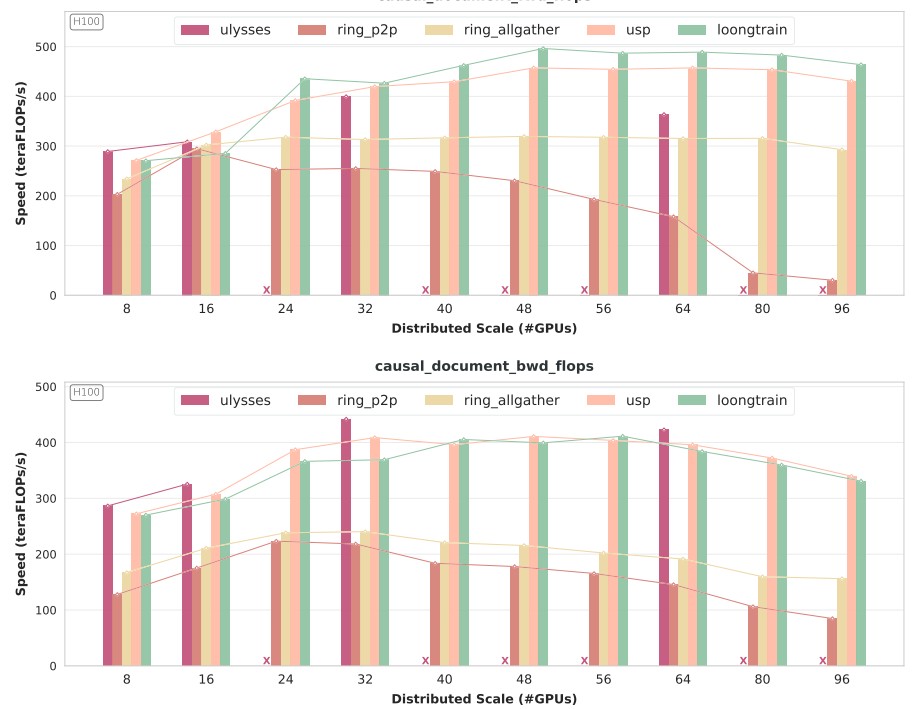

Figure 23: Forward and Backward TFLOPs of Context Parallel Attention on CAUSAL DOCU-MENT

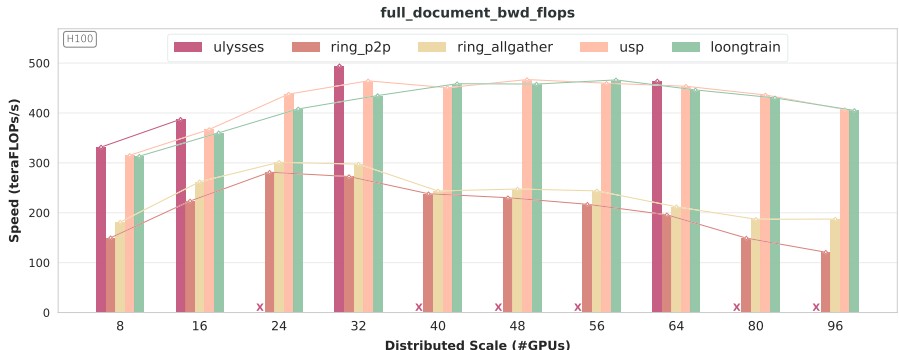

Figure 24: Backward TFLOPs of Context Parallel Attention on FULL DOCUMENT

### A.9    DETAILS OF DENSE KERNEL PERFORMANCE ON A800

We focus on reporting the FLOPs performance of the dense kernels on A800, since the memory consumption exhibits only small variations.

First, document input layout for the cuDNN fused kernel is specifically designed for the Hopper architecture. Therefore, we attempted to expand the input into a basic four-dimensional format to meet the kernel's layout requirements. However, in most scenarios, the kernel still could not provide support, and we ultimately excluded these experiments. Additionally, FA3 itself is a highly optimized implementation targeting the Hopper architecture, so we excluded the FA3 kernel as well. Similarly, FA2 only supports homogeneous masks, yet it still achieves competitive, often optimal performance in homogeneous mask scenarios, as shown in Figure 25 and 26.

It is worth noting that under the sliding window mask configuration, due to the window size design, when the sequence length is 1k, the configuration degenerates into a FULL mask, as shown in Figure 27. In such cases, all optimized kernels perform extremely well, which is a reasonable outcome of the experimental setup rather than an inherent advantage of any particular kernel.

FlashMask demonstrates comparable or even superior performance to FA2 across multiple experiments with homogeneous masks, and its main advantage is its native support for the vast majority of heterogeneous mask patterns, as shown in Figure 30. In these irregular sparse scenarios, FlashMask achieved the best results in our experiments.

We observed that FlexAttention delivers strong performance during the forward pass, but experiences a noticeable drop in backward performance, as shown in Figure 28. This is likely attributable to a combination of compilation optimizations, hardware capabilities, and evaluation methodology. We report this phenomenon faithfully in the paper while maintaining all experimental methods and measurement criteria consistent across all kernels.

Finally, although the Naive Torch implementation and PyTorch SDPA can handle arbitrary mask types, both inevitably face significant computational and memory overhead, as shown in Figure 31 and 29. As a result, their performance is far inferior to highly optimized kernels in most scenarios. This further suggests that for heterogeneous sparse attention, the fundamental bottleneck lies in kernel-level scheduling strategies and compute pipeline optimization, rather than merely mask support capability.

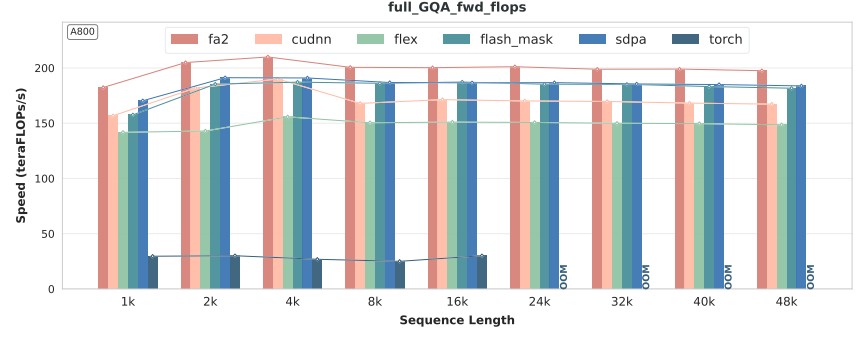

(a) FULL GQA Fwd TFLOPS on NVIDIA A800 GPU

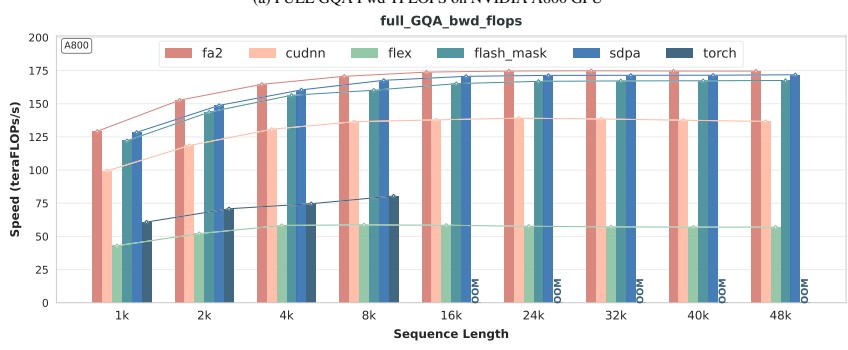

(b) FULL GQA Bwd TFLOPS on NVIDIA A800 GPU

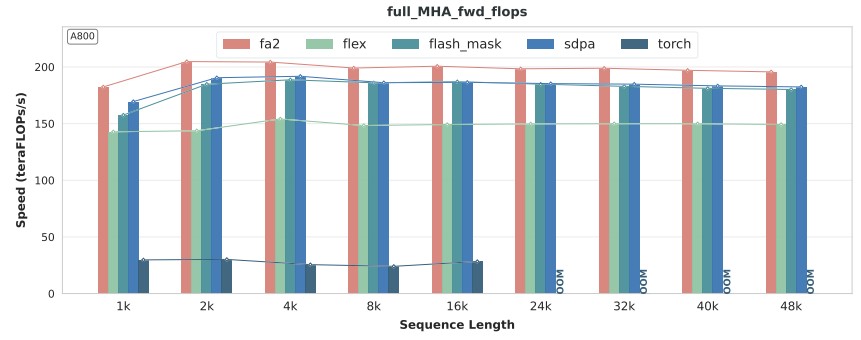

(c) FULL MHA Fwd TFLOPS on NVIDIA A800 GPU

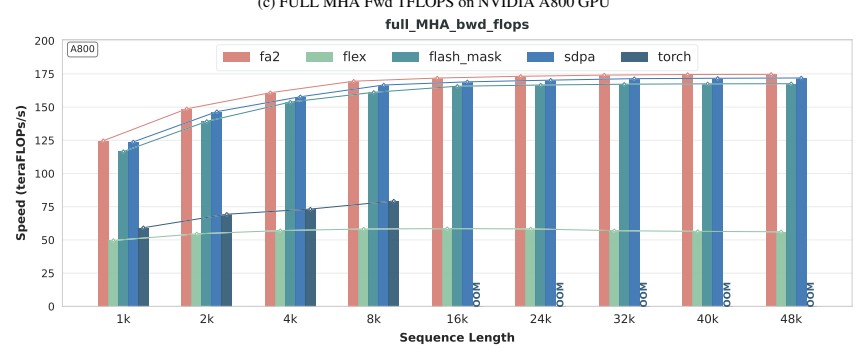

(d) FULL MHA Bwd TFLOPS on NVIDIA A800 GPU

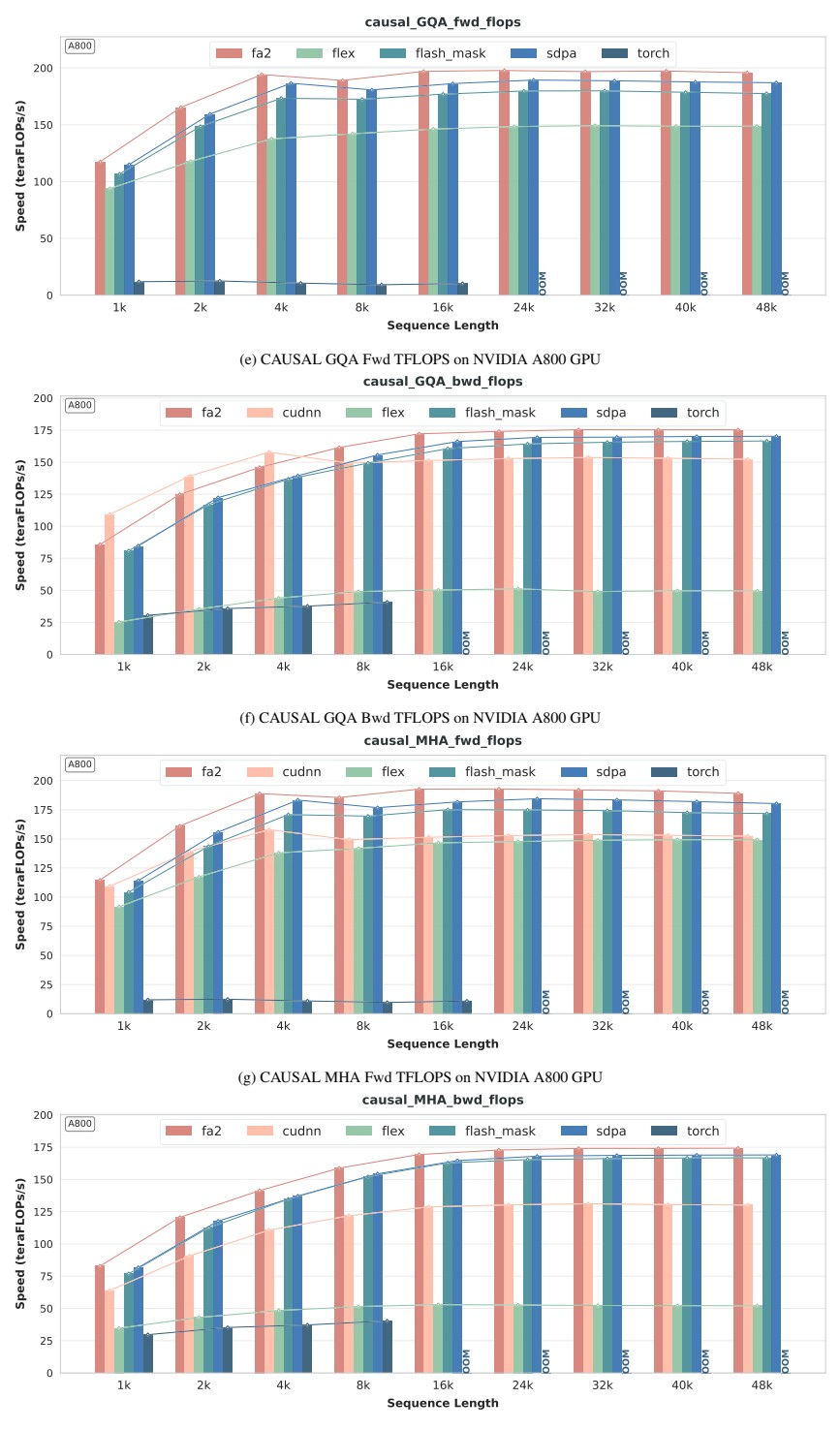

(e) CAUSAL GQA Fwd TFLOPS on NVIDIA A800 GPU

(f) CAUSAL GQA Bwd TFLOPS on NVIDIA A800 GPU

(g) CAUSAL MHA Fwd TFLOPS on NVIDIA A800 GPU

(h) CAUSAL MHA Bwd TFLOPS on NVIDIA A800 GPU

Figure 25: TFLOPS of FULL and CAUSAL on NVIDIA A800 GPU

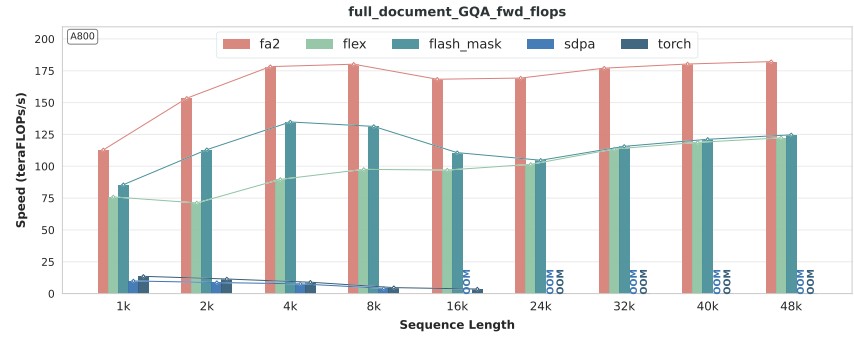

(a) FULL DOCUMENT GQA Fwd TFLOPS on NVIDIA A800 GPU

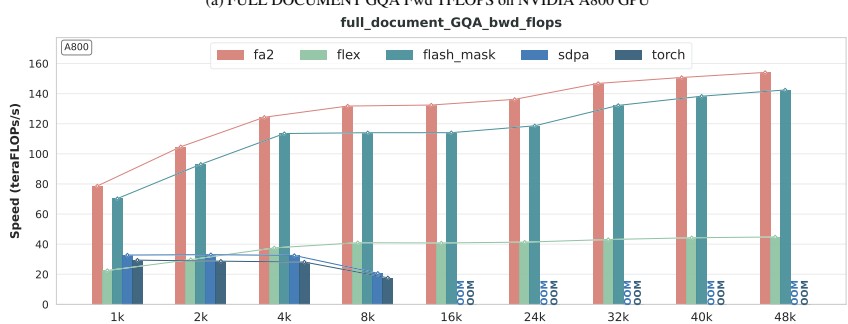

(b) FULL DOCUMENT GQA Bwd TFLOPS on NVIDIA A800 GPU

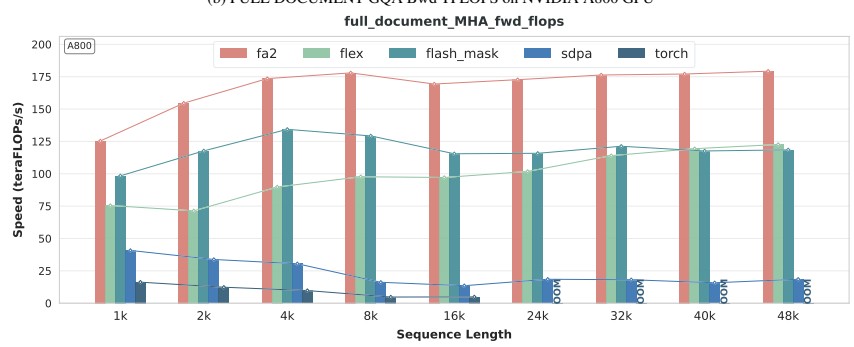

(c) FULL DOCUMENT MHA Fwd TFLOPS on NVIDIA A800 GPU

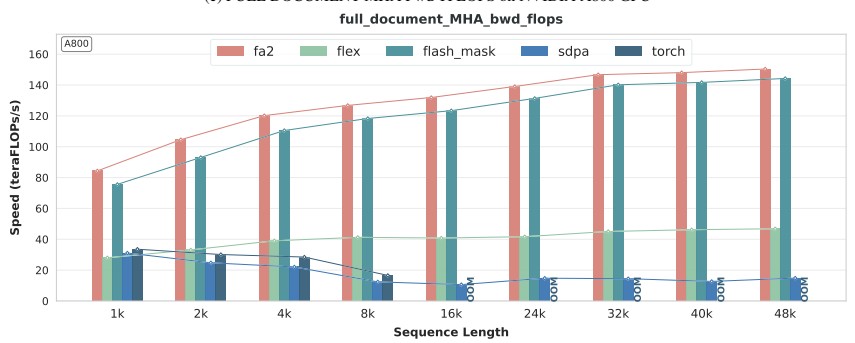

(d) FULL DOCUMENT MHA Bwd TFLOPS on NVIDIA A800 GPU

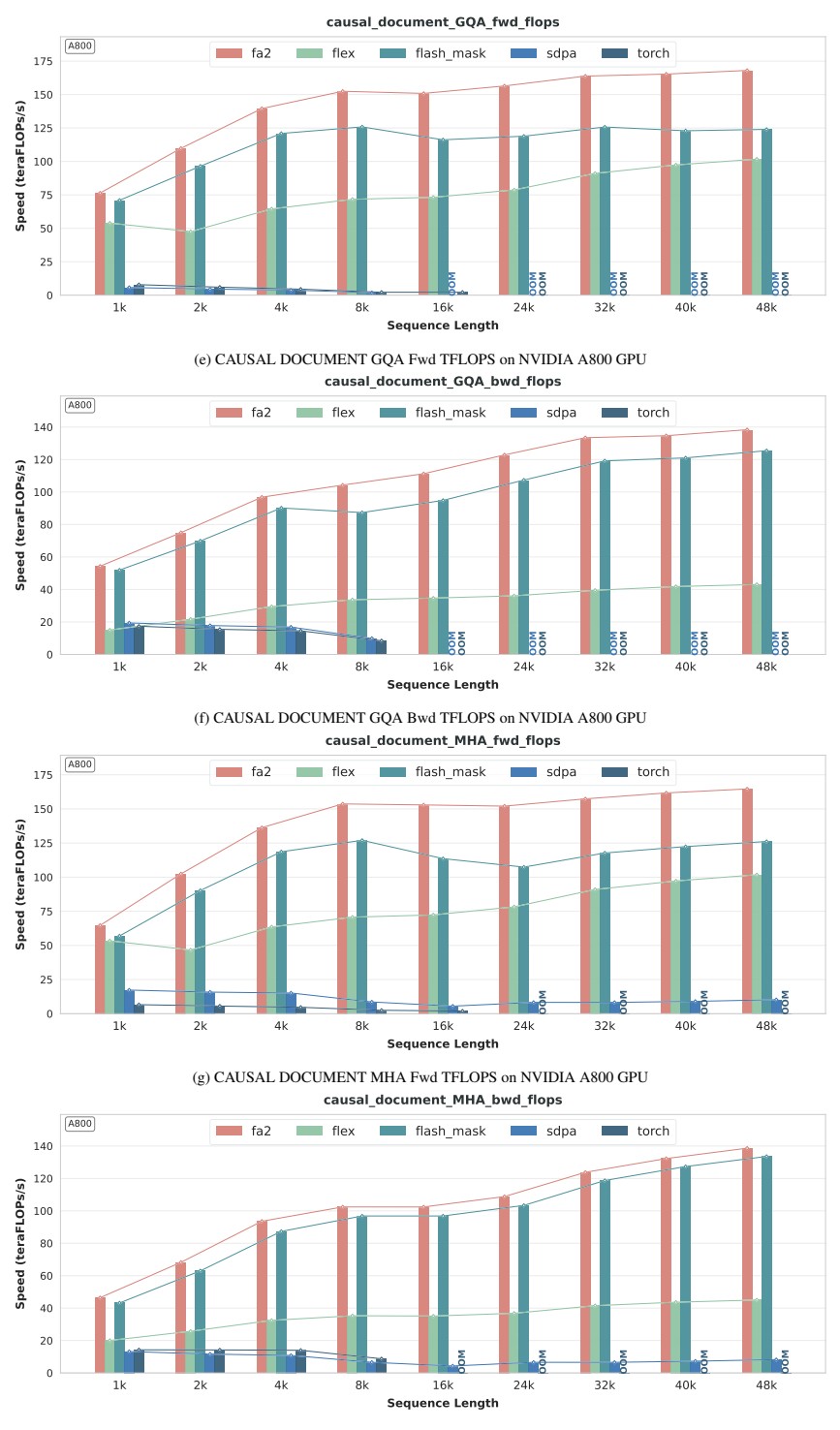

(e) CAUSAL DOCUMENT GQA Fwd TFLOPS on NVIDIA A800 GPU

(f) CAUSAL DOCUMENT GQA Bwd TFLOPS on NVIDIA A800 GPU

(g) CAUSAL DOCUMENT MHA Fwd TFLOPS on NVIDIA A800 GPU

(h) CAUSAL DOCUMENT MHA Bwd TFLOPS on NVIDIA A800 GPU

Figure 26: TFLOPS of FULL/CAUSAL DOCUMENT on NVIDIA A800 GPU

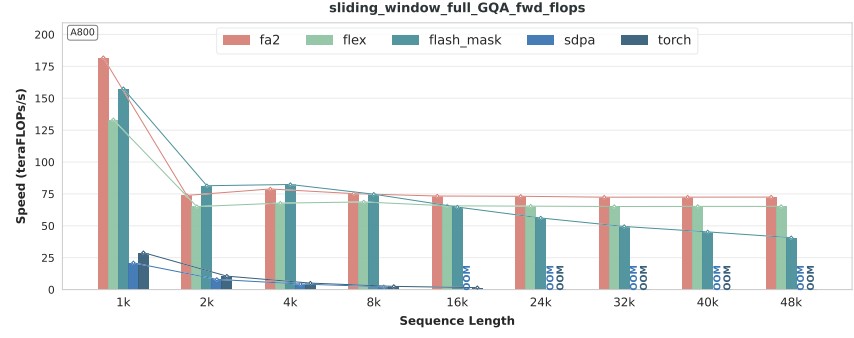

(a) FULL SLIDING WINDOW GQA Fwd TFLOPS on NVIDIA A800 GPU

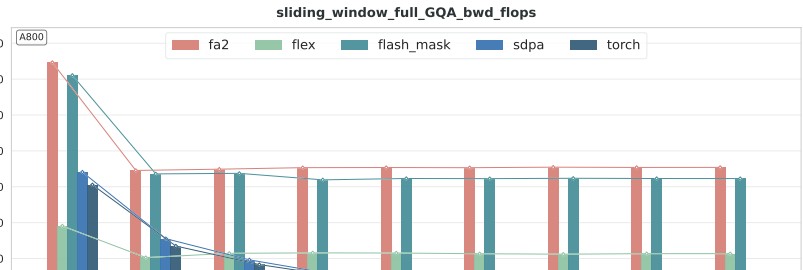

(b) FULL SLIDING WINDOW GQA Bwd TFLOPS on NVIDIA A800 GPU

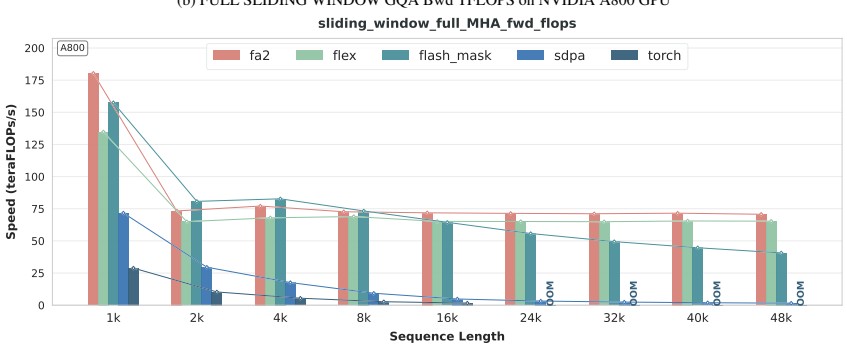

(c) FULL SLIDING WINDOW MHA Fwd TFLOPS on NVIDIA A800 GPU

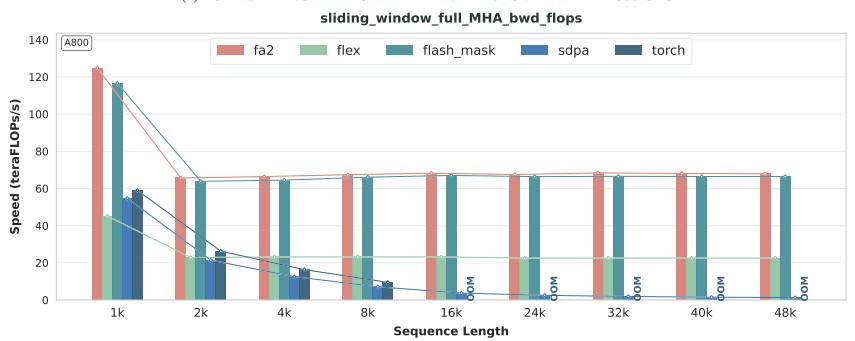

(d) FULL SLIDING WINDOW MHA Bwd TFLOPS on NVIDIA A800 GPU

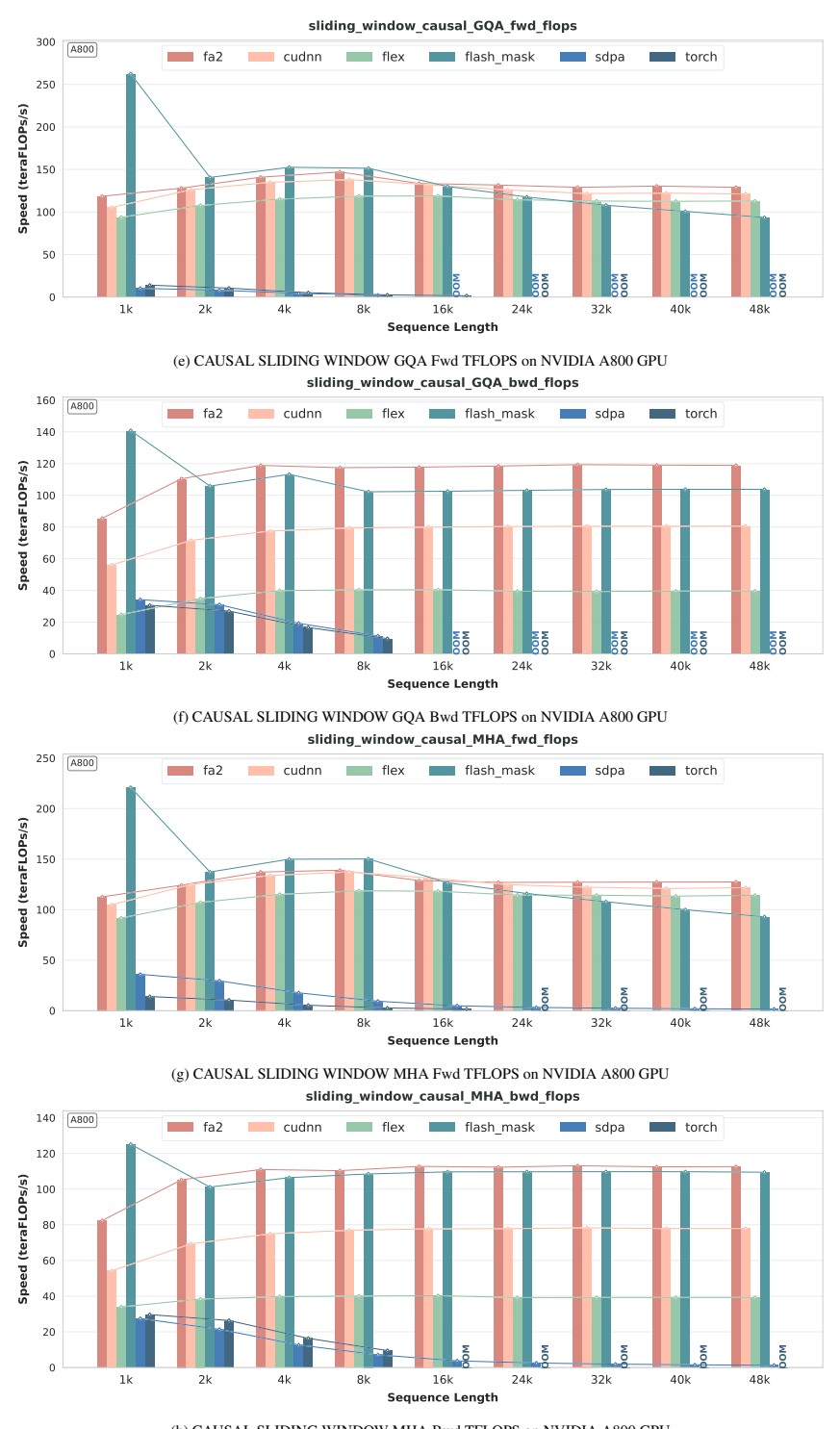

(e) CAUSAL SLIDING WINDOW GQA Fwd TFLOPS on NVIDIA A800 GPU

(f) CAUSAL SLIDING WINDOW GQA Bwd TFLOPS on NVIDIA A800 GPU

(g) CAUSAL SLIDING WINDOW MHA Fwd TFLOPS on NVIDIA A800 GPU

(h) CAUSAL SLIDING WINDOW MHA Bwd TFLOPS on NVIDIA A800 GPU

Figure 27: TFLOPS of FULL/CAUSAL SLIDING WINDOW on NVIDIA A800 GPU

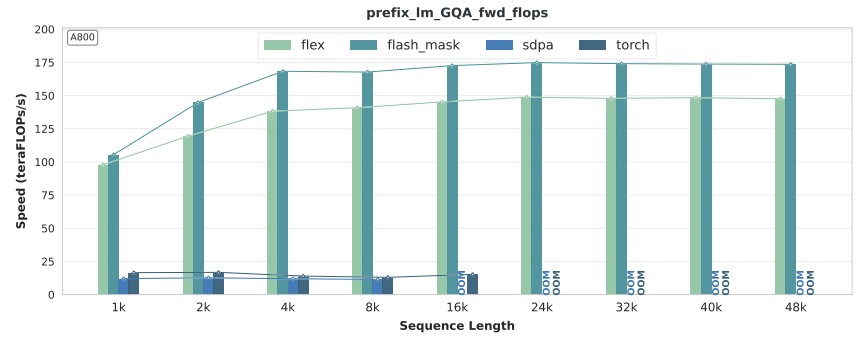

(a) PREFIX LM GQA Fwd TFLOPS on NVIDIA A800 GPU

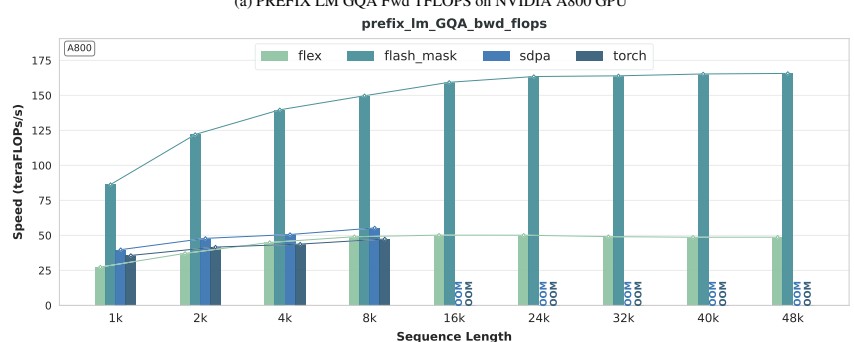

(b) PREFIX LM GQA Bwd TFLOPS on NVIDIA A800 GPU

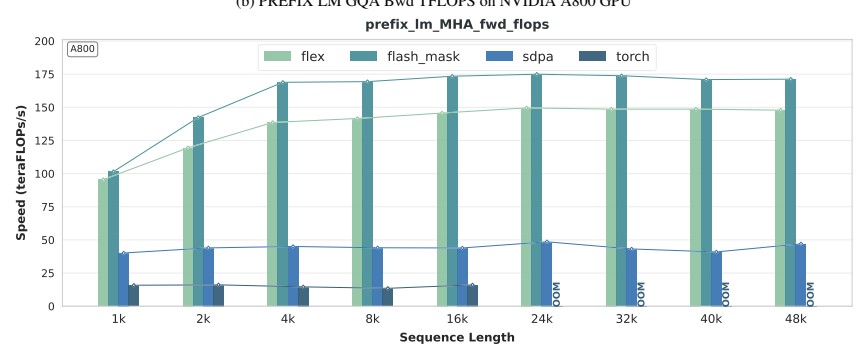

(c) PREFIX LM MHA Fwd TFLOPS on NVIDIA A800 GPU

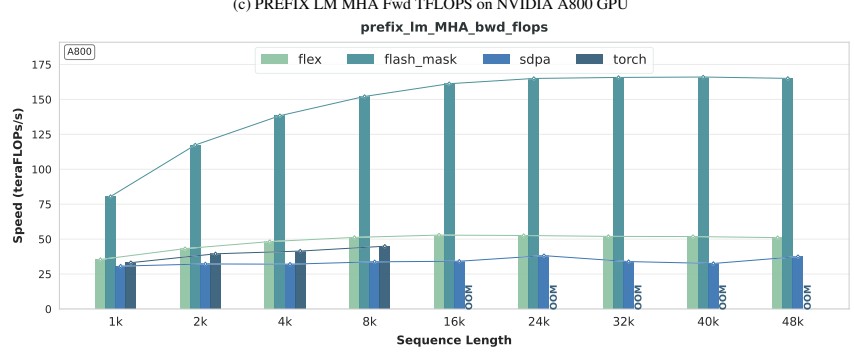

(d) PREFIX LM MHA Bwd TFLOPS on NVIDIA A800 GPU

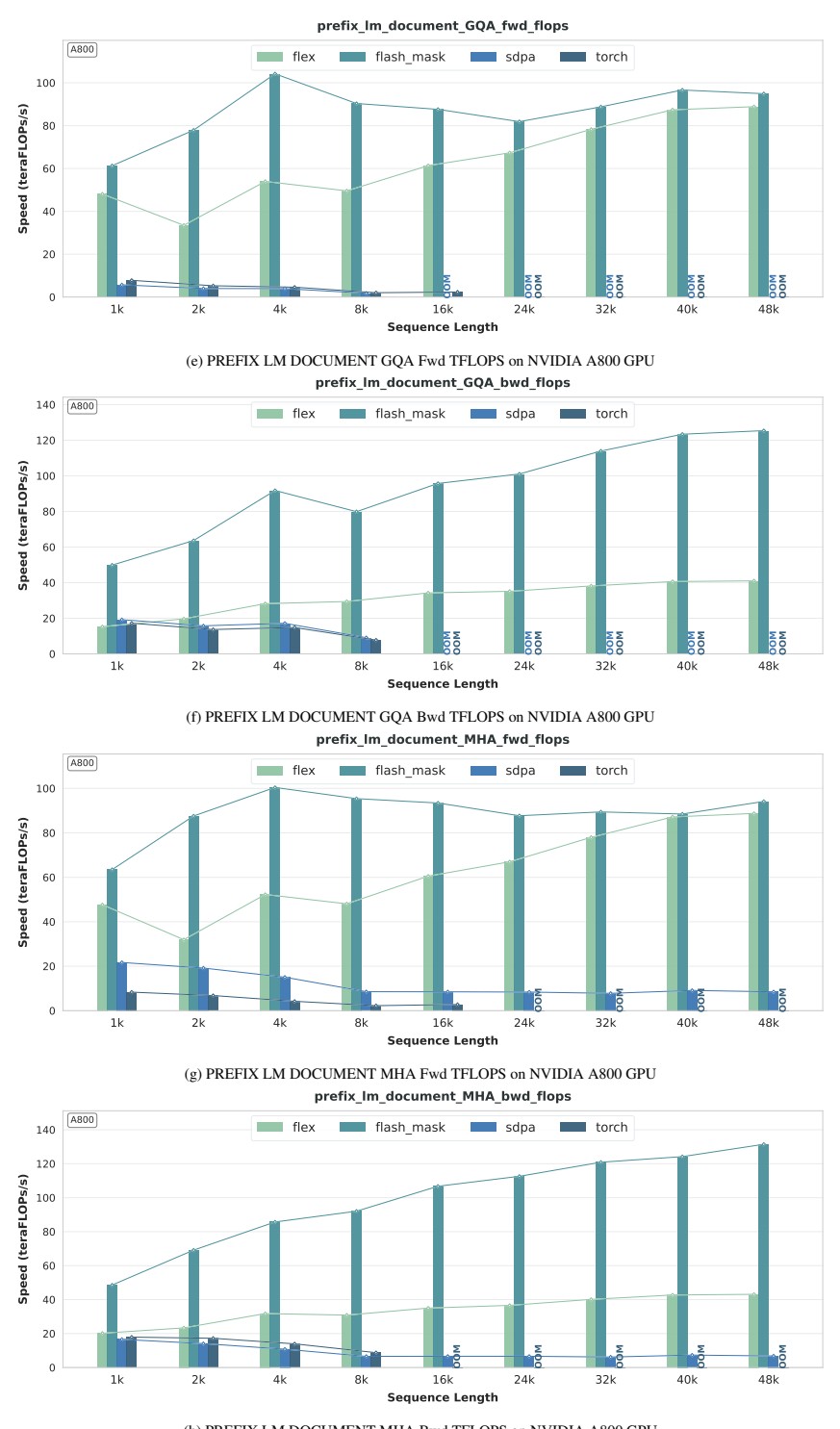

(e) PREFIX LM DOCUMENT GQA Fwd TFLOPS on NVIDIA A800 GPU

(f) PREFIX LM DOCUMENT GQA Bwd TFLOPS on NVIDIA A800 GPU

(g) PREFIX LM DOCUMENT MHA Fwd TFLOPS on NVIDIA A800 GPU

(h) PREFIX LM DOCUMENT MHA Bwd TFLOPS on NVIDIA A800 GPU

Figure 28: TFLOPS of PREFIX LM and PREFIX LM DOCUMENT on NVIDIA A800 GPU

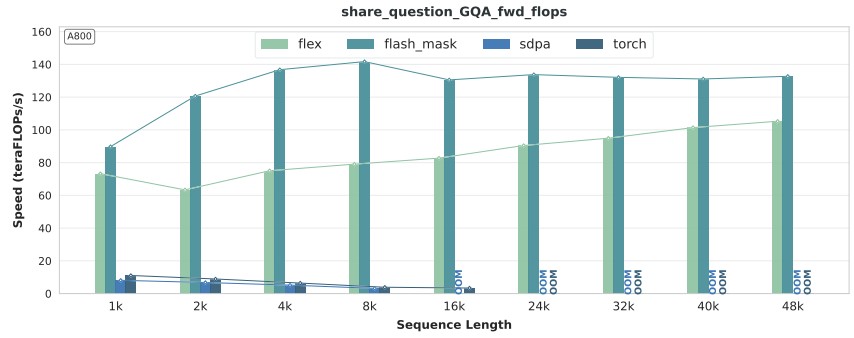

(a) SHARE QUESTION GQA Fwd TFLOPS on NVIDIA A800 GPU

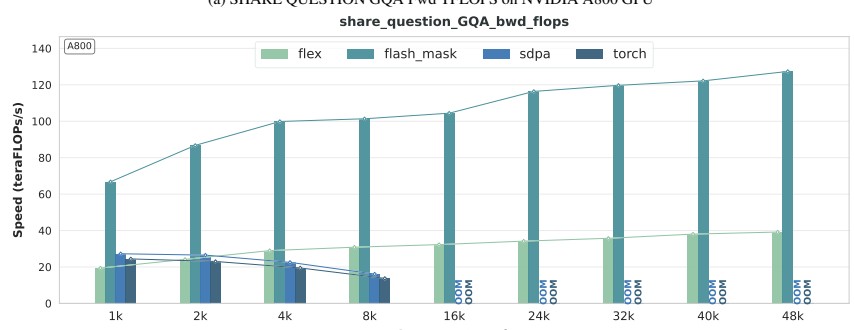

(b) SHARE QUESTION GQA Bwd TFLOPS on NVIDIA A800 GPU

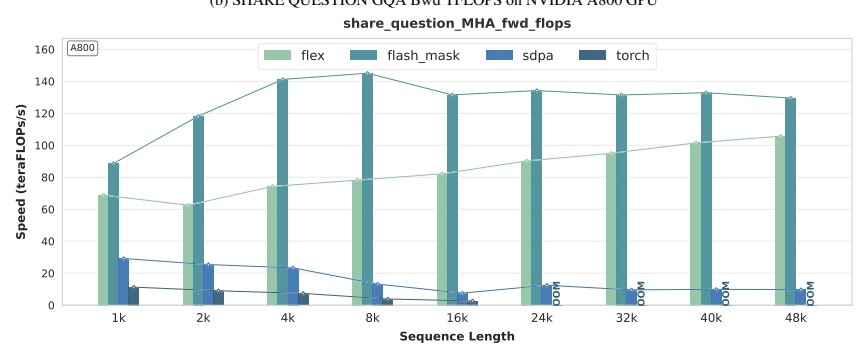

(c) SHARE QUESTION MHA Fwd TFLOPS on NVIDIA A800 GPU

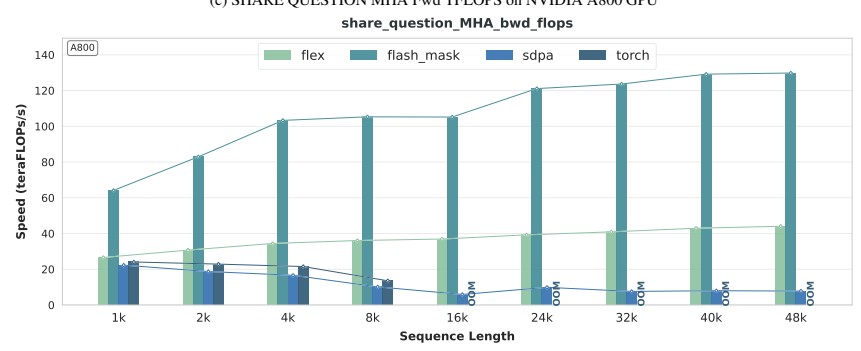

(d) SHARE QUESTION MHA Bwd TFLOPS on NVIDIA A800 GPU

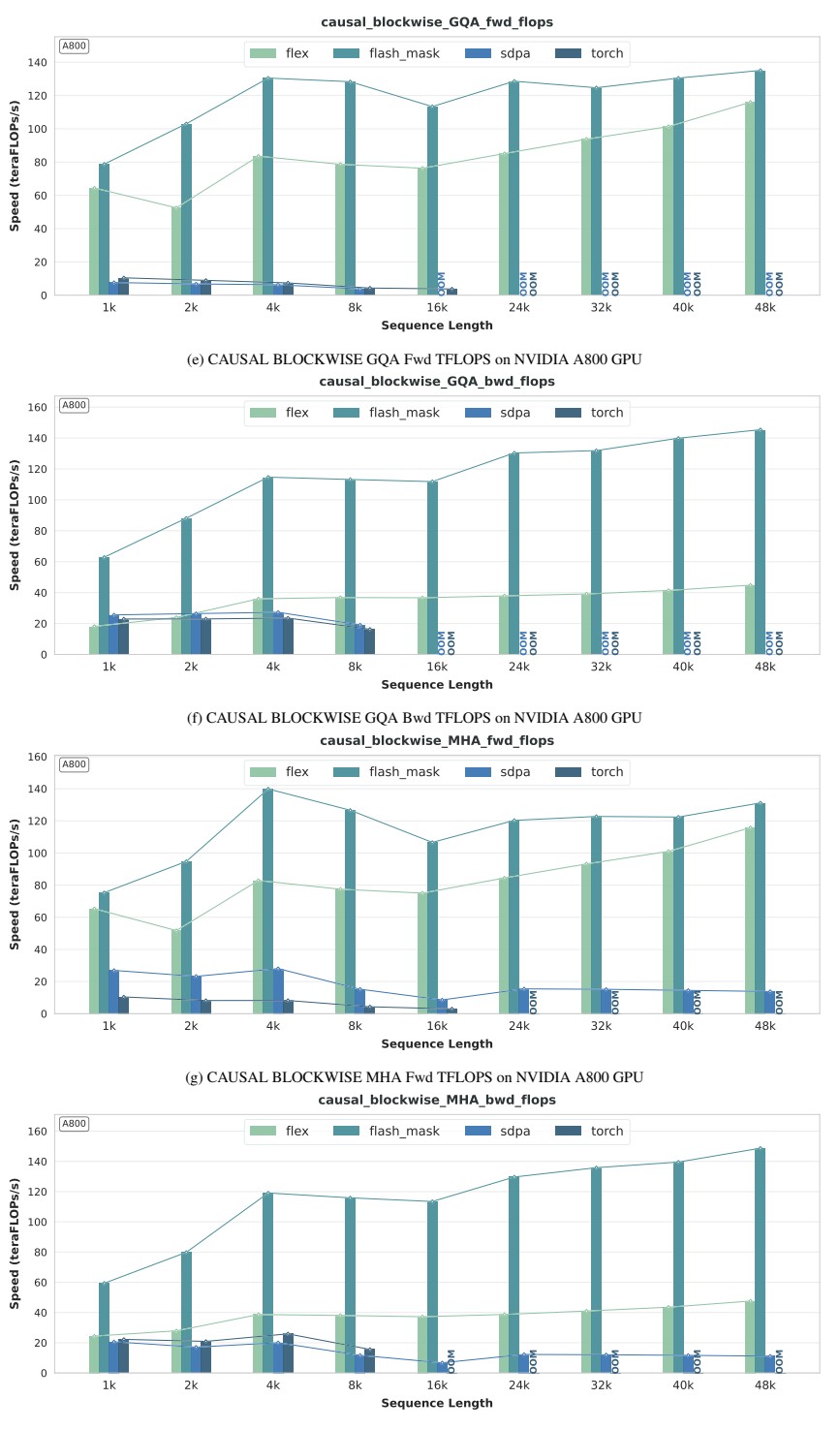

(e) CAUSAL BLOCKWISE GQA Fwd TFLOPS on NVIDIA A800 GPU

(f) CAUSAL BLOCKWISE GQA Bwd TFLOPS on NVIDIA A800 GPU

(g) CAUSAL BLOCKWISE MHA Fwd TFLOPS on NVIDIA A800 GPU

(h) CAUSAL BLOCKWISE MHA Bwd TFLOPS on NVIDIA A800 GPU

Figure 29: TFLOPS of SHARE QUESTION and CAUSAL BLOCKWISE on NVIDIA A800 GPU

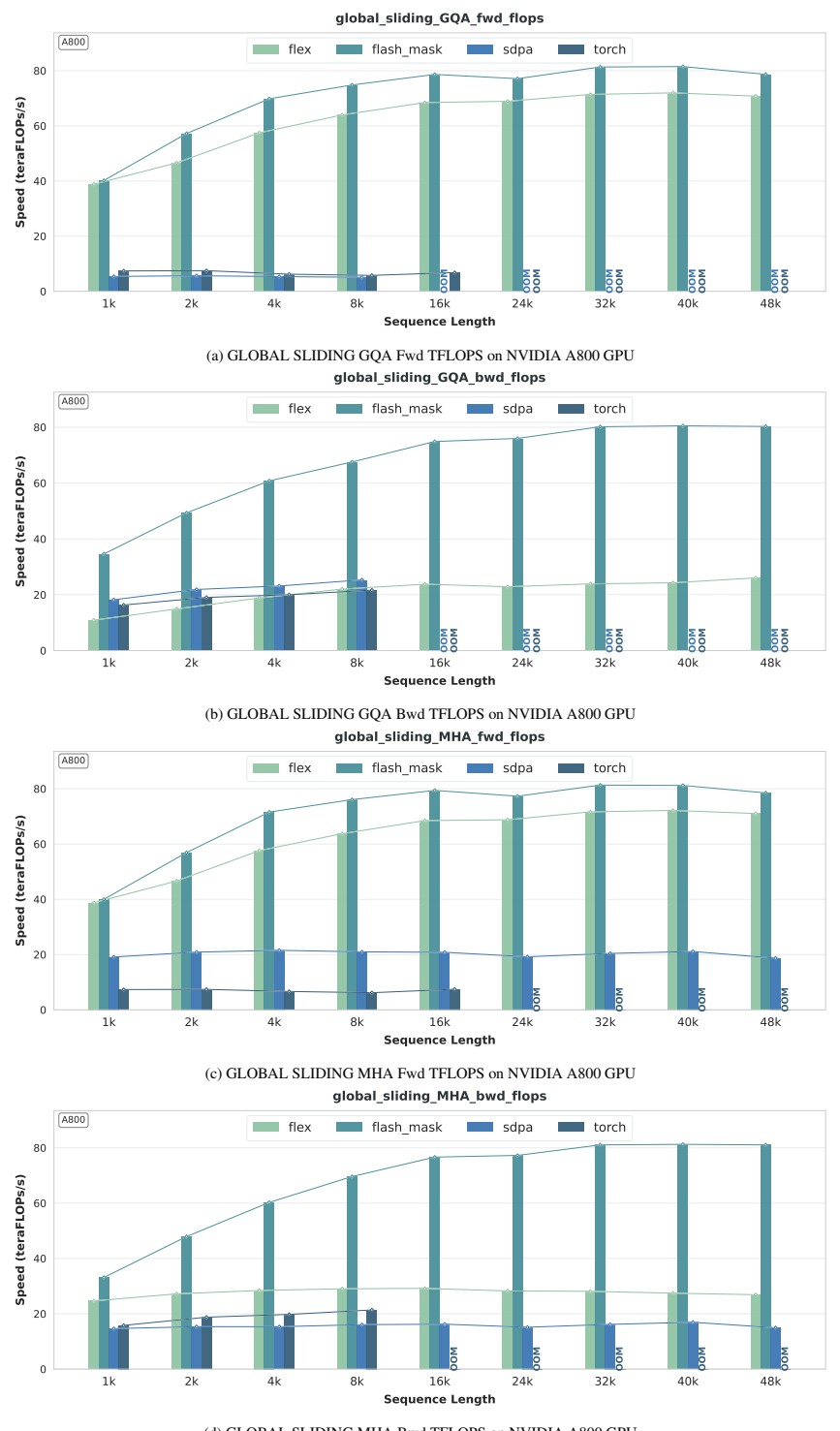

(a) GLOBAL SLIDING GQA Fwd TFLOPS on NVIDIA A800 GPU

(b) GLOBAL SLIDING GQA Bwd TFLOPS on NVIDIA A800 GPU

(c) GLOBAL SLIDING MHA Fwd TFLOPS on NVIDIA A800 GPU

(d) GLOBAL SLIDING MHA Bwd TFLOPS on NVIDIA A800 GPU

Figure 30: TFLOPS of GLOBAL SLIDING on NVIDIA A800 GPU

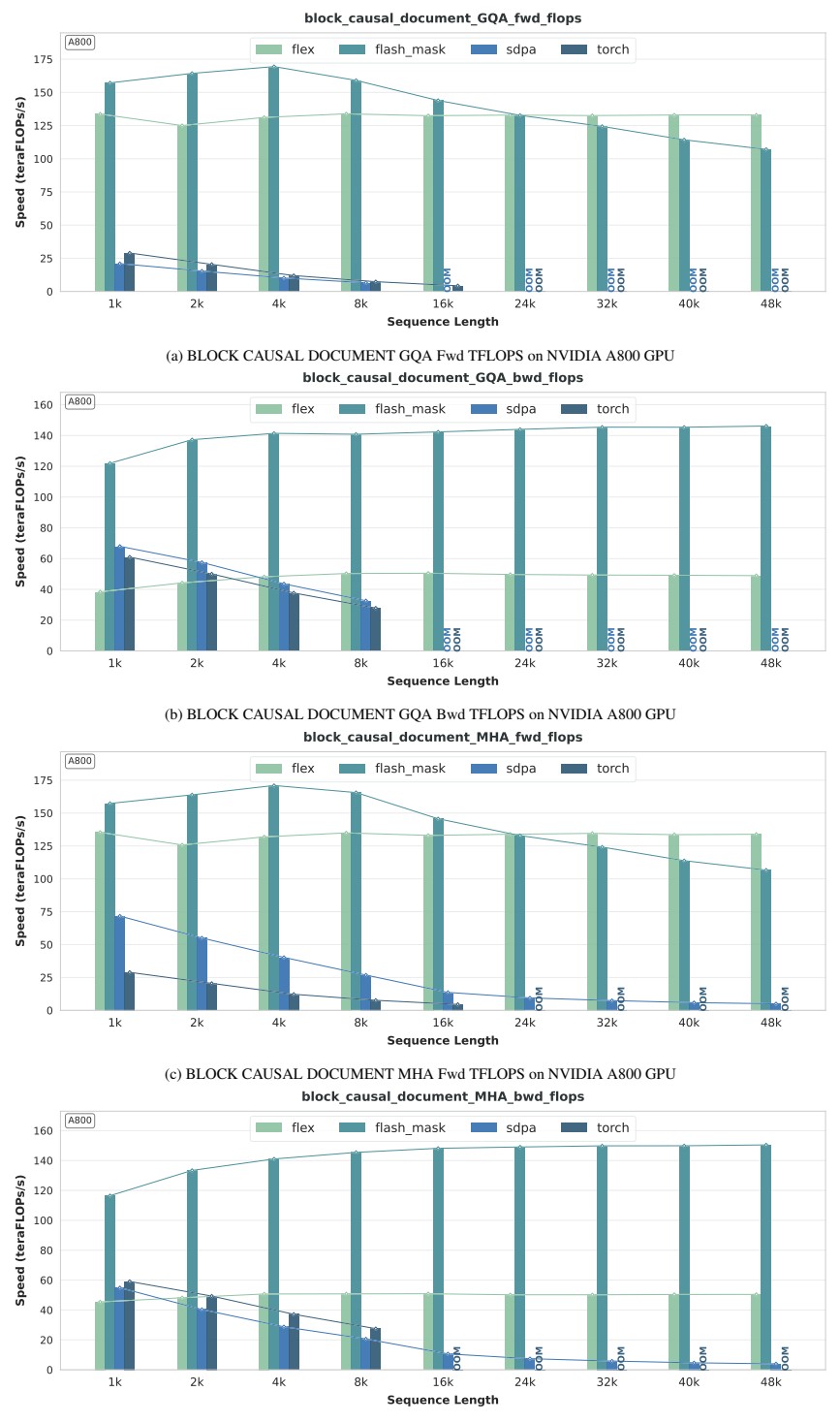

(a) BLOCK CAUSAL DOCUMENT GQA Fwd TFLOPS on NVIDIA A800 GPU

(b) BLOCK CAUSAL DOCUMENT GQA Bwd TFLOPS on NVIDIA A800 GPU

(c) BLOCK CAUSAL DOCUMENT MHA Fwd TFLOPS on NVIDIA A800 GPU

(d) BLOCK CAUSAL DOCUMENT MHA Bwd TFLOPS on NVIDIA A800 GPU

Figure 31: TFLOPS of BLOCK CAUSAL DOCUMENT on NVIDIA A800 GPU

