# OpenReview forum: "Long-Context Attention Benchmark: From Kernel Efficiency to Distributed Context Parallelism"
_ICLR.cc/2026/Conference — ICLR 2026 Poster_

### Official Review · Reviewer_1Et8 · 2025-10-20

**Soundness:** 3
**Presentation:** 3
**Contribution:** 2
**Rating:** 4
**Confidence:** 4

**Summary:**

The paper presents LongCA-Bench, a unified benchmark framework designed to evaluate long-context attention mechanisms across both single-device and distributed settings. The author integrates existing attention kernels (dense and sparse) and distributed context-parallel strategies into a modular, extensible evaluation platform. Through extensive experiments on up to 96 GPUs, the benchmark compares efficiency, scalability, and memory usage across diverse mask patterns, sequence lengths, and distributed configurations of existing methods. The work provides a valuable empirical reference for practitioners and researchers working on long-context training but does not seem to introduce novel attention algorithms or distributed mechanisms beyond benchmarking existing approaches.

**Strengths:**

1. **Comprehensive benchmarking.** The paper systematically benchmarks a wide range of attention implementations, including dense, sparse, and distributed mechanisms, under a unified framework. The experimental coverage (up to 96 GPUs and multiple mask types) is extensive and provides a clear view of current attention efficiency trends.
2. **Sound experimental methodology.** The experiments are well-organized, use realistic settings (e.g., long context lengths, different mask patterns), and report consistent metrics (throughput, memory). The methodology is reproducible and technically competent.
3. **Useful empirical reference.** The benchmark results could serve as a useful reference for practitioners and researchers seeking guidance on the performance trade-offs of existing attention kernels and distributed strategies.

**Weaknesses:**

1. **Limited Analysis.** The paper reports extensive throughput and memory results, but offers limited discussion on the underlying causes of observed performance trends of the benchmarked methods.
2. **Incomplete coverage of the most critical setting — distributed sparse attention.** The integration of sparse attention (particularly dynamic block-sparse attention with TopK/TopP selection criterion) into distributed contexts remains an unexplored and practically important challenge. The paper does not explore this aspect (does not touch block sparse distributed attention, only discusses full/causal/document), which somewhat limits its impact given the stated motivation of benchmarking “long-context attention.”

**Questions:**

1. How can the proposed framework offer insight on sparse attention in distributed settings? Typically, how should we overlap communication with computation, and address the load balancing problem of arbitrary block sparse pattern? The author does not need to provide a complete solution to this problem, and what I would like to see is how this work could contribute to future research on these challenges.
2. Could the authors provide more analysis to understand the trends observed in kernel or distributed scaling performance?

---

> ### Author Response · Authors · 2025-11-20
>
> We thank the reviewer's useful review of our work. Our responses to the weaknesses and questions you raised are as follows:
>
> ### **W1 & Q2: Limited Analysis.**
>
> Thank you for raising this point. We provide detailed analyses, as well as the corresponding experiments.
>
> - Analysis of Context Parallel Attention
>
> For context parallel attentions, in out original submission (Section 3.3 and Appendix 8), we provided a detailed analysis of distributed strategy, engineering details, and computation patterns. Based on these considerations, we analyzed the underlying reasons for the performance of each mechanism and examined why their behaviors vary across different scenarios.
>
> 1. Fully leveraging the multi-head attention structure yields significant performance improvements. In context parallelism, exploiting head-level parallelism enables more efficient computation utilization.
>
> 2. Distributed attention efficiency is fundamentally determined by communication–computation overlap. For example, mechanisms like Ring P2P are highly sensitive to variations in distributed environments: waiting in per-GPU computation, variable-length inputs, or inter-node link performance can significantly degrade overall throughput.
>
> 3. Context parallelism strategies are more prone to causing CPU stalls and CUDA synchronization waits, making engineering optimizations tailored to specific strategies and scenarios crucial.
>
> We later recognized that the original submission lacked a thorough theoretical and empirical analysis of distributed communication across different mechanisms. To address this, we have added a detailed communication analysis, including the exact communication volume of each method, the specific communication operators invoked, and the compute–communication overlap behavior. We further supplemented our work with benchmark results of communication operators in both single-node and multi-node settings to offer more direct empirical reference.
>
> - Ulysess:
>   - FWD: 2 All2All for Q and O, 2 All2All for K and V, the communication volume is $$ \frac{N-1}{N^2}th_{kv}d*2 + \frac{N-1}{N^2}th_{q}d * 2$$
>   - BWD: 2 All2All for dQ and dO, 2 All2All for dK and dV, the communication volume is $$\frac{N-1}{N^2}th_{kv}d*2 + \frac{N-1}{N^2}th_{q}d * 2$$
>   - Overlap: All communications are exposed outside the computation.
> - Ring-AllGather
>   - FWD: 2 AllGather for K and V, the communication volume is $$\frac{N-1}{N}th_{kv}d*2$$
>   - BWD: 2 AllGather for K and V, 2 ReduceScatter for dK and dV, the communication volume is $$\frac{N - 1}{N}th_{kv} d*2 + \frac{N-1}{N}th_{kv}d*2$$
>   - Overlap: All communications are exposed outside the computation.
> - Ring-P2P
>   - FWD: N-1 P2P for K and V, the communication volume is $$\frac{N-1}{N}th_{kv}d*2$$
>   - BWD: N-1 P2P for K and V, N-1 P2P dK and dV, the communication volume is $$\frac{N-1}{N}th_{kv}d*2 + \frac{N-1}{N}th_{kv}d*2$$
>   - Overlap: In theory, the FULL mask scenario achieves communication-computation overlap. Focusing solely on mask-sparse scenarios, computation and communication start simultaneously at each stage across different ranks, with the communication volume remaining unchanged. The portion of the entire pipeline’s communication that can be overlapped is approximately $\frac{\sum_i^mt_m^2/flops}{t/bandwidth}*k$, where t stands for the total sequence length, $t_m^2$ denotes the single computation area, and k merely represents a correction coefficient that exists as a constant for specific settings.
> - USP, we define Ulysses size = Na, ring-p2p size = Np, where Np×Na=N.
>   - FWD: 2 All2All for Q and O, 2 All2All for K and V, Np -1 P2P for K and V, the communication volume is $$\frac{N_a-1}{N_p*N_a^2}th_{kv}d*2 + \frac{N_a-1}{N_p*N_a^2}th_{q}d*2 + \frac{N_p-1}{N_p*N_a}th_{kv}d*2$$
>   - BWD: 2 All2All for dQ and dO, 2 All2All for dK and dV, Np -1 P2P for K and V, Np -1 P2P for dK and dV, the communication volume is $$\frac{N_a-1}{N_p*N_a^2}th_{kv}d*2 + \frac{N_a-1}{N_p*N_a^2}th_{q}d*2 + \frac{N_p-1}{N_p*N_a}th_{kv}d*2 + \frac{N_p-1}{N_p*N_a}th_{kv}d*2$$
>   - Overlap: the analysis remains the same as Ulysses and Ring P2P.

---

> ### Author Response · Authors · 2025-11-20
>
> ### **W1 & Q2: Limited Analysis contd.**
>
> - LoongTrain, we define intra_ring size = n, window number = k, where n×k=Np.
>   - FWD: 2 All2All for Q and O, 2 All2All for K and V, (n-1)*k P2P for intra K and V, k-1 P2P for inter K and V, the communication volume is $$\frac{N_a-1}{N_p*N_a^2}th_{kv}d*2 + \frac{N_a-1}{N_p*N_a^2}th_{q}d*2 + \frac{(n-1)*k}{N_p*N_a}th_{kv}d*2 + \frac{k-1}{N_p*N_a}th_{kv}d*2$$
>   - BWD: 2 All2All for dQ and dO, 2 All2All for dK and dV, (n-1)*k P2P for intra K and V, k-1 P2P for inter K and V, n*k P2P for intra dK and dV, k P2P for inter dK and dV, the communication volume is $$\frac{N_a-1}{N_p*N_a^2}th_{kv}d*2 + \frac{N_a-1}{N_p*N_a^2}th_{q}d*2 + \frac{(n-1)*k}{N_p*N_a}th_{kv}d*2 + \frac{k-1}{N_p*N_a}th_{kv}d*2 + \frac{1}{N_a}th_{kv}d*2 + \frac{k}{N_p*N_a}th_{kv}d*2$$
>   - Overlap: The forward is consistent with USP. For the backward, the intra and inter communications of K and V are the same as in USP. However, regarding intra dK and dV, unlike USP, which places intra K, V, dK, and dV in a single buffer, LoongTrain separates K, V from dK, dV into two buffers, requiring two separate P2P communications. Thus, in each intra stage of LoongTrain, the intra dK and dV communications are asynchronously initiated only after the completion of computations, overlapping with the correction of dK and dV in the next stage. That said, LoongTrain incurs one additional intra communication for dK and dV, leaving approximately one P2P communication exposed. After the final intra dK and dV communication concludes, LoongTrain initiates an inter dK and dV communication, which overlaps with the subsequent intra computations. However, the dK and dV communication for the last window remains exposed.
>
> We conducted additional experiments to measure communication bandwidth utilization for distributed attention mechanisms in LongCA-Bench, using NVIDIA H100 80GB HBM3 devices (with theoretical peak bandwidth of 450 GB/s for intra-node communication and 50 GB/s for inter-node communication). In these experiments, we recorded the actual communication time and data volume for each communication operator and calculated the effective speed of each communication. Furthermore, for the GQA scenario, we distinguished between Q/O and K/V communications. Finally, we computed both the average and maximum transfer speeds for each type of communication. It is worth noting that when both intra-node and inter-node communications of the same type occurred simultaneously, we reported the average speed by combining the intra-node and inter-node measurements, while additionally listing their respective maximum speeds separately.
>
> - 8 GPUs on a single machine
>
> | Speed (Mean, Max)                | All2All Q,O(dQ,dO) | All2All K,V(dK,dV) | AllGath K,V | ReduceScatter dK,dV | Intra P2P K,V(dK,dV) | Inter P2P K,V(dK,dV) |
> |----------------------------------|--------------------|--------------------|-------------|---------------------|----------------------|----------------------|
> | Ulysess                          | (296,323)          | (221,229)          |             |                     |                      |                      |
> | Ring-AllGather                   |                    |                    | (266,314)   | (317,319)           |                      |                      |
> | RingP2P                          |                    |                    |             |                     | (182,327)            |                      |
> | USP(a2a=4, p2p=2)                | (78,282)           | (161,189)          |             |                     | (289,298)            |                      |
> | LoongTrain(a2a=2,p2p=2,window_num=2) | (159,236)      | (110,167)          |             |                     | (270,296)            |                      |
>
> - 16 GPUs across 2 machines
>
> | Speed (Mean, Max)                | All2All Q,O(dQ,dO) | All2All K,V(dK,dV) | AllGather K,V | ReduceScatter dK,dV | Intra P2P K,V(dK,dV) | Inter P2P K,V(dK,dV) |
> |----------------------------------|--------------------|--------------------|---------------|---------------------|----------------------|----------------------|
> | Ulysess                          | (44,63)            | (50,59)            |               |                     |                      |                      |
> | Ring-AllGather                   |                    |                    | (166,229)     | (65,247)            |                      |                      |
> | RingP2P                          |                    |                    |               |                     | (25,293)             | (25,26)              |
> | USP(a2a=8, p2p=2)                | (93,280)           | (176,194)          |               |                     |                      | (17,26)              |
> | LoongTrain(a2a=4,p2p=2,window_num=2) | (76,286)      | (141,181)          |               |                     | (40,297)             | (40,26)              |

---

> ### Author Response · Authors · 2025-11-20
>
> ### **W1 & Q2: Limited Analysis contd.**
>
> - Analysis of Sparse Attention Kernels
>
> For sparse attention kernels, in our original manuscript (Section 3.2 and Appendix 6), we provided a detailed analysis of six existing block sparse attention implementations.
>
> 1. Scalability and Training Support: The TFLOPS performance of many kernels deteriorates markedly as context length increases (e.g., VSA). Moreover, several libraries (e.g., FlashInfer) lack backward-pass support, limiting their usefulness during training.
> 2. Hardware Utilization: Hardware-specific kernels demonstrate that targeted optimizations (e.g., using WGMMA and TMA on Hopper GPUs) are crucial for high performance. Generic implementations frequently fail to fully exploit these architectural features.
>
> 3. Flexibility vs. Efficiency: General kernels (e.g., FlexAttention) suffer from high memory overhead because of block-mask metadata, while specialized kernels achieve high speed but are brittle, lacking support for mechanisms like GQA or heterogeneous block sizes (e.g., VSA, FA2 Sparse).
>
> Moreover, in light of the growing prevalence of complex, trainable sparse attention mechanisms, such as Native Sparse Attention (NSA) [1] and Deepseek Sparse Attention (DSA) [2] (used in the recent DeepSeek-V3.2-Exp), we conduct supplementary performance experiments for NSA and DSA and extend the analysis by comparing these eight sparse attention kernels in the backward setting. Also, we find that these advanced, trainable methods can be effectively benchmarked within our block sparse attention framework because they all treat sparse attention modeling with uniform block masks. The evaluation and analysis are also added to our revised version. The experimental results of NSA and DSA are posted in Table 10 and Table 11 in our revised version.
>
> - Experimental Settings
>
> For NSA, we follow the setting from the original paper, with 64 query heads and 4 KV heads. In this configuration, each query group (comprising 16 query heads) selects the same KV indices. Each query selects 32 K-blocks with a block size of 64, resulting in a total of 2048 selected tokens. Since the selection attention module dominates the end-to-end overhead and aligns with our block sparse framework, our experiments focus on this module. This configuration maps to q_block_size=16 and k_block_size=64 within our block sparse attention benchmark framework. We conduct both NSA-Triton and NSA-Tilelang in the benchmark, implemented by Triton and Tilelang, respectively.
>
> For DSA, since it utilizes token-level sparse attention and is designed based on DeepseekMLA model architecture, we set the configuration to 128 query heads and 1 KV head, matching DeepseekMLA. Each query group therefore consists of 128 query heads sharing the same KV indices. We set the top-k selection size to 2048 tokens. In our framework, this corresponds to q_block_size=128 and k_block_size=1.  We benchmark this setup using the DSA-Tilelang and DSA-FlashMLA, implemented by Tilelang and CUDA, respectively.
>
> Integrating the analysis of the six block sparse attention mechanisms detailed in our submission with the two representative trainable implementations (NSA and DSA) discussed above, we derive the following critical insights:
>
> 1. Importance of Algorithm-Hardware Co-Design and Block Size Configuration. The block size configuration fundamentally dictates kernel performance and implementation strategy, necessitating a co-design approach between the algorithmic/model design and kernel optimization. NSA employs a query block size of 16. However, the WGMMA (Warp-Group Matrix-Matrix Accumulation) instruction on Hopper GPUs requires a minimum M-dimension of 64. As a result, computations must be padded to 64, resulting in significant computational waste. This aligns with our benchmark results, where both NSA-Triton and NSA-Tilelang achieve relatively lower performance (approx. 100 TFLOPS). Instead, DSA leverages MQA mode in MLA to form large query groups, ensuring the WGMMA M-dimension requirement ($\ge$64) is met. However, for its k_block_size=1 setting, standard TMA loading is inefficient due to low bandwidth utilization for sparse, gather-style access. Instead, the optimal kernel implementation utilizes cp.async to gather sparse K/V tokens into dense tiles in shared memory. In contrast, VSA maintains both Q and K block sizes at 64, allowing it to fully utilize TMA for efficient loading. Moreover, VSA and DSA-FlashMLA, which use hardware-specific implementations in Hopper GPUs, demonstrate significant greater performance compared to their comparators, necessitating hardware utilization to be considered in block sparse attention kernels. To conclude, high-performance sparse attention cannot rely on generic kernels alone. It requires algorithmic decisions—such as DSA’s use of MQA to group queries or VSA’s fixed block selection—that align with hardware constraints (e.g., WGMMA and TMA efficiency).

---

> ### Author Response · Authors · 2025-11-20
>
> ### **W1 & Q2: Limited Analysis contd.**
>
> 2. Challenges and Optimization Opportunities in Backward Kernels. There is a lack of backward functionality and significant performance gap between forward and backward passes, highlighting the need for specialized backward implementations and engineering tuning to facilitate the adoption of complex trainable sparse attention. While forward passes for mechanisms like VSA and DSA achieve 500–600 TFLOPS, backward performance lags significantly. Currently, only VSA (and VSA-Triton) reach approx. 300 TFLOPS, while DSA-Tilelang reaches approx. 100 TFLOPS.  Regarding of the kernel implementations, for scenarios with larger k_block_size (e.g., 64), the overall backward computation logic remains similar to dense attention kernels. However, the fact that the native VSA kernel falls behind its Triton counterpart at context lengths exceeding 64K indicates that further low-level engineering tuning is required to sustain performance at scale. For scenarios with small k_block_size (e.g., 1), the kernel strategy requires a fundamental structural shift. The standard backward loop order (Outer Loop over K/V, Inner Loop over Q) must be inverted to an Outer Loop over Q and Inner Loop over K/V to efficiently handle the fine-grained sparsity. Ultimately, bridging the significant performance gap between forward and backward passes is a prerequisite for the efficient training of complex sparse attention models. This necessitates a shift from generic implementations to specialized backward kernels that apply distinct optimization strategies—ranging from engineering fine-tuning for large blocks to fundamental kernel refactoring for small blocks—tailored to the specific granularity of each method.
>
> - Analysis of Dense Attention Kernels
>
> For dense attention kernels, in our original submission (Section 3.1 and Appendix 6), we provided a detailed analysis of the specific performance of each kernel in different scenarios and the underlying reasons.
>
> 1. The performance of dense kernels largely depends on the computation-to-memory ratio. Therefore, in most scenarios, as the sequence length scales, the kernel’s performance tends to improve. However, when the kernel becomes compute-bound, its performance generally plateaus.
>
> 2. Kernels optimized through compiler techniques often exhibit performance variability. Different compilation orders and specific hardware resources can significantly affect the kernel’s actual efficiency.
>
> Insight of algorithm-hardware co-design, we add experiments about dense kernels on A800 GPU and present the results in Appendix 9.
>
> Reference
>
> [1] Native Sparse Attention: Hardware-Aligned and Natively Trainable Sparse Attention. https://arxiv.org/abs/2502.11089
>
> [2] https://github.com/deepseek-ai/DeepSeek-V3.2-Exp

---

> ### Author Response · Authors · 2025-11-20
>
> ### **W2 & Q1 Distributed Sparse Attention.**
>
> Thank you for raising this point. As you correctly pointed out, our current paper primarily focuses on distributed benchmarking for full attention, causal attention, and document-based attention patterns. We did not include distributed sparse attention in the current work primarily because it introduces a series of unique, non-trivial challenges. We believe these challenges warrant treating distributed sparse attention as a a separate, large-scale research problem.
>
> - Core Challenges:
>
> Based on our analysis, extending dynamic sparse attention mechanisms to a distributed environment faces several core challenges:
>
> 1. Computation Imbalance:
>
> In sparse scenarios, especially dynamic sparse settings, the attention mask may differ for every head at every step, making it difficult to balance the computational load across workers. This issue is especially severe in TopP scenarios, where the sparsity ratios vary drastically across different heads. This leads to severe inter-head load imbalance, as the final execution time of a worker is bottlenecked by the head with the heaviest computational load.
>
> 2. Redundant KV Communication:
>
> In block-sparse scenarios, the KV blocks actually required by each Q block are often extremely sparse. However, existing distributed schemes (e.g., Ring Attention) are designed for dense attention and assume a communication pattern that requires full KV transmission. In sparse modes, this results in significant communication redundancy, wasting valuable network bandwidth.
>
> 3. Difficulty in overlapping computation and communication:
>
> Sparse attention greatly reduces FLOPs, but the communication volume in standard distributed schemes remains constant. This disrupts the balance between computation and communication, especially in scenarios involving inter-node communication, where the reduced computation time making it extremely difficult to hide (overlap) the high communication overhead.
>
> 4. Host Overhead:
>
> In dynamic sparse scenarios, the mask is only confirmed right before the attention calculation at each step. Therefore,, any optimization schemes used for addressing the above problems (e.g., sequence partitioning to balance computation vs. total communication) must have very low complexity. If the optimization algorithm itself is too complex, the overhead on the Host side can become a new bottleneck.
>
> - Potential Solution Directions for Future Work
>
> In response to these challenges, we outline the following potential solution directions for future work:
>
> 1. Computation Imbalance:
>
> - Ulysses-style Scheme: For scenarios with relatively short sequence lengths that do not involve cross-node communication, the standard Ulysses is a straightforward and effective solution. For TopP scenarios where sparsity and computation vary across heads, we can refer to DSV[1] and use a Balanced-Ulysses approach: reordering heads so that the total computational load assigned to each rank after the All-to-All operation is as balanced as possible.
>
> -  Ring-style Scheme: For scenarios with long sequences, we can adapt load balancing based on Ring-style algorithms.
>
>     - TopK: Since the number of K-blocks required for each Q-block is identical (constant computation), we can reuse existing load balancing schemes for Ring Attention, such as the ZigZag algorithm used in causal scenarios.
>     - TopP: The TopP is more complex as the number of K-blocks varies per Q-block. We can flatten the heads into the sequence length dimension, creating an attention map with a sequence length scaled by num_heads, where computation is concentrated on the diagonal. Then, we can employ sequence partitioning algorithms to ensure computational balance, specifically, ensuring that under equal length constraints, the "area" of the attention map (computation load) assigned to each rank is equal.

---

> ### Author Response · Authors · 2025-11-20
>
> ### **W2 & Q1 Distributed Sparse Attention contd.**
>
> 2. Redundant KV Communication:
>
> - Ulysses-style Scheme: Since the baseline communication volume of Ulysses is already low and it is typically used for shorter sequences without cross-node communication, the issue of redundant communication is relatively acceptable.
>
> - Ring-style Scheme: A straightforward approach is for each rank to communicate only the necessary KV blocks (which can be implemented via All-to-All-v). "Necessary KV" refers to the remote KV blocks required by the local Q, excluding local KV. While this eliminates all redundancy, the primitive efficiency of All-to-All-v is relatively low. Note that the total communication volume here is coupled with the sequence partitioning scheme (the load balancing solution mentioned above).
>
> 3. Overlapping computation and communication:
>
> - Ulysses-style Scheme: Given the low communication volume, it can be exposed outside the computation loop.
>
> - Ring-style Scheme: Building on the redundancy elimination described above, we can optimize for overlap by splitting the data required for communication. Communication can be performed in stages and overlapped with the computation of the previous stage.
>     - For TopK, we can split communication and computation by head.
>     - For TopP, we can further split the sequence length of the communicated data to enable fine-grained overlap. This requires precise tuning of the splitting stages and methods according to the specific scenario.
>
> - Hierarchical Communication Optimization Hierarchical optimization schemes that leverage the difference between intra-node and inter-node bandwidth (such as USP and LoongTrain) are also applicable to distributed block sparse scenarios.
>
> 4. Host-side Optimization:
>
> We observe that computation balancing methods are often coupled with the necessary communication volume; the optimal solution for load balancing is not necessarily the one with the minimum communication. Therefore, we need joint optimization of computational balance and total communication volume. Given the dynamic nature of sparse attention, the optimization algorithm must remain lightweight. We intend to use greedy or heuristic approaches rather than seeking a global optimum to minimize Host overhead.
>
> - Generalized Scenarios
>
> For arbitrary dynamic sparse scenarios (where the sparse pattern is not fixed), the problem becomes significantly more complex. Achieving a balance between computation and minimized communication in this context is an NP-hard problem.
>
> Computation cost model: $$f^* = \arg \min_{f} \max_{i} \{ \text{SumArea}(B_i) \}$$
>
> Communication cost model: $$f^* = \arg \min_{f} \max_{i} \{ \text{Sumkv}(B_i) - SumLocalkv(B_i)\}$$
>
> $$\mathrm{s.t.} \quad B_i \subseteq [C_1, C_2, \cdots, C_M], \quad \bigcup_{i=1}^{N} B_i = [C_1, C_2, \cdots, C_M],$$
>       $$B_i \cap B_j = \varnothing, \quad \forall i \neq j, \quad |B_i| = |B_j|, \quad \forall i, j \in \{1, \cdots, N\},$$
>
> Specifically, $B_i$ represents the collection of sparse blocks allocated to the i-th device, while $C_i$ refers to an individual sparse block. We aim to formulate an optimization objective by combining computation and communication costs, seeking a near-optimal solution with acceptable computational complexity.
>
> A more feasible path forward is similar to the design philosophy in mTraining[2]: pursuing Algorithm-System-Kernel Co-design. This involves pre-designing specific mask patterns and performing distributed adaptations and kernel optimizations targeted specifically for those patterns.
>
> Referrence:
>
> [1] DSV: https://arxiv.org/pdf/2502.07590
> @article{tan2025dsv,
>   title={Dsv: Exploiting dynamic sparsity to accelerate large-scale video dit training},
>   author={Tan, Xin and Chen, Yuetao and Jiang, Yimin and Chen, Xing and Yan, Kun and Duan, Nan and Zhu, Yibo and Jiang, Daxin and Xu, Hong},
>   journal={arXiv preprint arXiv:2502.07590},
>   year={2025}
> }
>
> [2] mtraining: https://arxiv.org/abs/2510.18830
> @article{li2025mtraining,
>   title={MTraining: Distributed Dynamic Sparse Attention for Efficient Ultra-Long Context Training},
>   author={Li, Wenxuan and Zhang, Chengruidong and Jiang, Huiqiang and Li, Yucheng and Yang, Yuqing and Qiu, Lili},
>   journal={arXiv preprint arXiv:2510.18830},
>   year={2025}
> }

---

> > ### Comment · Reviewer_1Et8 · 2025-11-21
> >
> > Thank you very much for the detailed rebuttal and for the substantial engineering effort that clearly went into building the benchmark. I appreciate the clarifications and the breadth of experiments.
> >
> > However, after carefully re-evaluating the manuscript together with the rebuttal, I still cannot identify new conceptual insights or previously unknown findings beyond a large-scale benchmarking effort. While the benchmark is certainly useful and the engineering work is nontrivial, the conclusions primarily reaffirm behaviors and trade-offs that are already well understood in the long-context and attention-kernel community.
> >
> > Below I summarize the main conclusions of the paper and explain why, to the best of my knowledge, they correspond to well-established or expected results.
> >
> > 1. **FA3 is the fastest dense attention kernel on H100.** This conclusion is highlighted in Section 3.1. However, FlashAttention-3 was explicitly designed for Hopper (H100) and its architectural features (e.g., TMA, WGMMA, warp specialization), as documented in its own paper and the CUTLASS source code. It is widely acknowledged among practitioners that FA3 is the intended high-performance backend for Hopper GPUs.
> >
> > 2. **cuDNN fused attention has strong performance but limited mask/generalization support**. Also discussed in Section 3.1, this behavior is directly stated in NVIDIA’s cuDNN API documentation.
> >
> > For dense kernel, the author says ``kernel selection should
> > be guided by the target hardware architecture'' which is a natural consequence of GPU architectural differences and aligns with standard practice.
> >
> > 3. **Block-sparse kernels each have known trade-offs (e.g. VSA fast but inflexible; FlashInfer flexible but memory-heavy).**
> > Section 3.2 reports results such as FA2 sparse not supporting 64×64 blocks, VSA not supporting 128×128 or GQA, and FlashInfer exhibiting metadata overhead and lacking backward support. These properties can be directly confirmed from the respective implementations and documentation. The broader conclusion—that specialized kernels are faster and more constrained, while general kernels are more flexible but slower—is an expected pattern in sparse-kernel design rather than a new observation.
> >
> > 4. **Ulysses is good for small scale but is not scalable.** Section 3.3 notes that Ulysses’ scalability is bounded by the number of attention heads and that All-to-All communication introduces cross-node bottlenecks. Both points follow directly from the mathematical formulation of Ulysses and are well documented in prior work (e.g., DeepSpeed Ulysses, Megatron-CP, LoongTrain). These scalability limitations are widely recognized in the ML systems community.
> >
> > 5. **Ring has good scalability but may have load balancing issue with mask.** The paper states that Ring P2P communication transfers fixed KV ratios regardless of masking, which follows directly from its construction. Similar analyses of the trade-offs between Ulysses and ring-based methods are available in public technical discussions (e.g., the linked Zhihu article https://zhuanlan.zhihu.com/p/689067888) as well as in prior papers such as DeepSpeed Ulysses, NVIDIA Megatron-CP, and LoongTrain.
> >
> > Overall, the benchmark offers a useful integration of many existing components and a broad experimental comparison. However, the conclusions appear to validate known behaviors rather than reveal new insights or advance our conceptual understanding of long-context attention.
> >
> > **Therefore, I will maintain my original score. While I appreciate the engineering value of this work, I believe its primary contribution aligns more closely with an open-source benchmarking project than with the expectation of a conference paper.**
> >
> > That said, I remain open to discussion. If there are specific findings or insights that the authors believe provide new conceptual contributions—or if I have overlooked something important—I would be glad to reconsider them.

---

> > > ### Author Response · Authors · 2025-11-26
> > >
> > > We sincerely thank the reviewer for the thoughtful and detailed follow-up. We fully agree that a benchmark paper must contribute more than a collection of experiments. It should unify previously fragmented efforts and reveal insights that are not available without a rigorous, controlled evaluation. We respectfully clarify that **LongCA-Bench aims to address precisely these objectives**, and that several of our findings and contributions go beyond the “expected” behaviors summarized in the review.
> > >
> > > In our opinion, the core objectives of the research in Benchmark and Dataset track of ICLR typically include:
> > > 1. Constructing a unified evaluation system for existing problems.
> > > 2. Uncovering new perspectives on known problems and designing evaluations around them.
> > > 3. Establishing evaluation systems for entirely new problem settings.
> > >
> > > LongCA-Bench follows the first direction precisely. It provides a unified, coherent evaluation framework for long-context attention, covering both kernel-level and distributed context-parallel mechanisms. A unified benchmark of this form is completely missing in the current literature. Moreover, our work identifies several practical but previously unreported challenges and evaluates them from multiple technical perspectives. These include:
> > >
> > > 1. **Establishing a deep and comprehensive understanding of long-context training is inherently challenging**. This research area requires tight co-design between algorithms and systems, substantial engineering effort, and broad expertise. A large body of prior work has already demonstrated both the importance and complexity of long-context training. Whether one aims to understand dense or sparse kernels themselves, or to extend them effectively to large-scale distributed context-parallel training, even running seemingly simple experiments often requires substantial time, effort, and multidisciplinary expertise. As a result, it is challenging for researchers to quickly build a comprehensive understanding of the long-context training landscape; a more systematic organization of existing findings and observed phenomena is needed.
> > >
> > > Of course, we acknowledge that many researchers possess some level of intuition, and certain experts have particularly deep insights, for example, the reviewer’s summarized conclusions such as: "FA3 is the fastest dense attention kernel on H100" or "Ulysses is good for small scale but is not scalable." **However, the insights presented by LongCA-bench go far beyond this:**
> > >
> > > -  For Dense Kernels: Although kernels like FlashAttention and cuDNN are familiar to many researchers, we provide the systematic evaluation of seven mainstream dense kernels across 12 different mask scenarios,  including six heterogeneous patterns that are crucial for real long-context workloads but unsupported by common kernels (e.g., FlashAttention or cuDNN). **Prior work has not examined how kernel performance changes across these diverse masks. We fill this gap and provide the necessary engineering implementations**. Note that even well-known kernels such as cuDNN are not plug-and-play as expected under regular masks.
> > >
> > > -  For Context Parallel Attention:
> > >     - Firstly, the actual performance of a distributed attention mechanism in large-scale distributed settings, especially under context-parallel training, **often diverges significantly from theoretical analysis**. Even if one is very familiar with the mathematical formulation and the overall architecture, this does not guarantee that real training performance will match expectations.
> > >     - Secondly, existing work does not provide a systematic, unified evaluation of mainstream context parallel mechanisms. In training frameworks such as DeepSpeed, or Megatron, it is difficult to directly obtain experimental results for their distributed attention components, **and likewise challenge to integrate alternative methods into these frameworks for fair (e.g., a consistent dispatch/undispatch pipeline and a unified communication implementation), unified comparison**.
> > >     - Thus, most importantly, **LongCA-bench fully reconstructs and optimizes the integrated context parallel mechanisms and reports their actual performance under real dataset distribution. In addition, LongCA-bench's support and experimental reporting for DOCUMENT FULL/CAUSAL context distributed scenarios are entirely new, offering insights that have not been available by previous work**.
> > >   - For Sparse Kernels: This is an active and rapidly evolving area. Comparing different sparse kernels within a unified and controlled environment is non-trivial, and the behaviors reported in documentation often fail to generalize across different scenarios.

---

> > > ### Author Response · Authors · 2025-11-26
> > >
> > > 2. **Lack of a Unified Evaluation Framework**. Although many studies have investigated long-context training, the field still lacks a comprehensive and unified evaluation framework.
> > >     - First, at the methodological level, both kernels and context-parallel mechanisms were originally designed for their own specific settings, resulting in inconsistent experimental scopes and making cross-method comparison difficult.
> > >     - Second, most implementations are not easily reproducible. Context-parallel mechanisms are often deeply embedded with their specific training frameworks, limiting extensibility and preventing systematic comparison when moving to new scenarios. For example, the standard bshd input layouts used in LoongTrain or TransformerEngine USP become impractical for long-context cases due to excessive padding tokens.
> > >
> > > Consequently, long-context training urgently needs a unified and extensible benchmarking framework that can (1) incorporate existing methods, (2) support new scenarios, and (3) refactor and optimize current approaches to enable efficient and fair comparisons.
> > >
> > > 3. **High experimental cost**. Distributed training is a capital-intensive research area, and many groups lack the hardware resources required to run comprehensive large-scale experiments. To reduce the burden on researchers as much as possible, it is essential to provide systematic evaluations and performance reports of current popular baselines, providing reliable references for future research.
> > >
> > > Building on the motivations above, we developed LongCA-Bench to make a meaningful contribution to this research area. Our contributions include:
> > >
> > > 1. **We introduce the first unified and extensible benchmark for long-context attention mechanisms**. It addresses the challenge that existing long-context training methods are tightly coupled to specific scenarios or frameworks, making fair and efficient comparison difficult. LongCA-Bench provides an inclusive, reconfigurable, and extensible platform that not only supports experienced infra researhers (like you), but also offers a clear and consistent entry point for newcomers seeking to understand or engage in this field.
> > > 2. **We systematically reveal, for the first time, the real performance of multiple mainstream attention kernels across up to 12 heterogeneous masking patterns**. This fills a major gap in exising work and provides practical engineering insights, offering researchers a comprehensive and in-depth understanding of kernel behavior under diverse real-world constraints.
> > > 3. **We report, reconstruct, and optimize the performance of large-scale distributed context-parallel mechanisms in realistic training settings**. We fully reconstructs and optimizes mainstream context-parallel strategies, and evaluates them under real training scenarios rather than idealized settings. Moreover, LongCA-Bench is the first to report results for distributed DOCUMENT FULL and DOCUMENT CAUSAL attention. As a reference, the results could significantly lower the experimental costs in this high-cost research domain.
> > >
> > > In summary, the above reflects our understanding of attention mechanisms in long-context training. With LongCA-Bench, we aim to provide a unified and coherent perspective that can guide future work in this area from conceptual insights to engineering practices. We believe this benchmark can offer meaningful value to infrastructure researchers at ICLR and other top venues by serving as a reliable foundation for future advances in long-context modeling.

---

### Official Review · Reviewer_Ehzx · 2025-10-31

**Soundness:** 3
**Presentation:** 3
**Contribution:** 3
**Rating:** 6
**Confidence:** 3

**Summary:**

This paper proposes a unified benchmark framework for evaluating various dense/sparse attention kernels and context-parallel mechanisms in long-sequence scenarios (up to 512K tokens). The framework provides standardized assessments of computational efficiency (in terms of throughput and peak memory usage), scalability, and usability. The evaluation focuses on two main dimensions: (1) attention mask patterns, and (2) sequence length and scale of distributed computation. Firstly, the study performs unified data preprocessing for different attention mask patterns to ensure fair comparisons. Next, it integrates over a dozen attention kernels and incorporates three distributed mechanisms: All-to-All, Ring P2P, and Hybrid. Through these evaluations, the paper draws several insightful conclusions about the computational efficiency pros and cons of various dense and sparse attention kernel implementations. This lays a solid foundation for future research to weigh different backend implementations, explore directions for kernel optimization, further improve kernel implementations, and perform fair comparisons with existing methods.

**Strengths:**

(1) Extensive Method Integration: This work uses a unified interface to integrate 12 representative attention kernels and 5 distributed mechanisms.
(2) Good Scalability: The evaluation is conducted on scenarios with sequence lengths up to 512K and across 96 GPUs.
(3) Practical Insights: Through experimental evaluation, the authors obtain insightful conclusions regarding the impact of mask patterns, the trade-offs between kernel efficiency and usability, and the scalability characteristics of different distributed mechanisms.

**Weaknesses:**

(1) Architectural Limitation: The study is limited to the Hopper architecture and does not discuss the generalization of experimental conclusions to other architectures.

(2) Performance Metric Limitation: The research only focuses on throughput and memory usage as performance metrics. It does not analyze how metrics such as memory bandwidth utilization and inter-node communication load vary over time across different kernels and distributed mechanisms.

**Questions:**

Please provide at lease some convincing explanations for the two points mentioned in the Weaknesses section. I am willing to raise my score if my concerns are addressed.

---

> ### Author Response · Authors · 2025-11-20
>
> We thank the reviewer's useful review of our work. Our responses to the weaknesses and questions you raised are as follows:
>
> ### **W1: Architectural limitation.**
> Thank you for rising this point. From the perspective of benchmark design, our benchmark could in principle be extended to clusters with different hardware architectures. Achieving this, however, requires that the evaluated methods first provide compatible implementations for these hardware architectures; otherwise, substantial engineering adaptation would be necessary for these methods. To improve generalizability within our available resources, we conduct additional experiments for kernels on a hired A800 PCIe 80GB GPU card. At this present, it is not practical for us to apply our benchmark to other hardware architectures like TPU or AMD MI300 systems currently. Sorry for that. The experiments are categorized by sparse kernels and dense kernels. We added experiment results about sparse kernels on A800 GPU in Figure 5 and  experiment results about dense kernels on A800 GPU in Appendix 9.
>
> - Experiments on Dense Kernels
>
> First, document input layout for the cuDNN fused kernel is specifically designed for the Hopper architecture. Therefore, we attempted to expand the input into a basic four-dimensional format to meet the kernel’s layout requirements. However, in most scenarios, the kernel still could not provide support, and we ultimately excluded these experiments. Additionally, FA3 itself is a highly optimized implementation targeting the Hopper architecture, so we excluded the FA3 kernel as well. Similarly, FA2 only supports homogeneous masks, yet it still achieves competitive, often optimal performance in homogeneous mask scenarios.
> It is worth noting that under the sliding window mask configuration, due to the window size design, when the sequence length is 1k, the configuration degenerates into a FULL mask. In such cases, all optimized kernels perform extremely well, which is a reasonable outcome of the experimental setup rather than an inherent advantage of any particular kernel.
>
> FlashMask demonstrates comparable or even superior performance to FA2 across multiple experiments with homogeneous masks, and its main advantage is its native support for the vast majority of heterogeneous mask patterns. In these irregular sparse scenarios, FlashMask achieved the best results in our experiments.
>
> We observed that FlexAttention delivers strong performance during the forward pass, but experiences a noticeable drop in backward performance. This is likely attributable to a combination of compilation optimizations, hardware capabilities, and evaluation methodology. We report this phenomenon faithfully in the paper while maintaining all experimental methods and measurement criteria consistent across all kernels.
>
> Finally, although the Naive Torch implementation and PyTorch SDPA can handle arbitrary mask types, both inevitably face significant computational and memory overhead. As a result, their performance is far inferior to highly optimized kernels in most scenarios. This further suggests that for heterogeneous sparse attention, the fundamental bottleneck lies in kernel-level scheduling strategies and compute pipeline optimization, rather than merely mask support capability.
>
> - Experiments on Sparse Kernels
>
> Experiments are conducted in the same setting as in Section 3.2. We use FlashAttention2 backend in FlashInfer for comparisons and VSA is excluded because it is implemented tailored for Hopper GPUs.
>
> Based on the above experiments, we find similar conclusions with benchmark results in Hopper GPUs.
>
> 1. Flexibility vs. Efficiency. General kernels (e.g., FlexAttention) suffer from high memory overheads due to block mask metadata. In contrast, specialized kernels achieve high speeds but are brittle, lacking support for varied mechanisms like GQA or heterogeneous block sizes (e.g., VSA Triton, FA2 Sparse).
>
> 2. Lack of Training Support. The absence of backward pass support in kernels (e.g., FlashInfer) significantly limits their applicability for training scenarios.
>
> 3. Hardware Utilization.  Hardware-specific kernels demonstrate that tailored optimizations (e.g., FA2 Sparse, FlashInfer) are critical for high performance. However, generic implementations often fail to fully leverage the total performance of the underlying architecture (e.g., VSA Triton).

---

> ### Author Response · Authors · 2025-11-20
>
> ### **W2: Performance Metric Limitation.**
> Thank you for your valuable comments and suggestions. We recognized the missing of communication analysis, and thus we have supplemented LongCA-Bench with a detailed theoretical analysis and targeted experimental results on the communication aspects of the distributed mechanisms we reproduced and optimized. In the theoretical analysis, for each mechanism, we provide a detailed breakdown of the communication operators involved, their invocation timing, and the number of calls. We also distinguish the types of communication data and present the theoretical single-GPU communication volume for each mechanism. Additionally, we conduct a dedicated analysis of the overlap between communication and computation. The details are as follows:
>
> - Ulysess:
>   - FWD: 2 All2All for Q and O, 2 All2All for K and V, the communication volume is $$ \frac{N-1}{N^2}th_{kv}d*2 + \frac{N-1}{N^2}th_{q}d * 2$$
>   - BWD: 2 All2All for dQ and dO, 2 All2All for dK and dV, the communication volume is $$\frac{N-1}{N^2}th_{kv}d*2 + \frac{N-1}{N^2}th_{q}d * 2$$
>   - Overlap: All communications are exposed outside the computation.
> - Ring-AllGather
>   - FWD: 2 AllGather for K and V, the communication volume is $$\frac{N-1}{N}th_{kv}d*2$$
>   - BWD: 2 AllGather for K and V, 2 ReduceScatter for dK and dV, the communication volume is $$\frac{N - 1}{N}th_{kv} d*2 + \frac{N-1}{N}th_{kv}d*2$$
>   - Overlap: All communications are exposed outside the computation.
> - Ring-P2P
>   - FWD: N-1 P2P for K and V, the communication volume is $$\frac{N-1}{N}th_{kv}d*2$$
>   - BWD: N-1 P2P for K and V, N-1 P2P dK and dV, the communication volume is $$\frac{N-1}{N}th_{kv}d*2 + \frac{N-1}{N}th_{kv}d*2$$
>   - Overlap: In theory, the FULL mask scenario achieves communication-computation overlap. Focusing solely on mask-sparse scenarios, computation and communication start simultaneously at each stage across different ranks, with the communication volume remaining unchanged. The portion of the entire pipeline’s communication that can be overlapped is approximately $\frac{\sum_i^mt_m^2/flops}{t/bandwidth}*k$, where t stands for the total sequence length, $t_m^2$ denotes the single computation area, and k merely represents a correction coefficient that exists as a constant for specific settings.
> - USP, we define Ulysses size = Na, ring-p2p size = Np, where Np×Na=N.
>   - FWD: 2 All2All for Q and O, 2 All2All for K and V, Np -1 P2P for K and V, the communication volume is $$\frac{N_a-1}{N_p*N_a^2}th_{kv}d*2 + \frac{N_a-1}{N_p*N_a^2}th_{q}d*2 + \frac{N_p-1}{N_p*N_a}th_{kv}d*2$$
>   - BWD: 2 All2All for dQ and dO, 2 All2All for dK and dV, Np -1 P2P for K and V, Np -1 P2P for dK and dV, the communication volume is $$\frac{N_a-1}{N_p*N_a^2}th_{kv}d*2 + \frac{N_a-1}{N_p*N_a^2}th_{q}d*2 + \frac{N_p-1}{N_p*N_a}th_{kv}d*2 + \frac{N_p-1}{N_p*N_a}th_{kv}d*2$$
>   - Overlap: the analysis remains the same as Ulysses and Ring P2P.
> - LoongTrain, we define intra_ring size = n, window number = k, where n×k=Np.
>   - FWD: 2 All2All for Q and O, 2 All2All for K and V, (n-1)*k P2P for intra K and V, k-1 P2P for inter K and V, the communication volume is $$\frac{N_a-1}{N_p*N_a^2}th_{kv}d*2 + \frac{N_a-1}{N_p*N_a^2}th_{q}d*2 + \frac{(n-1)*k}{N_p*N_a}th_{kv}d*2 + \frac{k-1}{N_p*N_a}th_{kv}d*2$$
>   - BWD: 2 All2All for dQ and dO, 2 All2All for dK and dV, (n-1)*k P2P for intra K and V, k-1 P2P for inter K and V, n*k P2P for intra dK and dV, k P2P for inter dK and dV, the communication volume is $$\frac{N_a-1}{N_p*N_a^2}th_{kv}d*2 + \frac{N_a-1}{N_p*N_a^2}th_{q}d*2 + \frac{(n-1)*k}{N_p*N_a}th_{kv}d*2 + \frac{k-1}{N_p*N_a}th_{kv}d*2 + \frac{1}{N_a}th_{kv}d*2 + \frac{k}{N_p*N_a}th_{kv}d*2$$
>   - Overlap: The forward is consistent with USP. For the backward, the intra and inter communications of K and V are the same as in USP. However, regarding intra dK and dV, unlike USP, which places intra K, V, dK, and dV in a single buffer, LoongTrain separates K, V from dK, dV into two buffers, requiring two separate P2P communications. Thus, in each intra stage of LoongTrain, the intra dK and dV communications are asynchronously initiated only after the completion of computations, overlapping with the correction of dK and dV in the next stage. That said, LoongTrain incurs one additional intra communication for dK and dV, leaving approximately one P2P communication exposed. After the final intra dK and dV communication concludes, LoongTrain initiates an inter dK and dV communication, which overlaps with the subsequent intra computations. However, the dK and dV communication for the last window remains exposed.

---

> ### Author Response · Authors · 2025-11-20
>
> ### **W2: Performance Metric Limitation contd.**
> We conducted additional experiments to measure communication bandwidth utilization for distributed attention mechanisms in LongCA-Bench, using NVIDIA H100 80GB HBM3 devices (with theoretical peak bandwidth of 450 GB/s for intra-node communication and 50 GB/s for inter-node communication). In these experiments, we recorded the actual communication time and data volume for each communication operator and calculated the effective speed of each communication. Furthermore, for the GQA scenario, we distinguished between Q/O and K/V communications. Finally, we computed both the average and maximum transfer speeds for each type of communication. It is worth noting that when both intra-node and inter-node communications of the same type occurred simultaneously, we reported the average speed by combining the intra-node and inter-node measurements, while additionally listing their respective maximum speeds separately.
>
> - 8 GPUs on a single machine
>
> | Speed (Mean, Max)                | All2All Q,O(dQ,dO) | All2All K,V(dK,dV) | AllGath K,V | ReduceScatter dK,dV | Intra P2P K,V(dK,dV) | Inter P2P K,V(dK,dV) |
> |----------------------------------|--------------------|--------------------|-------------|---------------------|----------------------|----------------------|
> | Ulysess                          | (296,323)          | (221,229)          |             |                     |                      |                      |
> | Ring-AllGather                   |                    |                    | (266,314)   | (317,319)           |                      |                      |
> | RingP2P                          |                    |                    |             |                     | (182,327)            |                      |
> | USP(a2a=4, p2p=2)                | (78,282)           | (161,189)          |             |                     | (289,298)            |                      |
> | LoongTrain(a2a=2,p2p=2,window_num=2) | (159,236)      | (110,167)          |             |                     | (270,296)            |                      |
>
> - 16 GPUs across 2 machines
>
> | Speed (Mean, Max)                | All2All Q,O(dQ,dO) | All2All K,V(dK,dV) | AllGather K,V | ReduceScatter dK,dV | Intra P2P K,V(dK,dV) | Inter P2P K,V(dK,dV) |
> |----------------------------------|--------------------|--------------------|---------------|---------------------|----------------------|----------------------|
> | Ulysess                          | (44,63)            | (50,59)            |               |                     |                      |                      |
> | Ring-AllGather                   |                    |                    | (166,229)     | (65,247)            |                      |                      |
> | RingP2P                          |                    |                    |               |                     | (25,293)             | (25,26)              |
> | USP(a2a=8, p2p=2)                | (93,280)           | (176,194)          |               |                     |                      | (17,26)              |
> | LoongTrain(a2a=4,p2p=2,window_num=2) | (76,286)      | (141,181)          |               |                     | (40,297)             | (40,26)              |

---

> ### Author Response · Authors · 2025-11-20
>
> ### **W2: Performance Metric Limitation contd.**
>
> Regarding the memory bandwidth issue you mentioned, we are not entirely sure we have interpreted your concern correctly. Based on our current understanding, however in distributed settings, the impact of memory bandwidth on actual end-to-end performance is very limited, which is why we did not treat it as a standalone evaluation metric. For attention kernels, memory bandwidth typically becomes a dominant factor only in scenarios with low arithmetic intensity or problem sizes too small to fully saturate the compute units. In practice, most optimized attention kernels maximize on-chip data reuse and minimize dependence on global memory accesses, thereby greatly reducing their sensitivity to memory bandwidth. As a result, performance differences between kernels are primarily driven by algorithmic structure and tiling or blocking strategies, rather than by raw memory bandwidth.
>
> For the dense kernels evaluated in our benchmark (FULL and CAUSAL), mainstream optimized implementations (such as FA3, cuDNN fused kernels, and etc.) have undergone extensive targeted optimizations. These kernels typically rely on fine-grained tiling strategies, on-chip shared memory reuse, register-level data rearrangement, and minimization of unnecessary global memory operations, enabling the operator to operate in a high–arithmetic-intensity regime rather than being memory-bandwidth-bound. Therefore, in these dense scenarios, kernel performance is almost unaffected by differences in memory bandwidth.
>
> In heterogeneous sparse scenarios, the main performance bottleneck is also not memory bandwidth, but rather effective scheduling strategies, data layouts under sparse patterns, execution structures that avoid warp divergence, and maintaining high computational utilization under irregular access patterns. In other words, for heterogeneous sparse attention, the key challenges lie in improving computational efficiency, load balancing, and effective utilization of on-chip resources, rather than relying on memory bandwidth itself.

---

### Official Review · Reviewer_nrHk · 2025-11-01

**Soundness:** 4
**Presentation:** 4
**Contribution:** 3
**Rating:** 8
**Confidence:** 5

**Summary:**

This paper proposes LongCA-bench, a unified benchmark for evaluating long-context attention mechanisms in large language model (LLM) training, covering both single-device kernels (dense and sparse) and distributed context parallel strategies. The benchmark addresses critical gaps in existing evaluations—such as incomplete operator comparisons and framework-specific context parallel designs—by integrating 12 dense/sparse kernels, 5 distributed attention mechanisms, and 14 attention mask patterns. It conducts large-scale experiments on up to 96 NVIDIA H100 GPUs, evaluating performance across sequence lengths (up to 512K tokens) and distributed scales. Key findings include hardware-optimized kernels (e.g., FlashAttention-3) outperforming general ones in regular masks, sparse kernels facing limitations in backward computation and flexibility, and hybrid distributed designs (USP, LoongTrain) balancing scalability and efficiency. The work provides actionable guidance for selecting attention mechanisms in ultra-long context training.

**Strengths:**

1. LongCA-bench unifies diverse attention kernels (dense/sparse) and distributed mechanisms under a modular interface, enabling fair cross-method comparisons—addressing the fragmentation of existing evaluations.
2. It systematically explores two understudied but impactful dimensions: 14 attention mask patterns (static/dynamic, regular/heterogeneous) and extreme long sequences (up to 512K) with large-scale distributed training (up to 96 GPUs), filling gaps in prior work.
3. The benchmark uses real-world datasets (Pile, ProLong64K/512K) and realistic sampling strategies, ensuring results reflect actual LLM training scenarios. It also provides open-source code for reproducibility.
4. Beyond performance metrics (TFLOPs, peak memory), the paper reveals trade-offs (e.g., hardware optimization vs. mask compatibility, computation-communication overlap in distributed systems) and identifies bottlenecks (e.g., sparse kernel backward pass inefficiency).

**Weaknesses:**

0. Why no linear attention kernels?
1. The optimized distributed attention mechanisms only support 4 mask patterns (FULL, CAUSAL, FULL/CAUSAL DOCUMENT), excluding heterogeneous and dynamic masks—restricting its applicability to complex long-context tasks.
2. The benchmark excludes FlexAttention from full evaluations due to severe out-of-memory issues, and most sparse kernels lack backward computation support or flexibility (e.g., fixed block sizes), limiting insights into trainable sparse attention.
3. Key findings (e.g., FlashAttention-3’s superiority) are tailored to NVIDIA H100 GPUs, reducing generalizability to other hardware architectures (e.g., AMD GPUs, TPUs).
4. Only 5 distributed strategies are evaluated, with Ring All-Gather’s results omitted due to resource constraints—missing opportunities to compare with emerging context parallel designs.

**Questions:**

see Weaknesses

---

> ### Author Response · Authors · 2025-11-20
>
> We thank the reviewer's useful review of our work. Our responses to the weaknesses and questions you raised are as follows:
>
> ### **W0: Why no linear attention kernels?**
> Thank you for your valuable comments and suggestions.
>
> LongCA-Bench aims to provide a comprehensive evaluation of attention mechanisms built upon the softmax operator, including dense, sparse kernels, and distributed context parallel attention, all of which follow this computational paradigm. On one hand, vanilla attention remains the mainstream attention mechanism adopted by the industry, and its effectiveness has been validated, and its architecture scales well to large-scale distributed training settings, better leveraging the scaling law. On the other hand, the inherent limitations of the softmax operator make  the quadratic computational and storage (i.e., GPU memory) complexity of attention mechanisms increasingly problematic in long-context scenarios. Addressing these issues requires progress from both the operator level and the distributed-systems perspective. Therefore, our LongCA-Bench is designed for both kernel-level and distributed attention-module-level evaluation in terms of softmax-operator-based attention.
>
> In terms of linear attention, recent research on linear attention mainly falls into two directions:
>
> The first direction focuses on the linear attention mechanism itself, including the work such as RWKV [2] and Mamba [3]. RWKV combines features of RNNs [1] and vanilla attention, redesigning attention computation into an RNN-like temporal update mechanism via linear operations. Mamba is built on a selective state space model architecture and also achieves linear computational complexity. It integrates forget gates and block designs into the linear attention framework, significantly improving performance in the scope of linear complexity. Although these linear attention kernels aim to alleviate attention's long-context complexity, they fundamentally the computational mode based on the softmax operator. For this reason, we exclude this type of kernels from our benchmark. We will also consider conducting a unified evaluation of these linear-attention kernels under a consistent setting in future work.
>
> The second direction extends linear attention ideas to large language model (LLM) training, including Lightning Attention [4] and Kimi Linear [5]. Based on extensive experimental studies, MiniMax adopts a hybrid architecture combining linear attention and MoE: every seven Lightning attention layers (a variant of linear attention) are followed by one classic transformer layer. MiniMax also explores specific drop strategies for MoE, thereby reconstructing and optimizing the entire model training architecture. This design comes from a trade-off between training cost, inference cost, and model performance after extensive experimental verification. Considering the state capacity limitations of linear attention, Kimi Linear further extends Gated DeltaNet within the hybrid architecture to achieve finer-grained state adjustment. These work involves restructuring and optimizing the complete model architectures and therefore fall outside the scope of our benchmark. They are better suited for systematic comparison in an end-to-end model-level evaluation.
>
> Reference
>
> [1] Transformers are RNNs: Fast Autoregressive Transformers with Linear Attention
>
> [2] RWKV: Reinventing RNNs for the Transformer Era
>
> [3] Demystify Mamba in Vision: A Linear Attention Perspective
>
> [4] MiniMax-01: Scaling Foundation Models with Lightning Attention
>
> [5] Kimi Linear: An Expressive, Efficient Attention Architecture

---

> ### Author Response · Authors · 2025-11-20
>
> ### **W1: Heterogeneous and dynamic masks are not evaluated in distributed attention mechanisms.**
> Thank you for your valuable comments and suggestions.
>
> First, several state-of-the-art attention kernels, such as FA3 and the cuDNN fused kernel, do not support heterogeneous masks and have limited mask expressive flexibility. For example, if a query sequence q[0,1023] needs to compute attention with two disjoint kv segments (e.g., kv[16,128] and kv[256,512]), the kernel’s input structure cannot represent this mask pattern in a unified manner. In order to execute this computation task, multiple kernel launches need to be scheduled, which relies on support from distributed strategies.
>
> However, current representative context parallelism attention mechanisms, including ring-p2p, usp, and loongtrain, only support full and causal masks. Although we reconstructed and optimized these mechanisms in LongCA-bench to further support full and causal document masks while maintaining efficiency, these mechanisms' architectural designs inherently limit support for heterogeneous masks. Moreover, heterogeneous masks introduce severe computational load imbalance in these mechanisms, which significantly degrades performance in distributed environments.
>
> Designing a universal and efficient context parallelism mechanism to handle heterogeneous masks requires joint optimization at both algorithmic level and  system-engineering level. Of course, one may adopt an all-to-all–based approach by first obtaining the full sequence via All-to-all communication, then applying kernels that support heterogeneous-mask computation, such as Flex Attention or SDPA. However, these kernels typically deliver limited performance, and the approach itself is constrained by the number of heads. As a result, its practical applicability is limited and the overall performance is relatively poor.

---

> ### Author Response · Authors · 2025-11-20
>
> ### **W2: flexAttention issue and lack of sparse backward support.**
> Thank you for your valuable comments and suggestions.
>
> We evaluate FlexAttention separately because generating dynamic block-sparse patterns incurs an $O(S^2)$ block-mask memory overhead. Thus, Table 6 reports only its TFLOPS with 4 heads and a 16K context. Most of its memory cost comes from mask construction, highlighting the need for efficient mask representations in block-sparse settings.
>
> For trainable sparse attention, in light of the growing prevalence of complex, trainable sparse attention mechanisms, such as Native Sparse Attention (NSA) [1] and Deepseek Sparse Attention (DSA) [2] (used in the recent DeepSeek-V3.2-Exp), we conduct supplementary performance experiments for NSA and DSA and extend the analysis by comparing these eight sparse attention kernels in the backward setting. Also, we find that these advanced, trainable methods can be effectively benchmarked within our block sparse attention framework because they all treat sparse attention modeling with uniform block masks. The evaluation and analysis are also added to our revised version. The experimental results of NSA and DSA are posted in Table 10 and Table 11 in our revised version.
>
> - Experimental Settings
>
> For NSA, we follow the setting from the original paper, with 64 query heads and 4 KV heads. In this configuration, each query group (comprising 16 query heads) selects the same KV indices. Each query selects 32 K-blocks with a block size of 64, resulting in a total of 2048 selected tokens. Since the selection attention module dominates the end-to-end overhead and aligns with our block sparse framework, our experiments focus on this module. This configuration maps to q_block_size=16 and k_block_size=64 within our block sparse attention benchmark framework. We conduct both NSA-Triton and NSA-Tilelang in the benchmark, implemented by Triton and Tilelang, respectively.
>
> For DSA, since it utilizes token-level sparse attention and is designed based on DeepseekMLA model architecture, we set the configuration to 128 query heads and 1 KV head, matching DeepseekMLA. Each query group therefore consists of 128 query heads sharing the same KV indices. We set the top-k selection size to 2048 tokens. In our framework, this corresponds to q_block_size=128 and k_block_size=1.  We benchmark this setup using the DSA-Tilelang and DSA-FlashMLA, implemented by Tilelang and CUDA, respectively.
>
> - Analysis
>
> Integrating the analysis of the six block sparse attention mechanisms detailed in our submission with the two representative trainable sparse attention (NSA and DSA) discussed above, we derive the following critical insight about the backward implementation of trainable sparse attention.
>
> There is a lack of backward functionality and significant performance gap between forward and backward passes, highlighting the need for specialized backward implementations and engineering tuning to facilitate the adoption of complex trainable sparse attention. While forward passes for mechanisms like VSA and DSA achieve 500–600 TFLOPS, backward performance drops sharply. Currently, only VSA (and VSA-Triton) reach approx. 300 TFLOPS, whereas DSA-Tilelang reaches approx. 100 TFLOPS. Regarding of the kernel implementations, for scenarios with larger k_block_size (e.g., 64), the overall backward computation logic remains similar to dense attention kernels. However, the fact that the native VSA kernel falls behind its Triton counterpart at context lengths exceeding 64K suggesets that additional bottom-level engineering is required to sustain performance at scale. For scenarios with small k_block_size (e.g., 1), the kernel strategy requires a fundamental structural shift. The standard backward loop order (Outer Loop over K/V, Inner Loop over Q) must be inverted to an Outer Loop over Q and Inner Loop over K/V to efficiently handle the fine-grained sparsity. Ultimately, bridging the performance gap between forward and backward passes is a prerequisite for the efficient training of complex sparse attention models. This requires a shift from generic implementations to specialized backward kernels that apply distinct optimization strategies, ranging from engineering-level tuning for large blocks to fundamental kernel refactoring for small blocks, for the specific method.
>
> Reference
>
> [1] Native Sparse Attention: Hardware-Aligned and Natively Trainable Sparse Attention. https://arxiv.org/abs/2502.11089
>
> [2] https://github.com/deepseek-ai/DeepSeek-V3.2-Exp

---

> ### Author Response · Authors · 2025-11-20
>
> ### **W3: Hardware architecture limitation.**
> Thank you for rising this point. From the perspective of benchmark design, our benchmark could in principle be extended to clusters with different hardware architectures. Achieving this, however, requires that the evaluated methods first provide compatible implementations for these hardware architectures; otherwise, substantial engineering adaptation would be necessary for these methods. To improve generalizability within our available resources, we conduct additional experiments for kernels on a hired A800 PCIe 80GB GPU card. At this present, it is not practical for us to apply our benchmark to other hardware architectures like TPU or AMD MI300 systems currently. Sorry for that. The experiments are categorized by sparse kernels and dense kernels. We added experiment results about sparse kernels on A800 GPU in Figure 5 and  experiment results about dense kernels on A800 GPU in Appendix 9.
>
> - Experiments on Dense Kernels
>
> First, document input layout for the cuDNN fused kernel is specifically designed for the Hopper architecture. Therefore, we attempted to expand the input into a basic four-dimensional format to meet the kernel’s layout requirements. However, in most scenarios, the kernel still could not provide support, and we ultimately excluded these experiments. Additionally, FA3 itself is a highly optimized implementation targeting the Hopper architecture, so we excluded the FA3 kernel as well. Similarly, FA2 only supports homogeneous masks, yet it still achieves competitive, often optimal performance in homogeneous mask scenarios.
> It is worth noting that under the sliding window mask configuration, due to the window size design, when the sequence length is 1k, the configuration degenerates into a FULL mask. In such cases, all optimized kernels perform extremely well, which is a reasonable outcome of the experimental setup rather than an inherent advantage of any particular kernel.
>
> FlashMask demonstrates comparable or even superior performance to FA2 across multiple experiments with homogeneous masks, and its main advantage is its native support for the vast majority of heterogeneous mask patterns. In these irregular sparse scenarios, FlashMask achieved the best results in our experiments.
>
> We observed that FlexAttention delivers strong performance during the forward pass, but experiences a noticeable drop in backward performance. This is likely attributable to a combination of compilation optimizations, hardware capabilities, and evaluation methodology. We report this phenomenon faithfully in the paper while maintaining all experimental methods and measurement criteria consistent across all kernels.
>
> Finally, although the Naive Torch implementation and PyTorch SDPA can handle arbitrary mask types, both inevitably face significant computational and memory overhead. As a result, their performance is far inferior to highly optimized kernels in most scenarios. This further suggests that for heterogeneous sparse attention, the fundamental bottleneck lies in kernel-level scheduling strategies and compute pipeline optimization, rather than merely mask support capability.
>
> - Experiments on Sparse Kernels
>
> Experiments are conducted in the same setting as in Section 3.2. We use FlashAttention2 backend in FlashInfer for comparisons and VSA is excluded because it is implemented tailored for Hopper GPUs.
>
> Based on the above experiments, we find similar conclusions with benchmark results in Hopper GPUs.
>
> 1. Flexibility vs. Efficiency. General kernels (e.g., FlexAttention) suffer from high memory overheads due to block mask metadata. In contrast, specialized kernels achieve high speeds but are brittle, lacking support for varied mechanisms like GQA or heterogeneous block sizes (e.g., VSA Triton, FA2 Sparse).
>
> 2. Lack of Training Support. The absence of backward pass support in kernels (e.g., FlashInfer) significantly limits their applicability for training scenarios.
>
> 3. Hardware Utilization.  Hardware-specific kernels demonstrate that tailored optimizations (e.g., FA2 Sparse, FlashInfer) are critical for high performance. However, generic implementations often fail to fully leverage the total performance of the underlying architecture (e.g., VSA Triton).
>
>
> ### **W4: Missing Ring All-Gather’s results.**
> Thank you for your valuable comments and suggestions. Due to previous limitations in resources and time, we excluded the experimental results for ring all-gather. We have now added the experiments on Ring All-Gather and updated the results in the paper.  We revised Figure 6 and Appendix 7 to update additional experimental results.

---

### Author Response · Authors · 2025-11-30
**Rebuttal Summary and Clarifications for AC Review**

Dear PCs, SACs, ACs, and Reviewers,

We sincerely thank the PCs, SACs, ACs, and Reviewers for your careful evaluation of our submission.

This note provides a concise, author-side summary of our rebuttal. Our work addresses the lack of systematic, fair, and reproducible evaluation of attention kernels and distributed context-parallel strategies for long-context LLM training. The paper proposes a unified and extensible benchmark that integrates representative attention kernels and context-parallel mechanisms, evaluating them across key dimensions such as mask patterns and sequence length/distributed scale. Comprehensive large-scale experiments reveal method-specific trade-offs and provide practical guidance for designing and deploying attention mechanisms in long-context training.

In response to the discussion, we have added several supplementary appendices and revised parts of the paper for completeness and thoroughly address the Reviewers' concerns raised during the review process.

We believe our rebuttal thoroughly addresses the Reviewers' concerns and would lead them to form a more positive assessment of our paper. However, due to the data-leak issue on openreview.net, ACs are asked to estimate how the reviewer's impressions would have changed, incursing additional work. To assist with this process, we provide this concise summary to help the AC efficiently and accurately assess the paper and the reviewers’ feedback during the decision-making period.

For the AC’s convenience, the following parts summarize (1) the key strengths highlighted by the Reviewers and (2) how our rebuttal addresses their concerns. Detailed responses can be found in the corresponding sections of the discussion.

Thank you again for your time and consideration.

Best regards,

The Authors of Submission 3282.

---

> ### Author Response · Authors · 2025-11-30
> **Strengths**
>
> The strengths of our paper are highlighted by the reviewers:
>
> - **Unified and comprehensive benchmarking framework:** Provides a **modular, extensible interface** that unifies **dense/sparse attention kernels and multiple distributed mechanisms**, resolving fragmentation in prior evaluations (nrHk, Ehzx, 1Et8).
> - **Extensive coverage and scalability:** Conducts **systematic evaluations across 14 mask patterns**, sequence lengths up to **512K**, and **large-scale distributed settings (up to 96 GPUs)**, offering unprecedented coverage of long-context scenarios (nrHk, Ehzx, 1Et8).
> - **Realistic and reproducible experimental design:** Uses **real-world datasets** and practical sampling strategies, reporting consistent metrics (throughput, memory) with **open-source, reproducible methodology** (nrHk, 1Et8).
> - **Practical insights and diagnostics:** Reveals **critical performance trade-offs** (e.g., kernel optimization vs. mask compatibility, computation–communication overlap) and identifies **key bottlenecks** such as sparse-kernel backward inefficiencies, providing actionable guidance for practitioners (nrHk, Ehzx, 1Et8).

---

> ### Author Response · Authors · 2025-11-30
> **Summary for the response to weaknesses and questions of reviewer nrHk**
>
> - **W0: Why no linear attention kernels?** LongCA-Bench focuses specifically on attention mechanisms built on the softmax operator, including dense kernels, sparse kernels, and distributed context-parallel methods at the module level. These families of methods constitute the dominant, scalable solutions used in current tier-1 distributed LLM training, yet they continue to face quadratic complexity in long-context settings. In contrast, Linear-attention research either (1) redesigns the entire model architecture (e.g., RWKV, Mamba) or (2) restructures the end-to-end LLM training pipeline (e.g., Lightning, Kimi Linear). Because these approaches fundamentally differ from softmax-based attention, they fall outside the scope of our kernel-level and module-level benchmarking. They merit a separate, end-to-end model-level evaluation in future work.
> - **W1: Heterogeneous and dynamic masks are not evaluated in distributed attention mechanisms.** Current high-performance attention kernels (e.g., FA3, cuDNN fused) cannot express heterogeneous masks and require multiple launches. Existing context-parallel methods (ring-p2p, USP, and LoongTrain, etc) inherently support only full or causal masks and exhibit severe load imbalance when handling heterogeneous masks. Achieving a universal and efficient solution, thus, requires a joint algorithm–system redesign. While an all-to-all strategy combined with heterogeneous-mask kernels is theoretically possible, it delivers poor performance and scales poorly in practice, limiting its real-world applicability.
> - **W2: flexAttention issue and lack of sparse backward support.** We evaluate FlexAttention separately because dynamic block-sparse masks incur O(S²) mask-memory overhead. **For trainable sparse attention, we add comprehensive NSA and DSA benchmarks and show they fit naturally into our block-sparse framework.** Our extended analysis reveals a major gap between forward and backward performance across all trainable sparse methods, underscoring the need for specialized backward kernels, engineering-level tuning for large blocks and fundamental kernel redesign for fine-grained sparsity.
> - **W3:  Hardware architecture limitation.** Our benchmark can, in principle, be extended to other hardware architectures, but doing so requires that evaluated kernels already support those platforms; otherwise, substantial re-engineering is unavoidable. **Within our available resources, we added A800 experiments for both dense and sparse kernels.** The results largely align those on Hopper: hardware-specialized kernels achieve the best efficiency but offer limited flexibility; general kernels (e.g., FlexAttention) incur high mask- metadata overhead; and many sparse kernels still lack usable backward implementations. Overall, heterogeneous sparse attention remains fundamentally constrained by kernel-level scheduling and architecture-specific optimizations.
> - **W4: Missing ring All-Gather’s results.** We have added the Ring All-Gather experiments and updated the corresponding results.

---

> ### Author Response · Authors · 2025-11-30
> **Summary for the response to weaknesses and questions of reviewer Ehzx**
>
> - **W1: Architectural limitation.** Consistent with our response to Reviewer nrHk’s W3, our benchmark can, in principle, extend to other hardware architectures. **Within our available resources, we conducted A800 experiments for both dense and sparse kernels.**
> - **W2:Performance metric limitation. We added a comprehensive communication analysis to LongCA-Bench, including theoretical breakdowns (operators, timings, call counts, communication volume, and computation-communication overlap behavior) for all distributed mechanisms, as well as new empirical measurements of actual bandwidth utilization on H100 GPUs.** The revised paper now provides a comprehensive view of communication cost and overlap for Ulysses, Ring-AllGather, Ring-P2P, USP, and LoongTrain. Based on our evaluation and analysis, memory bandwidth has minimal impact on attention performance. Performance differences are mainly due to kernel design, scheduling, data layout, and on-chip utilization rather than raw memory limits.

---

> ### Author Response · Authors · 2025-11-30
> **Summary for the response to weaknesses and questions of reviewer 1Et8**
>
> - **W1 & Q2: Limited analysis. We have significantly expanded our analysis by adding detailed communication profiling and empirical measurements for distributed context-parallel attentions, benchmarking two complex trainable sparse kernels (NSA and DSA) alongside six block sparse kernels, and extending dense kernel experiments to A800 GPUs. Key insights include:** (1) Multi-head parallelism and algorithm–hardware co-design are essential for achieving high performance; (2) sparse kernels are highly sensitive to block size and hardware utilization, with backward passes substantially lagging forward passes and requiring specialized kernel engineering; (3) dense kernels are compute-bound, with performance heavily influenced by compiler- and hardware-specific optimizations; and (4) across both sparse and dense attention settings, performance is dominated by computation–communication overlap, scheduling, and on-chip resource utilization, rather than by raw memory bandwidth.
> - **W2 & Q1: Distributed sparse attention.** We did not cover distributed sparse attention due to several intrinsic challenges, including severe per-head computation imbalance, redundant KV communication, difficulty achieving computation and communication overlap, and substantial host-side overhead caused by dynamic masks. We outline future directions to address these challenges, including load-balanced partitioning (e.g., Balanced-Ulysses, Ring-style), selective KV communication, fine-grained overlap strategies, lightweight host-side heuristic algorithms, and algorithm–system–kernel co-design with pre-defined mask patterns to enable efficient execution of dynamic sparse attention in distributed settings.
> - **Rebuttal discussion about the contribution.** We appreciate the reviewer’s thoughtful feedback and fully agree that a benchmark should deliver more than a collection of experiments. It should unify fragmented efforts and reveal insights that are otherwise inaccessible. LongCA-Bench is designed precisely for this purpose. It provides **the first unified and extensible evaluation framework** for long-context attention, covering dense kernels, sparse kernels, and distributed context-parallel mechanisms under realistic training settings. Unlike prior work that evaluates methods in isolation, LongCA-Bench systematically reconstructs, standardizes, and tests mainstream approaches across diverse masks, heterogeneous scenarios, and actual distributed workflows. This unified design **reveals previously unreported practical challenges and performance behaviors.** For example, kernel performance under **12 real-world masking patterns**, the **discrepancies between theoretical and actual throughput** in context-parallel mechanisms, and **the first reported results for distributed DOCUMENT FULL/CAUSAL attention**. Given the heavy engineering cost and lack of standardized methodology in current long-context research, LongCA-Bench provides **a coherent, reproducible, and comprehensive foundation** that lowers the barrier to rigorous experimentation and helps both newcomers and experts obtain a clearer understanding of the long-context training landscape.

---

### Meta-Review · Area_Chair_zJuJ · 2026-01-13

**Summary:**

This paper introduces LongCA-Bench, a unified benchmark for long-context attention that spans (i) single-GPU dense kernels, (ii) block-sparse kernels, and (iii) distributed context parallel attention. The main value is the unification + coverage: standardized data/mask interfaces, apples-to-apples comparisons across many mask patterns, and scaling studies up to very long context and large GPU counts. Overall, I agree with the reviewers that this benchmark fills a real gap for practitioners trying to navigate the kernel vs. distributed-parallelism design space.

**Reviewer Concerns:**

The main pushback in the reviews is not about correctness, but about “what is the conceptual contribution”: one reviewer views the conclusions as largely confirming expected behaviors, and notes missing coverage of some critical regimes (e.g., distributed sparse attention, heterogeneous/dynamic masks in distributed settings).

I think the authors did a nice job in rebuttal

## Addressed concerns:

the authors added (1) missing Ring All-Gather results, (2) a more explicit communication analysis for context parallel mechanisms (including overlap and measured bandwidth/utilization), and (3) additional experiments beyond H100 (A800) to partially address hardware generality. They also expanded the sparse-kernel section with additional representative trainable sparse kernels (and highlighted the forward/backward gap), which improves the benchmark’s usefulness.



## Remaining concerns:

1. the benchmark still does not cover distributed sparse attention (arguably the most practically hard and important case), and distributed mechanisms still largely focus on FULL/CAUSAL/DOCUMENT-style masks (heterogeneous/dynamic masks remain unaddressed).

2. the paper is strongest as an engineering + standardization contribution, and the authors should make that framing crisp (benchmarks can be valuable even when they validate known intuitions, but the paper should be careful not to oversell novelty of the findings).

**Reviewer Scores:**

I think all reviewers would likely stay or slightly bump up their scores if a rebuttal phase had happened. Which would likely result in an average rating > 6.

---

### Decision · Program_Chairs · 2026-01-26

Accept (Poster)